# TUNING FREQUENCY BIAS IN NEURAL NETWORK TRAINING WITH NONUNIFORM DATA

**Annan Yu**
Cornell University
ay262@cornell.edu

**Yunan Yang**
ETH Zürich
yyn0410@gmail.com

**Alex Townsend**
Cornell University
townsend@cornell.edu

## ABSTRACT

Small generalization errors of over-parameterized neural networks (NNs) can be partially explained by the frequency bias phenomenon, where gradient-based algorithms minimize the low-frequency misfit before reducing the high-frequency residuals. Using the Neural Tangent Kernel (NTK), one can provide a theoretical analysis for training where data are drawn from constant or piecewise-constant probability densities. Since most training data sets are not drawn from such distributions, we use the NTK model and a data-dependent quadrature rule to theoretically quantify the frequency bias of NN training given nonuniform data. By replacing the loss function with a selected Sobolev norm, we can amplify or dampen the intrinsic frequency bias in NN training.

## 1 INTRODUCTION

Neural networks (NNs) are often trained in supervised learning on a small data set. They are observed to provide accurate predictions for a large number of test examples that are not seen during training. A mystery is how training can achieve small generalization errors in an overparameterized NN and a so-called "double-descent" risk curve (Belkin et al., 2019). In recent years, a potential answer has emerged called "frequency bias," which is the phenomenon that in the early epochs of training, an overparameterized NN finds a low-frequency fit of the training data while higher frequencies are learned in later epochs (Rahaman et al., 2019; Yang & Salman, 2019; Xu, 2020). In addition to generalization errors, it is often useful to understand the convergence rate for each spectral component of the data mismatch to study the robustness of the NN under noises. Currently, frequency bias is theoretically understood via the Neural Tangent Kernel (NTK) (Jacot et al., 2018) for uniform training data (Arora et al., 2019; Cao et al., 2019; Basri et al., 2019) and data distributed according to a piecewise-constant probability measure (Basri et al., 2020). However, most training data sets in practice are highly clustered and not uniform. Yet, frequency bias is still observed during NN training (Fridovich-Keil et al., 2021), even though the theory is absent. This paper proves that frequency bias is present when there is nonuniform training data by using a new viewpoint based on a data-dependent quadrature rule. We use this theory to propose new loss functions for NN training that accelerate its convergence and improve stability with respect to noise.

An NN function is a map $\mathcal{N} : \mathbb{R}^d \to \mathbb{R}$ given by

$$\mathcal{N}(\mathbf{x}) = \mathbf{W}_{N_L} \sigma \left( \mathbf{W}_{N_L-1} \sigma \left( \cdots \left( \mathbf{W}_2 \sigma \left( \mathbf{W}_1 \mathbf{x} + \mathbf{b}_1 \right) + \mathbf{b}_2 \right) + \cdots \right) + \mathbf{b}_{N_L-1} \right) + \mathbf{b}_{N_L},$$

where $\mathbf{W}_i \in \mathbb{R}^{m_i \times m_{i-1}}$ are weights, $m_0 = d$, $\mathbf{b}_i \in \mathbb{R}^{m_i}$ are biases, and $N_L$ is the number of layers. Here, $\sigma$ is the activation function applied entry-wise to a vector, i.e., $\sigma(\mathbf{a})_j = \sigma(\mathbf{a}_j)$. In this paper, we consider ReLU NNs, which are NNs for which $\sigma$ is the ReLU function given by $\text{ReLU}(t) = \max(t, 0)$. Since $\text{ReLU}(\alpha t) = \alpha \text{ReLU}(t)$ for any $\alpha > 0$, we assume that the input of the NN is normalized so that $\mathbf{x} \in \mathbb{S}^{d-1}$. To introduce a continuous perspective, we assume that there is an underlying target function $g : \mathbb{S}^{d-1} \to \mathbb{R}$ and the training samplers $\mathbf{x}_i$'s follow distribution $\mu(\mathbf{x})$. Given training data $\{(\mathbf{x}_i, y_i)\}_{i=1}^n$ drawn from $g$, where $\mathbf{x}_i \in \mathbb{S}^{d-1}$ and $y_i \approx g(\mathbf{x}_i)$, our goal is to train the NN, in a way that is robust when the sampling of $y_i$ from $g(\mathbf{x}_i)$ is contaminated by noises, so that $\mathcal{N}$ uniformly approximates $g$ on $\mathbb{S}^{d-1}$. One standard training procedure is a gradient-based optimization algorithm that minimizes the residual in the squared $L^2(d\mu)$ norm, i.e.,

$$\Phi(\mathbf{W}) = \frac{A_d}{2} \int_{\mathbb{S}^{d-1}} |g(\mathbf{x}) - \mathcal{N}(\mathbf{x}; \mathbf{W})|^2 d\mu(\mathbf{x}) \approx \frac{A_d}{2n} \sum_{i=1}^{n} |y_i - \mathcal{N}(\mathbf{x}_i; \mathbf{W})|^2, \qquad (1)$$

where $A_d$ is the Lebesgue measure of the hypersphere $\mathbb{S}^{d-1}$ and $\mathbf{W}$ represents the weights and bias terms. Similar to most theoretical studies investigating frequency bias, we restrict ourselves to 2-layer NNs (Arora et al., 2019; Basri et al., 2019; Su & Yang, 2019; Cao et al., 2019).

To study NN training, it is common to consider the dynamics of $\Phi(\mathbf{W})$ as one optimizes the coefficients in $\mathbf{W}$. For example, the gradient flow of the NN weights is given by $\frac{d\mathbf{W}}{dt} = -\frac{\partial \Phi}{\partial \mathbf{W}}$. Define the residual $z(\mathbf{x}; \mathbf{W}) = g(\mathbf{x}) - \mathcal{N}(\mathbf{x}; \mathbf{W})$. Applying gradient flow with the population loss gives us

$$\frac{dz(\mathbf{x}; \mathbf{W})}{dt} = -A_d \int_{\mathbb{S}^{d-1}} K(\mathbf{x}, \mathbf{x}'; \mathbf{W}) z(\mathbf{x}'; \mathbf{W}) d\mu(\mathbf{x}'), \tag{2}$$

where $K(\mathbf{x}, \mathbf{x}'; \mathbf{W}) = \left\langle \frac{\partial \mathcal{N}(\mathbf{x}; \mathbf{W})}{\partial \mathbf{W}}, \frac{\partial \mathcal{N}(\mathbf{x}'; \mathbf{W})}{\partial \mathbf{W}} \right\rangle$. Under the assumptions that the weights do not change much during training, one can consider the NTK given the underlying time-independent distribution of $\mathbf{W}$, i.e., $K^\infty(\mathbf{x}, \mathbf{x}') = \mathbb{E}_{\mathbf{W}}[K(\mathbf{x}, \mathbf{x}'; \mathbf{W})]$ (Du et al., 2018). Based on eq. (2), one can understand the decay of the residual by studying the reproducing kernel Hilbert space (RKHS) through a spectral decomposition of the integral operator $\mathcal{L}$ defined by $(\mathcal{L}z)(\mathbf{x}) = \int K^\infty(\mathbf{x}, \mathbf{x}') z(\mathbf{x}') d\mu(\mathbf{x}')$. Most results in the literature require $\mu(\mathbf{x})$ to be the uniform distribution over the sphere so that the eigenfunctions of $\mathcal{L}$ are spherical harmonics and the eigenvalues have explicit forms (Cao et al., 2019; Basri et al., 2019; Scetbon & Harchaoui, 2021). These explicit formulas for the eigenvalues and eigenfunctions of $\mathcal{L}$ rely on the Funk–Hecke theorem, which provides a formula allowing one to express an integral over a hypersphere by an integral over an interval (Seeley, 1966). The frequency bias of NN training can be explained by the fact that low-degree spherical harmonic polynomials are eigenfunctions of $\mathcal{L}$ associated with large eigenvalues (Basri et al., 2019). Thus, for uniform training data, the optimization of the weights and biases of an NN tends to fit the low-frequency components of the residual first.

When $\mu(\mathbf{x})$ is nonuniform, it is difficult to analyze the spectral properties of $\mathcal{L}$ and thus the frequency bias properties of NN training. Since the Funk–Hecke formula no longer holds, there are only a few special cases where frequency bias is understood (Williams & Rasmussen, 2006, Sec. 4.3). Although one may derive asymptotic bounds for the eigenvalues (Widom, 1963; 1964; Bach & Jordan, 2002), it is hard to obtain formulas for the eigenfunctions, and one usually relies on numerical approximations (Baker, 1977). For the ReLU-based NTK, Basri et al. (2020) provided explicit eigenfunctions assuming that the $\mu(\mathbf{x})$ is piecewise constant on $\mathbb{S}^1$, but this analysis does not generalize to higher dimension. To study the frequency bias of NN training, one needs to understand both the eigenvalues and eigenfunctions of $\mathcal{L}$, and this remains a significant challenge for a general $\mu(\mathbf{x})$ due to the absence of the Funk–Hecke formula.

To overcome this challenge, we take a different point-of-view. While it is standard to discretize the integral in eq. (1) using a Monte Carlo-like average, we discretize it using a data-dependent quadrature rule where the nodes are at the training data. That is, we investigate the frequency bias of NN training when minimizing the residual in the standard squared $L^2$ norm:

$$\widetilde{\Phi}(\mathbf{W}) = \frac{1}{2} \int_{\mathbb{S}^{d-1}} |g(\mathbf{x}) - \mathcal{N}(\mathbf{x}; \mathbf{W})|^2 d\mathbf{x} \approx \frac{1}{2} \sum_{i=1}^{n} c_i |y_i - \mathcal{N}(\mathbf{x}_i; \mathbf{W})|^2, \tag{3}$$

where $c_1, \ldots, c_n$ are the quadrature weights associated with the (nonuniform) input data $\mathbf{x}_1, \ldots, \mathbf{x}_n$. If $\mathbf{x}_1, \ldots, \mathbf{x}_n$ are drawn from a uniform distribution over the hypersphere, then one can select $c_i = A_d/n$ for $1 \le i \le n$; otherwise, one can choose any quadrature weights so that the integration rule is accurate (see section 4.2). If $\mathbf{x}_1, \ldots, \mathbf{x}_n$ are drawn independently at random from $\mu(\mathbf{x})$, then it is often reasonable to select $c_i = 1/(np(\mathbf{x}_i))$, where $d\mu(\mathbf{x}) = p(\mathbf{x})d\mathbf{x}$. While $c_1, \ldots, c_n$ depend on $\mathbf{x}_1, \ldots, \mathbf{x}_n$, the continuous expression for $\widetilde{\Phi}(\mathbf{W})$ is always unaltered in eq. (3). Therefore, we can use the Funk–Hecke formula to analyze the eigenvalues and eigenfunctions of $\tilde{\mathcal{L}}$ defined by $(\tilde{\mathcal{L}}z)(\mathbf{x}) = \int_{\mathbb{S}^{d-1}} K^\infty(\mathbf{x}, \mathbf{x}') z(\mathbf{x}') d\mathbf{x}'$, revealing the frequency bias. We address that by choosing eq. (3) as a loss function instead of eq. (1), we are enforcing the frequency bias of NN training, whereas eq. (1) does not ensure such spectral property (see Section 6.1 for an illustration).

To further tune the NN frequency bias during training, we also propose to minimize the residual in a squared Sobolev $H^s$ norm for a carefully selected $s \in \mathbb{R}$. Unlike the $L^2$ norm (the case of $s = 0$), the $H^s$ norm for $s \ne 0$ has its own frequency bias. For $s > 0$, $H^s$ penalizes high frequencies more than low, while for $s < 0$, low frequencies are penalized the most. We implement the squared $H^s$ norm using a quadrature rule, which induces a different integral operator $\mathcal{L}_s$. We

analyze the eigenvalues and eigenfunctions of $\mathcal{L}_s$, and consequently, the frequency bias in the NN training using the Funk–Hecke formula. Given our new understanding of frequency bias, we select $s$ so that the $H^s$ norm amplifies, dampens, counterbalances, or reverses the natural frequency bias from an overparameterized NN training.

**Contributions.** Here are our three main contributions to analyzing and tuning NN frequency bias.

(1) From our quadrature point-of-view, we analyze the frequency bias in training a 2-layer overparameterized ReLU NN with nonuniform training data. In Theorem 2, we show that the theory of frequency bias in Basri et al. (2019) for uniform training data continues to hold in the nonuniform case up to quadrature errors. In Theorem 3, we provide control of the quadrature errors.

(2) We use our understanding of frequency bias to modify the usual squared $L^2$ loss function to a squared $H^s$ norm. By selecting $s$, we can amplify or dampen the intrinsic frequency bias in NN training, accelerate the convergence of gradient-based optimization procedures, and separate out noises of particular frequencies.

(3) A potential issue with the $H^s$ norm is the difficulties of implementing with high-dimensional training data. Using an image dataset of dimension $28^2 = 784$, we show how to use an encoder-decoder architecture to implement a practical version of the squared $H^s$ norm loss and adjust the frequency bias in NN training to suppress noises of different frequencies (see Figure 4).

## 2 PRELIMINARIES AND NOTATION

For $d > 1$, let $g : \mathbb{S}^{d-1} \to \mathbb{R}$ be a square-integrable function defined on $\mathbb{S}^{d-1}$. The function $g$ can be expressed in a spherical harmonic expansion given by

$$g(\mathbf{x}) = \sum_{\ell=0}^{\infty} \sum_{p=1}^{N(d,\ell)} \hat{g}_{\ell,p} Y_{\ell,p}(\mathbf{x}), \qquad \hat{g}_{\ell,p} = \int_{\mathbb{S}^{d-1}} g(\mathbf{x}) Y_{\ell,p}(\mathbf{x}) d\mathbf{x}, \qquad (4)$$

where $Y_{\ell,p}$ is the spherical harmonic basis function of degree $\ell$ and order $p$ (Dai & Xu, 2013). Here, $N(d,\ell)$ is the number of spherical harmonic functions of degree $\ell$ so that $N(d,0) = 1$ and $N(d,\ell) = \frac{(2\ell+d-2)\Gamma(\ell+d-2)}{\Gamma(\ell+1)\Gamma(d-1)}$ for $\ell \geq 1$. The set $\{Y_{\ell,p}\}_{\ell \geq 0, 1 \leq p \leq N(d,\ell)}$ is an orthonormal basis for $L^2(\mathbb{S}^{d-1})$. Let $\mathcal{H}_\ell^d$ be the span of $\{Y_{\ell,p}\}_{p=1}^{N(d,\ell)}$, and $\Pi_\ell^d = \bigoplus_{j=0}^{\ell} \mathcal{H}_j^d$ be the space of spherical harmonics of degree $\leq \ell$.

Given distinct training data $\{\mathbf{x}_i\}_{i=1}^n$ from $\mathbb{S}^{d-1}$ and evaluations $y_i = g(\mathbf{x}_i)$ for $1 \leq i \leq n$, our goal is to understand the intrinsic frequency bias behavior of training a 2-layer ReLU NN given by

$$\mathcal{N}(\mathbf{x}) = \frac{1}{\sqrt{m}} \sum_{r=1}^{m} a_r \text{ReLU}(\mathbf{w}_r^\top \mathbf{x} + b_r), \qquad \text{ReLU}(t) = \max(t, 0), \qquad (5)$$

where $m$ is the number of hidden neurons, $\mathbf{w}_1, \ldots, \mathbf{w}_m$ and $a_1, \ldots, a_m$ are weights, and $b_1, \ldots, b_m$ are biases. We use the same setup as in (Basri et al., 2020): assuming that (1) $\mathbf{w}_1, \ldots, \mathbf{w}_m$ are initialized independently and identically distributed (i.i.d.) from Gaussian random variables with covariance matrix $\kappa^2 \mathbf{I}$, where $\kappa > 0$, (2) the biases are initialized to 0, and (3) $a_1, \ldots, a_m$ are initialized i.i.d. as $+1$ with probability $\frac{1}{2}$ and $-1$ otherwise, and $\{a_r\}$ are not updated during training.

We use a gradient-based optimization scheme to train for the weights and biases and aim to minimize the residual defined by a symmetric positive definite (SPD) matrix $\mathbf{P}$, which can be written as

$$\Phi_{\mathbf{P}}(\mathbf{W}) = \frac{1}{2}(\mathbf{y} - \mathbf{u})^\top \mathbf{P}(\mathbf{y} - \mathbf{u}), \qquad (6)$$

where $\mathbf{y} = (g(\mathbf{x}_1), \ldots, g(\mathbf{x}_n))^\top$ and $\mathbf{u} = (\mathcal{N}(\mathbf{x}_1), \ldots, \mathcal{N}(\mathbf{x}_n))^\top$. For example, we have $\mathbf{P} = A_d n^{-1} \mathbf{I}$ in eq. (1) and $\mathbf{P} = \text{diag}(c_1, \ldots, c_n)$ in eq. (3). Recall that $\mathbf{W}$ represents all the weights and biases of the NN. Given the loss function, we train the NN based on the gradient descent algorithm:

$$\mathbf{w}_r(k+1) - \mathbf{w}_r(k) = -\eta \frac{\partial \Phi_{\mathbf{P}}}{\partial \mathbf{w}_r}, \qquad b_r(k+1) - b_r(k) = -\eta \frac{\partial \Phi_{\mathbf{P}}}{\partial b_r}, \qquad 1 \leq r \leq m, \quad (7)$$

where $k$ is the iteration number and $\eta > 0$ is the learning rate. The matrix $\mathbf{P}$ induces an inner product $\langle \boldsymbol{\xi}, \boldsymbol{\zeta} \rangle_{\mathbf{P}} = \boldsymbol{\xi}^\top \mathbf{P} \boldsymbol{\zeta}$, which leads to a finite-dimensional Hilbert space with the

norm $\|\boldsymbol{\xi}\|_{\mathbf{P}} = \sqrt{\langle \boldsymbol{\xi}, \boldsymbol{\xi} \rangle_{\mathbf{P}}}$. Given a matrix $\mathbf{A} \in \mathbb{R}^{n \times n}$, we define its operator norm $\|\mathbf{A}\|_{\mathbf{P}} = \sup_{\boldsymbol{\xi} \in \mathbb{R}^n, \|\boldsymbol{\xi}\|_{\mathbf{P}}=1} \|\mathbf{A}\boldsymbol{\xi}\|_{\mathbf{P}}$. We also define a finite positive number that depends on $\mathbf{P}$:

$$M_{\mathbf{P}} = \sup_{\boldsymbol{\xi} \in \mathbb{R}^n, \|\boldsymbol{\xi}\|_2=1} \|\boldsymbol{\xi}\|_{\mathbf{P}} = \sup_{\boldsymbol{\xi} \in \mathbb{R}^n \setminus \{\mathbf{0}\}} \sqrt{\frac{\boldsymbol{\xi}^\top \mathbf{P} \boldsymbol{\xi}}{\boldsymbol{\xi}^\top \boldsymbol{\xi}}} = \sup_{\boldsymbol{\zeta} \in \mathbb{R}^n \setminus \{\mathbf{0}\}} \sqrt{\frac{\boldsymbol{\zeta}^\top \mathbf{P}^{1/2} \mathbf{P} \mathbf{P}^{1/2} \boldsymbol{\zeta}}{\boldsymbol{\zeta}^\top \mathbf{P}^{1/2} \mathbf{P}^{1/2} \boldsymbol{\zeta}}} = \sup_{\boldsymbol{\zeta} \in \mathbb{R}^n, \|\boldsymbol{\zeta}\|_{\mathbf{P}}=1} \|\mathbf{P}\boldsymbol{\zeta}\|_2 . \quad (8)$$

Note that by the third expression in eq. (8), we also have $M_{\mathbf{P}} = \left\|\mathbf{P}^{1/2}\right\|_2 = \sqrt{\|\mathbf{P}\|_2}$. Furthermore, we define the matrix $\mathbf{H}^\infty \in \mathbb{R}^{n \times n}$ by

$$H_{ij}^\infty = \mathbb{E}_{\mathbf{w} \sim \mathcal{N}(\mathbf{0}, \kappa^2 \mathbf{I})} \left[ \frac{\mathbf{x}_i^\top \mathbf{x}_j + 1}{2} \mathbb{I}_{\{\mathbf{w}^\top \mathbf{x}_i, \mathbf{w}^\top \mathbf{x}_j \geq 0\}} \right] = \frac{(\mathbf{x}_i^\top \mathbf{x}_j + 1)(\pi - \arccos(\mathbf{x}_i^\top \mathbf{x}_j))}{4\pi}. \quad (9)$$

Note that due to the introduction of the biases, $\mathbf{H}^\infty$ is slightly different than the one in (Du et al., 2018; Arora et al., 2019). In fact, in contrast with (Du et al., 2018), $\mathbf{H}^\infty$ defined in eq. (9) is SPD regardless of the distribution of the training data, as shown in the supplementary material.

**Proposition 1.** *If $\mathbf{x}_1, \ldots, \mathbf{x}_n$ are distinct, then $\mathbf{H}^\infty$ in eq. (9) is SPD.*

As a consequence of Proposition 1, the matrix $\mathbf{H}^\infty \mathbf{P}$ has positive eigenvalues, which we denote by $\lambda_{n-1} \geq \cdots \geq \lambda_0 > 0$. In fact, let $\boldsymbol{\Lambda} = \mathrm{diag}(\lambda_0, \ldots, \lambda_{n-1})$. Then, $\mathbf{H}^\infty \mathbf{P}$ is self-adjoint in $(\mathbb{R}^n, \langle \cdot, \cdot \rangle_{\mathbf{P}})$ (see appendix A) and can be diagonalized as

$$\mathbf{H}^\infty \mathbf{P} = \mathbf{P}^{-1/2} \mathbf{P}^{1/2} \mathbf{H}^\infty \mathbf{P}^{1/2} \mathbf{P}^{1/2} = \mathbf{P}^{-1/2} \mathbf{V}^{-1} \boldsymbol{\Lambda} \mathbf{V} \mathbf{P}^{1/2}, \quad (10)$$

where $\mathbf{P}^{1/2} \mathbf{H}^\infty \mathbf{P}^{1/2} = \mathbf{V}^{-1} \boldsymbol{\Lambda} \mathbf{V}$ is SPD and therefore diagonalizable. One can view $\mathbf{H}^\infty$ as coming from sampling a continuous kernel $K^\infty : \mathbb{S}^{d-1} \times \mathbb{S}^{d-1} \to \mathbb{R}$ given by

$$K^\infty(\mathbf{x}, \mathbf{y}) = K^\infty(\langle \mathbf{x}, \mathbf{y} \rangle) = \frac{(\langle \mathbf{x}, \mathbf{y} \rangle + 1)(\pi - \arccos(\langle \mathbf{x}, \mathbf{y} \rangle))}{4\pi}, \quad (11)$$

where $\langle \cdot, \cdot \rangle$ is the $\ell^2$ inner-product. The eigenvalues and eigenfunctions of $K^\infty$ are known explicitly via the Funk–Hecke formula (Basri et al., 2019):

$$\int_{\mathbb{S}^{d-1}} K^\infty(\mathbf{x}, \mathbf{y}) Y_{\ell,p}(\mathbf{y}) d\mathbf{y} = \mu_\ell Y_{\ell,p}(\mathbf{x}), \qquad \ell \geq 0. \quad (12)$$

The explicit formulas for $\mu_\ell$ with $\ell \geq 0$ are given in the supplementary material. We find that $\mu_\ell > 0$ for all $\ell$ and $\mu_\ell$ is asymptotically $\mathcal{O}(\ell^{-d})$ for large $\ell$ (Basri et al., 2019; Bietti & Mairal, 2019).

## 3 TRAINING CONVERGENCE WITH A GENERAL LOSS FUNCTION

Given the NN model in eq. (5) and a general loss function $\Phi_{\mathbf{P}}$ in eq. (6), we are interested in the convergence rate of NN training. We study this by analyzing the convergence rate for each harmonic component. We start by presenting a convergence result that holds for any SPD matrix $\mathbf{P}$. It says that up to an error $\boldsymbol{\epsilon}$, which can be made arbitrarily small by taking $\kappa$ small enough and $m$ large enough, the residual of the NN at the $k$th iteration is approximately $(\mathbf{I} - 2\eta \mathbf{H}^\infty \mathbf{P})^k \mathbf{y}$.

**Theorem 1.** *In eq. (5), suppose that $\mathbf{w}_1, \ldots, \mathbf{w}_m$ are initialized i.i.d. from Gaussian random variables with covariance matrix $\kappa^2 \mathbf{I}$, $b_1, \ldots, b_m$ are initialized to zero, and $a_1, \ldots, a_m$ are initialized i.i.d. as $+1$ with probability $1/2$ and $-1$ otherwise. Suppose the NN is trained with training data $(\mathbf{x}_i, y_i)$ for $1 \leq i \leq n$, loss function $\Phi_{\mathbf{P}}$ in eq. (6) for a SPD matrix $\mathbf{P}$, and the training procedure is the gradient descent update rule eq. (7) with step size $\eta$. Let $\mathcal{N}_k$ be the NN function after the $k$th iteration and $\mathbf{u}(k) = (\mathcal{N}_k(\mathbf{x}_1), \ldots, \mathcal{N}_k(\mathbf{x}_n))$, where $\mathcal{N}_0$ is the initial NN function. Let an accuracy goal $0 < \epsilon < 1$, a probability of failure $0 < \delta < 1$, and a time span $T > 0$ be given. Then, there exist constants $C_1, C_2 > 0$ that depend only on the dimension $d$ such that if $0 < \eta \leq 1/(2M_{\mathbf{P}}^2 n)$ (see eq. (8)), $\kappa \leq C_1 \epsilon M_{\mathbf{P}}^{-1} \sqrt{\delta/n}$, and $m$ satisfies*

$$m \geq C_2 \left( \frac{M_{\mathbf{P}}^6 n^3}{\kappa^2 \epsilon^2} \left( \lambda_0^{-4} + \eta^4 T^4 \epsilon^4 \right) + \frac{M_{\mathbf{P}}^4 n^2 \log(n/\delta)}{\epsilon^2} \left( \lambda_0^{-2} + \eta^2 T^2 \epsilon^2 \right) \right), \quad (13)$$

*then with probability $\geq 1 - \delta$, we have*

$$\mathbf{y} - \mathbf{u}(k) = (\mathbf{I} - 2\eta \mathbf{H}^\infty \mathbf{P})^k \mathbf{y} + \boldsymbol{\epsilon}(k), \qquad \|\boldsymbol{\epsilon}(k)\|_{\mathbf{P}} \leq \epsilon, \qquad 0 \leq k \leq T. \quad (14)$$

*Here, $\mathbf{H}^\infty$ follows eq. (9), $\mathbf{y} = (g(\mathbf{x}_1), \ldots, g(\mathbf{x}_n))^\top$, and $\lambda_0$ is the smallest eigenvalue of $\mathbf{H}^\infty \mathbf{P}$.*

We defer the proof to the supplementary material, which uses techniques from (Su & Yang, 2019). The main idea behind the proof is that $\mathbf{I} - 2\eta\mathbf{H}^\infty\mathbf{P}$ is close to the transition matrix for the residual $\mathbf{y} - \mathbf{u}(k)$ when $m$ is large. By taking $\kappa$ small, we can control the size of $\mathbf{u}(0)$ and therefore obtain $\mathbf{y} - \mathbf{u}(k) \approx (\mathbf{I} - 2\eta\mathbf{H}^\infty\mathbf{P})^k(\mathbf{y} - \mathbf{u}(0)) \approx (\mathbf{I} - 2\eta\mathbf{H}^\infty\mathbf{P})^k\mathbf{y}$. As $\eta$ decreases, the gradient descent algorithm gets closer to the gradient flow algorithm (Du et al., 2018), which allows us to more accurately quantify the frequency bias (see section 4). For a fixed $n$, if $\mathbf{y}$ is an eigenvector associated to $\lambda_0$, then $\left\|(\mathbf{I} - 2\eta\mathbf{H}^\infty\mathbf{P})^k\mathbf{y}\right\| = \exp(-k\log((1-2\eta\lambda_0)^{-1}))\|\mathbf{y}\|$, where $\log((1-2\eta\lambda_0)^{-1})$ is called the convergence rate (Su & Yang, 2019). If we assume $\lambda_{\min}(\mathbf{P}) = \mathcal{O}(1/n)$, which is the case of eq. (1), then as shown in (Su & Yang, 2019, Thm. 2) and (Nguyen et al., 2021), we expect that $\lambda_0 \to 0$ as $n \to \infty$. Hence, as $n \to \infty$, there exists a labeling $\mathbf{y}$ on the training data, depending on $n$, that makes the convergence rate vanish. However, as suggested by (Su & Yang, 2019; Cao et al., 2019), for a fixed bandlimited target function $g$, its convergence rate in early epochs stays constant as $n \to \infty$. We make similar observations in section 4.

# 4 FREQUENCY BIAS WITH AN $\mathbf{L}^2$-BASED LOSS FUNCTION

The mean-squared loss function in eq. (1) corresponds to setting $c_i = A_d/n$ for $1 \le i \le n$ in eq. (3). When $\mu$ is uniform, eq. (1) and eq. (3) are equivalent; when $\mu$ is nonuniform, we introduce a quadrature rule with nodes $\mathbf{x}_1, \ldots, \mathbf{x}_n$ and weights $c_1, \ldots, c_n$ to approximate the $L^2$ loss function eq. (3). The weights are selected so that for low-frequency functions $f : \mathbb{S}^{d-1} \to \mathbb{R}$, the quadrature error

$$E_{\boldsymbol{c}}(f) = \int_{\mathbb{S}^{d-1}} f(\mathbf{x})d\mathbf{x} - \sum_{i=1}^n c_i f(\mathbf{x}_i) \tag{15}$$

is relatively small.[1] A reasonable quadrature rule has positive weights for numerical stability and satisfies $\sum_{i=1}^n c_i = A_d$ so that it exactly integrates constants. The continuous squared $L^2$ loss function based on the Lebesgue measure is then discretized to be the square of a weighted discrete $\ell^2$ norm (see eq. (3)). Hence, we take $\mathbf{P} = \mathbf{D}_{\boldsymbol{c}} = \text{diag}(c_1, \ldots, c_n)$, which is SPD as the $c_i$'s are positive. For a vector $\mathbf{v} \in \mathbb{R}^n$, we write $\|\mathbf{v}\|_{\mathbf{c}}^2 = \mathbf{v}^\top\mathbf{D}_{\boldsymbol{c}}\mathbf{v}$ and set $c_{\max} = \max_{1 \le i \le n}\{c_i\}$.

We now apply Theorem 1 to study the frequency bias of NN training with the squared $L^2$ loss eq. (3). We state these results in terms of quadrature errors. Recall our continuous setup where we assume that the training data is taken from a function $g : \mathbb{S}^{d-1} \to \mathbb{R}$ so that $y_i = g(\mathbf{x}_i)$ for $1 \le i \le n$. We further assume that $g$ is bandlimited with bandlimit $L$ where $g = g_0 + \cdots + g_L$ and $g_\ell \in \mathcal{H}_\ell^d$ for $0 \le \ell \le L$. With $1 \le i \le n$, $j, \ell \ge 0$, and $1 \le p \le N(d, \ell)$, we define quadrature errors as

$$e_{j,\ell,p}^a = E_{\boldsymbol{c}}(g_j Y_{\ell,p}), \quad e_{i,\ell,p}^b = E_{\boldsymbol{c}}(K^\infty(\mathbf{x}_i, \cdot)Y_{\ell,p}), \quad e_{j,\ell}^c = E_{\boldsymbol{c}}(g_j g_\ell), \quad e_{i,\ell}^d = E_{\boldsymbol{c}}(K^\infty(\mathbf{x}_i, \cdot)g_\ell). \tag{16}$$

We interpret $g_\ell = 0$ when $\ell > L$ and function products are interpreted as pointwise products.

## 4.1 A FREQUENCY-BASED FORMULA FOR THE TRAINING ERROR

We obtain a similar result to (Arora et al., 2019, Thm. 4.1) when using loss function $\tilde{\Phi}$ in eq. (3). Instead of expressing the training error using the spectrum of $\mathbf{H}^\infty\mathbf{D}_{\boldsymbol{c}}$, we directly relate the training error to the frequency components of $g$ and the eigenvalues of the continuous kernel $K^\infty$.

**Theorem 2.** *Under the same setup and assumptions of Theorem 1, let $\mathbf{P} = \mathbf{D}_{\boldsymbol{c}}$ and $M_{\mathbf{P}} = \sqrt{c_{max}}$. If $g : \mathbb{S}^{d-1} \to \mathbb{R}$ is a bandlimited function with bandlimit $L$ and $1 - 2\eta\mu_\ell > 0$ for all $0 \le \ell \le L$ (see eq. (12)), then with probability $\ge 1 - \delta$ we have*

$$\|\mathbf{y} - \mathbf{u}(k)\|_{\boldsymbol{c}} = \sqrt{\sum_{\ell=0}^L (1-2\eta\mu_\ell)^{2k}\|g_\ell\|_{L^2}^2 + \varepsilon_1(k) + \varepsilon_2 + \varepsilon_3(k)}, \quad |\varepsilon_3(k)| \le \epsilon, \quad 0 \le k \le T, \tag{17}$$

*where $\varepsilon_1(k)$ and $\varepsilon_2$ satisfy*

$$|\varepsilon_1(k)| \le \left|\sum_{j=0}^L \sum_{\ell=0}^L (1-2\eta\mu_j)^k(1-2\eta\mu_\ell)^k e_{j,\ell}^c\right|, \quad |\varepsilon_2| \le \sum_{\ell=0}^L \frac{\sqrt{A_d}}{\mu_\ell} \max_{1 \le i \le n}\left|e_{i,\ell}^d\right|.$$

---

[1]In the case where we do not have a good quadrature rule associated with $\{\mathbf{x}_i\}_{i=1}^n$, we usually have very limited understanding of the spectral property of the target function given its values at $\{\mathbf{x}_i\}_{i=1}^n$. Hence, we assume the existence of a good quadrature rule for theoretical purposes.

The proof of Theorem 2 is postponed to the supplementary material. The idea is that by Theorem 1, we know $\mathbf{y} - \mathbf{u}(k) = (\mathbf{I} - 2\eta\mathbf{H}^\infty\mathbf{D_c})^k \mathbf{y} + \boldsymbol{\varepsilon}_3(k)$. Using Funk–Hecke formula and quadrature, we have that $1 - 2\eta\mu_\ell$ are roughly the eigenvalues of $\mathbf{I} - 2\eta\mathbf{H}^\infty\mathbf{D_c}$ and $\mathbf{y}_\ell = (g_\ell(\mathbf{x}_1), \ldots, g_\ell(\mathbf{x}_n))^\top$ are associated eigenvectors. Hence, $\mathbf{y} - \mathbf{u}(k) \approx \sum_{\ell=0}^L (1 - 2\eta\mu_\ell)^k \mathbf{y}_\ell$. This can be made precise by introducing $\varepsilon_2$. Finally, up to some quadrature error $\varepsilon_1$, we have $\langle g_j, g_\ell \rangle_{L^2} \approx A_d n^{-1} \mathbf{y}_j^\top \mathbf{y}_\ell$, which gives us eq. (17). Since $\sum_{i=1}^n c_i = A_d$, For a fixed data distribution $\mu$, we expect that $c_{\max} = \mathcal{O}(n^{-1})$ as $n \to \infty$ so that $\eta$ does not decay as $n \to \infty$. Up to a quadrature error, $\|\mathbf{y} - \mathbf{u}(k)\|_\mathbf{c}$ is close to the $L^2$ norm of the residual function $g - \mathcal{N}_k$. Explicit formulas for the eigenvalues $\{\mu_\ell\}$ (see eq. (12)) are given in (Basri et al., 2019), and it was shown that $\mu_\ell = \mathcal{O}(\ell^{-d})$ (Bietti & Mairal, 2019). Theorem 2 demonstrates the frequency bias in NN training as the rate of convergence for frequency $0 \le \ell \le L$ is $1 - 2\eta\mu_\ell$, which is close to 1 when $\ell$ is large. As $\eta \to 0$, we have $(1 - 2\eta\mu_\ell)^{2t/\eta} \to e^{-4\mu_\ell t}$, which gives the convergence rate for frequency $\ell$ using gradient flow. Therefore, we expect that NN training approximates the low-frequency content of $g$ faster than its high-frequency one, which is similar to the case of training with uniform data (Basri et al., 2019).

## 4.2 Estimating the quadrature errors

We now quantify the quadrature errors in Theorem 2. If we can design a quadrature rule at the training data $\mathbf{x}_1, \ldots, \mathbf{x}_n$ such that the quadrature error satisfies

$$|E_\mathbf{c}(h)| = \left| \int_{\mathbb{S}^{d-1}} h(\mathbf{x})d\mathbf{x} - \sum_{i=1}^n c_i h(\mathbf{x}_i) \right| \le \gamma_{n,\ell} \|h\|_{L^\infty}, \qquad h \in \Pi_\ell^d, \quad \ell \ge 0, \qquad (18)$$

for some constant $\gamma_{n,\ell} \ge 0$, then we can bound the terms in eq. (16). We expect that for each fixed $\ell$, $\gamma_{n,\ell} \to 0$ as $n \to \infty$ as this is saying that integrals can be calculated more accurately for a denser set of quadrature nodes. In practice, it can require a large amount of training data to make $\gamma_{n,\ell}$ small when $d$ is large. Under the assumption that our quadrature rule satisfies eq. (18) with reasonably small $\gamma_{n,\ell}$'s when $\ell$ is small, we can bound the quadrature errors appearing in Theorem 2.

**Theorem 3.** *Under the same assumptions of Theorem 2, and that the quadrature rule satisfies eq. (18), there exist constants $C_1, C_2 > 0$ only depending on the dimension $d$ such that the terms $|\varepsilon_1(k)|$ and $|\varepsilon_2|$ in Theorem 2 satisfy*

$$|\varepsilon_1(k)| \le C_1 \left( \frac{L^3}{\ell} + L^2\gamma_{n,\ell} \right) \max_{0 \le j \le L} \|g_j\|_{L^\infty}, \qquad |\varepsilon_2| \le C_2 \left( \frac{L^2}{\ell} + L\gamma_{n,\ell} \right) \max_{0 \le j \le L} \|g_j\|_{L^\infty} \sum_{j=0}^L \mu_j^{-1}$$

*for all $k \ge 0$, $\ell \ge 1$, where $g = g_0 + \cdots + g_L$ with $g_j \in \mathcal{H}_j^d$ and $\gamma_{n,\ell}$ satisfies eq. (18).*

The proof is in the supplementary material. Theorem 3 states that $\varepsilon_1(k)$ and $\varepsilon_2$ can be made arbitrarily small if the quadrature errors converge to 0 as the number of nodes $n \to \infty$. In particular, if there is a sequence $\{\ell_n\}$ that increases to $\infty$ such that the quadrature rule is exact for all functions $h \in \Pi_{\ell_n}^d$, i.e., $E_\mathbf{c}(h) = 0$, where $\ell_n \to \infty$ (see e.g. (Mhaskar et al., 2000)), the rates of convergence of $\varepsilon_1(k)$ and $\varepsilon_2$ are both $\mathcal{O}(1/\ell_n)$ for a fixed $g$. Without the quadrature being exact, we still have nice convergence provided the quadrature errors are small, as the following corollary shows.

**Corollary 1.** *Suppose there exists a sequence $\ell_n \to \infty$ such that $\gamma_{n,\ell_n} \to 0$ as $n \to \infty$. Then, for a fixed $L \ge 0$, we have $\max_{k \ge 0, g \in \Pi_L^d} |\varepsilon_1(k)| / \|g\|_{L^2}^2$ and $\max_{g \in \Pi_L^d} |\varepsilon_2| / \|g\|_{L^2} \to 0$ as $n \to \infty$.*

Corollary 1 shows that as $n$ increases, the quadrature errors $\varepsilon_1(k)$ and $\varepsilon_2$ converge to zero. Moreover, this convergence is uniform in the sense that it does not depend on the specific choice of $g \in \Pi_L^d$. Here, we normalize $\varepsilon_1(k)$ and $\varepsilon_2$ by $\|g\|_{L^2}^2$ and $\|g\|_{L^2}$, respectively, to obtain the "relative" quadrature errors that do not scale when $g$ is multiplied by a scalar (see (17)).

## 5 Frequency bias with a Sobolev-norm loss function

The frequency bias during the training of an overparameterized NN has several consequences. In many situations, worse convergence rates for high-frequency components of a function are beneficial since the NN training procedure is less sensitive to the oscillatory noise in the data, acting as

a low-pass filter. This significantly improves the generalization error of overparameterized NNs. However, in other situations, NN training struggles to accurately learn the high-frequency content of $g$, resulting in slow convergence. To precisely control the frequency bias of NN training, we propose to train a NN with a loss function that has intrinsic spectral bias.

Let $\mathcal{D}'(\mathbb{S}^{d-1})$ be the space of distributions on $\mathbb{S}^{d-1}$. Given $s \in \mathbb{R}$, consider $\mathcal{L}^s : \mathcal{D}'(\mathbb{S}^{d-1}) \to \mathcal{D}'(\mathbb{S}^{d-1})$, where $\mathcal{L}^s = \left(I + (-\Delta)^{1/2}\right)^s$ and $\Delta$ is the Laplace–Beltrami operator on the sphere. We follow (Barceló et al., 2021) and define the spherical Sobolev space $H^s(\mathbb{S}^{d-1}) = \{f \in \mathcal{D}'(\mathbb{S}^{d-1}) : \mathcal{L}^s f \in L^2(\mathbb{S}^{d-1})\}$, equipped with a norm equivalent to eq. (1.24) in (Barceló et al., 2021),

$$\|f\|^2_{H^s(\mathbb{S}^{d-1})} = \sum_{\ell=0}^{\infty} \sum_{p=1}^{N(d,\ell)} (1+\ell)^{2s} \left|\hat{f}_{\ell,p}\right|^2, \tag{19}$$

where $\hat{f}_{\ell,p}$ are the spherical harmonic coefficients of $f$ (see eq. (4)) and $N(d,\ell)$ is given in section 2. We propose to set the loss function to be $\frac{1}{2}\|g - \mathcal{N}\|^2_{H^s}$ in replace of $\frac{1}{2}\|g - \mathcal{N}\|^2_{L^2}$ in eq. (3). When $s = 0$, it reduces to the $L^2$ norm. If $s > 0$, the high-frequency spherical harmonic coefficients are amplified by $(1+\ell)^{2s}$. The high-frequency components of the residual are then penalized more in the loss function, and one can expect the NN training to learn the high-frequency components faster with the squared $H^s$ loss function than the case of eq. (3). Similarly, if $s < 0$, the high-frequency spherical harmonic coefficients are dampened by $(1+\ell)^{2s}$ and one expects the NN training captures the high-frequency components of the residual more slowly with the squared $H^s$ loss function. However, when $s < 0$, the training is more robust to the high-frequency noise in the data. By tuning the parameter $s$, we can control the frequency bias in NN training (see Theorem 4). The choice of $s$ for a particular application can be determined from theory or by cross-validation.

First, we justify that the residual function is indeed in $H^s$. Since we assume that $g$ is bandlimited, $g \in H^s$ for all $s \in \mathbb{R}$. Proposition 2 shows that we could consider $s < 3/2$ for ReLU-based NNs.

**Proposition 2.** *Suppose $\mathcal{N} : \mathbb{S}^{d-1} \to \mathbb{R}$ is a 2-layer ReLU NN (see eq. (5)). Then, we have $\mathcal{N} \in H^s(\mathbb{S}^{d-1})$ for all $s < 3/2$. Moreover, if $s \geq 3/2$, $\mathcal{N} \in H^s(\mathbb{S}^{d-1})$ if and only if $\mathcal{N}$ is affine.*

The proof is deferred to the supplementary material. When $s \geq 3/2$, the residual function $\mathcal{N} - g$ may not be in $H^s$. However, we can still truncate the sum in eq. (19) to a maximum frequency $\ell_{\max}$ to train the NN, although the sum can no longer be interpreted as an approximation of some Sobolev norm at the continuous level. We discretize the Sobolev-based loss function as

$$\Phi_s(\mathbf{W}) = \frac{1}{2} \sum_{\ell=0}^{\ell_{\max}} \sum_{p=1}^{N(d,\ell)} (1+\ell)^{2s} \left(\sum_{i=1}^{n} c_i Y_{\ell,p}(\mathbf{x}_i)(g - \mathcal{N})(\mathbf{x}_i)\right)^2 = \frac{1}{2}(\mathbf{y} - \mathbf{u})^\top \mathbf{P}_s (\mathbf{y} - \mathbf{u}), \tag{20}$$

where $\mathbf{u}$ and $\mathbf{y}$ follow eq. (6), and $\mathbf{P}_s = \sum_{\ell=0}^{\ell_{\max}} \sum_{p=1}^{N(d,\ell)} (1+\ell)^{2s} \mathbf{P}_{\ell,p}$, $\mathbf{P}_{\ell,p} = \mathbf{a}_{\ell,p} \mathbf{a}_{\ell,p}^\top$, and $(\mathbf{a}_{\ell,p})_i = c_i Y_{\ell,p}(\mathbf{x}_i)$. We assume that $\mathbf{P}_s$ is SPD, which requires that $(\ell_{\max} + 1)^2 \geq n$. Next, we present our convergence theorem for Sobolev training.

**Theorem 4.** *Suppose $g \in \Pi_L^d$ and $\Phi_s$ is the loss function in eq. (20), where $\mathbf{P}_s$ is SPD and $\ell_{max} \geq L$. Under the assumptions of Theorem 1, if $1 - 2\eta\mu_\ell(1 + \ell)^{2s} > 0$ for all $0 \leq \ell \leq L$, then with probability $\geq 1 - \delta$ over the random initialization, we have*

$$\mathbf{y} - \mathbf{u}(k) = \sum_{\ell=0}^{L} \left(1 - 2\eta\mu_\ell(1+\ell)^{2s}\right)^k \mathbf{y}^\ell + \boldsymbol{\varepsilon}_1 + \boldsymbol{\varepsilon}_2(k), \qquad \|\boldsymbol{\varepsilon}_2(k)\|_{\mathbf{P}_s} \leq \epsilon, \quad 0 \leq k \leq T, \tag{21}$$

*where $\mathbf{y}^\ell = (g_\ell(\mathbf{x}_1), \ldots, g_\ell(\mathbf{x}_n))^\top$ and $\boldsymbol{\varepsilon}_1$ satisfies*

$$\|\boldsymbol{\varepsilon}_1\|_{\mathbf{P}_s} \leq \sum_{\ell=0}^{L} \mu_\ell^{-1} \|\boldsymbol{\varepsilon}_1^\ell\|_{\mathbf{P}_s}, \quad (\varepsilon_1^\ell)_i = e_{i,\ell}^d + \sum_{j=0}^{\ell_{max}} \frac{(1+j)^{2s}}{(1+\ell)^{2s}} \sum_{p=1}^{N(d,j)} e_{\ell,j,p}^a \left(\mu_j Y_{j,p}(\mathbf{x}_i) + e_{i,j,p}^b\right).$$

Compared to Theorem 2, Theorem 4 says that up to the level of quadrature errors, the convergence rate of the degree-$\ell$ component is $1 - 2\eta\mu_\ell(1 + \ell)^{2s}$. In particular, since $\mu_\ell = \mathcal{O}(\ell^{-d})$, there is an $s^* > 0$, which depends on $d$, such that $(1 + \ell)^{2s^*}\mu_\ell$ can be bounded from above and below for all $\ell \geq 0$ by positive constants that are independent of $\ell$. This means for any $s > s^*$, we expect to reverse the frequency bias behavior of NN training. Figure 2 shows the reversal of frequency bias as $s$ increases from $-1$ to $4$ (see section 6.1).

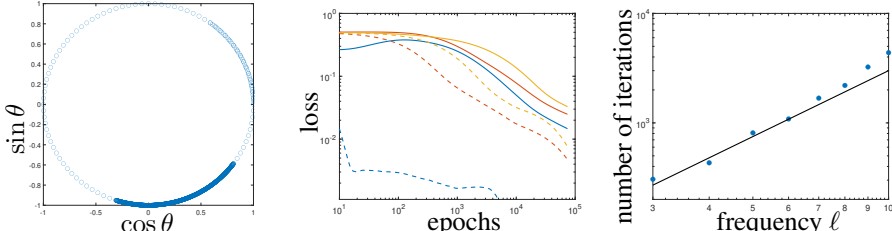

Figure 1: Left: the nonuniform data $\mathbf{x} = (\cos\theta, \sin\theta)$ on the unit circle $\mathbb{S}^1$. Middle: the change of frequency loss for $\ell = 1$ (blue), 5 (red) and 9 (yellow) against the number of iterations for loss function $\Phi$ (solid lines) and $\widetilde{\Phi}$ (dashed lines). Right: the number of iterations for the NN training to achieve a fixed loss threshold in learning $g_\ell(\mathbf{x}) = \sin(\ell\theta)$ for $3 \le \ell \le 10$ given the loss function $\widetilde{\Phi}$. The black line represents the $\mathcal{O}(\ell^2)$ rate based on the analysis in (Basri et al., 2019).

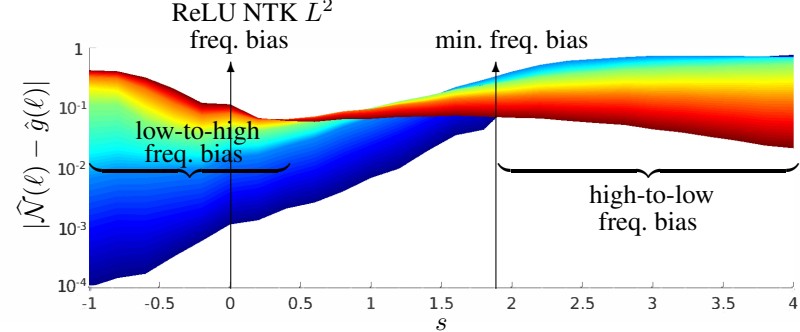

Figure 2: Frequency bias during NN training with a squared $H^s$ loss function. The blue-to-red rainbow corresponds to low-to-high frequency losses $|\widehat{\mathcal{N}}(\ell) - \hat{g}(\ell)|$ for the frequency index $\ell$ ranging from 1 (blue) to 9 (red) with nonuniform training data, respectively. Here, an overparameterized 2-layer ReLU NN $\mathcal{N}(\mathbf{x})$ is trained for 5000 epochs to learn function $g(\mathbf{x})$ given the $H^s$ loss with $-1 \le s \le 4$. The inversion of the rainbow demonstrates the reversal of frequency bias.

## 6 EXPERIMENTS AND DISCUSSION

This section presents three experiments with synthetic and real-world datasets to investigate the frequency bias of NN training using squared $L^2$ loss and squared $H^s$ loss. The first two experiments learn functions on $\mathbb{S}^1$ and $\mathbb{S}^2$, respectively. In the third test, we train an autoencoder on the MNIST dataset for a denoising task. One can find more details in the supplementary material.

### 6.1 LEARNING TRIGONOMETRIC POLYNOMIALS ON THE UNIT CIRCLE

First, we consider learning a function on $\mathbb{S}^1$. We create a set of $n = 1140$ nonuniform data $\{\mathbf{x}_i\}_{i=1}^n$, as seen in Figure 1, and compute the quadrature weights $\{c_i\}_{i=1}^n$ for the loss function $\widetilde{\Phi}$ in eq. (3) by solving a constrained quadratic program (see appendix F.1). We train a 2-layer ReLU NN to learn $g(\mathbf{x}) = \tilde{g}(\theta) = \sum_{\ell=1}^9 \sin(\ell\theta)$, where $\mathbf{x} = (\cos\theta, \sin\theta)$. We define the frequency loss $|\widehat{\mathcal{N}}(\ell) - \hat{g}(\ell)|$ where $\mathcal{N}$ and $\hat{g}$ are the Fourier coefficients of $\mathcal{N}$ and $g$, respectively (see appendix F.1). In Figure 1, we plot the frequency loss for $\ell = 1, 5, 9$ in different colors to illustrate how well the NN fits each frequency component. The solid and dashed lines correspond to the loss function $\Phi$ in eq. (1) and $\widetilde{\Phi}$ in eq. (3), respectively. Our observations collaborate with the theoretical statements in Theorem 2. Figure 1 also shows that it takes asymptotically $\mathcal{O}(\ell^2)$ iterations to learn the $\ell$th frequency $\sin(\ell\theta)$ given the loss function $\widetilde{\Phi}$. A similar plot appears in Basri et al. (2019) for uniform training data.

We also use the squared $H^s$ norm as the loss function to learn $g$. After 5000 epochs, we plot the $\ell$th frequency loss with $\ell$ ranging from 1 (blue) to 9 (red) in Figure 2, given different $s$ values. As $s$ increases, the higher-frequency components are learned faster. When $s > 2$, the frequency bias is reversed in the sense that higher-frequency parts are learned faster than the lower-frequency ones rather than a low-frequency bias under the squared $L^2$ loss (see Theorem 2). The gradually changing "rainbow" in Figure 2 shows that the smoothing property of an overparameterized NN can be compensated by the $H^s$ loss function for large enough $s$, corroborating Theorem 4.

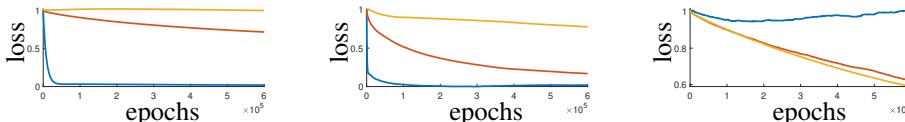

Figure 3: Frequency losses for $\ell = 4$ (blue), 10 (red), and 20 (yellow), when learning a function on $\mathbb{S}^2$ using different squared $H^s$ loss functions. Compared to low-frequency bias in the cases of $s = -1$ (left) and $s = 0$ (middle), we observe a high-frequency bias when $s = 2.5$ (right).

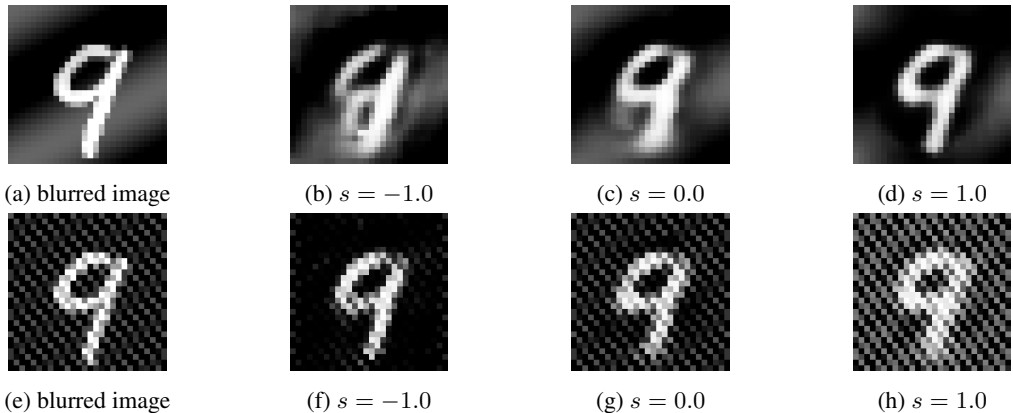

| (a) blurred image | (b) $s = -1.0$ | (c) $s = 0.0$ | (d) $s = 1.0$ |

| (e) blurred image | (f) $s = -1.0$ | (g) $s = 0.0$ | (h) $s = 1.0$ |

Figure 4: Outputs of a squared $H^s$ loss-trained autoencoder on a typical test image when the input images are contaminated by low-frequency noise (top row) and high-frequency noise (bottom row).

## 6.2 LEARNING SPHERICAL HARMONICS ON THE UNIT SPHERE

Similar to the previous example in $\mathbb{S}^1$, we design an experiment on $\mathbb{S}^2$. We utilize a data set $\{\mathbf{x}_i\}_{i=1}^{2500}$ in (Wright & Michaels, 2015), which comes with carefully designed positive quadrature weights $\{c_i\}_{i=1}^{2500}$. We test the squared $H^s$ loss function in NN training with a target function $g(\mathbf{x}) = \sum_{\ell=0,\ell \text{ even}}^{30} Y_{\ell,0}$ defined on $\mathbb{S}^2$ that involves many high-frequency components. The results are shown in Figure 3 with different $s$ values. The natural low-frequency bias of the NN in the case of $L^2$-based training (i.e., $s = 0$) is enhanced when $s = -1$, and is totally reversed when $s = 2.5$.

## 6.3 AUTOENCODER ON THE MNIST DATASET

The idea of Sobolev training is also useful for high-dimensional training data. In Figure 4, we present the results of the autoencoder for image denoising using the MNIST dataset (LeCun et al., 2010). The outputs of the autoencoder are presented when trained with the squared $H^s$ norm as the loss function. We contaminate the dataset with random low-frequency noise (top row) and high-frequency noise (bottom row). When high-frequency noise is present, the $H^s$ loss function generally performs better with $s < 0$, while the case of $s > 0$ helps image deblurring when the input image suffers from low-frequency noise. This corroborates our discussion in Section 5. In appendix G, we theoretically justify this phenomenon by studying the frequency bias in operator learning.

## 7 CONCLUSIONS

A frequency bias phenomenon is observed in NN training with nonuniform training data. Instead of the standard mean-squared loss function in eq. (1), we propose the use of a different loss function in eq. (3), which involves quadrature weights and has a natural continuous analog. With eq. (3), we rigorously analyze frequency bias with nonuniform training data using the Funk–Hecke formula. By changing the loss function to a squared $L^2$-type Sobolev norm, we can control the frequency bias in NN training, which can accelerate NN training convergence and improve robustness under noises.

ACKNOWLEDGMENTS

We thank Aparna Gupte and Liu Zhang for discussions and some initial experiments. The work was supported by the National Science Foundation grants DMS-1913129, DMS-1952757, and DMS-2045646, as well as the Simon Foundation. Y. Yang acknowledges support from Dr. Max Rössler, the Walter Haefner Foundation, and the ETH Zürich Foundation. We also gratefully thank the anonymous referees for their constructive comments, suggestions, and discussions.

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

SUPPLEMENTARY MATERIAL

This is the supplementary material for the paper titled "Tuning Frequency Bias in Neural Network Training with Nonuniform Data."

The supplementary material is organized as follows. In Appendix A, we recall the notations used in the paper and introduce additional concepts for our analysis. In Appendix B, we prove that $\mathbf{H}^\infty$ is a symmetric and positive definite matrix. In Appendix C, we prove Theorem 1 of the paper, while the proofs of results in Section 4 and Section 5 are given in Appendix D and Appendix E, respectively. In Appendix F, we provide further details of our three experiments. The observation made in the third experiment is then briefly justified in Appendix G, in which we study frequency bias in a general operator learning setting. In Appendix H, we formally discuss the computation of positive quadrature weights.

## A  PRELIMINARIES AND NOTATION

For $d > 1$, let $g : \mathbb{S}^{d-1} \to \mathbb{R}$ be a square-integrable function defined on $\mathbb{S}^{d-1}$. The function $g$ has a spherical harmonic expansion given in Section 2. We denote the space of harmonic functions of degree $\ell$ by $\mathcal{H}_\ell^d$, which is the span of $\{Y_{\ell,p}\}_{p=1}^{N(d,\ell)}$. We further denote the space of spherical harmonics of degree $\leq \ell$ by $\Pi_\ell^d = \bigoplus_{j=0}^\ell \mathcal{H}_j^d$.

Given distinct training data $\{\mathbf{x}_i\}_{i=1}^n$ from $\mathbb{S}^{d-1}$ and evaluations $y_i = g(\mathbf{x}_i)$ for $1 \leq i \leq n$, our goal is to understand the intrinsic frequency bias behavior of training a 2-layer ReLU NN given in eq. (5). It is important for the theory that we initialize the weights as independently and identically distributed (iid) Gaussian random variables with a covariance matrix $\kappa^2\mathbf{I}$, the bias terms are initialized to zero, and the coefficients, i.e., $a_1, \ldots, a_m$, are initialized iid as $+1$ with probability $1/2$ and $-1$ otherwise. During the training process, the values of $\{a_r\}$ are not updated.

We train with the loss function given in eq. (6) so that the gradient descent algorithm for NN training is given by eq. (7). An important object in understanding the frequency bias of NN training is the symmetric and positive definite matrix $\mathbf{H}^\infty \in \mathbb{R}^{n \times n}$ in eq. (9). Since $\mathbf{H}^\infty$ and $\mathbf{P}$ are symmetric positive definite matrices (see Proposition 1), $\mathbf{H}^\infty\mathbf{P}$ has positive real eigenvalues. To see this, note that $\mathbf{H}^\infty\mathbf{P} = \mathbf{H}^\infty\mathbf{P}^{1/2}\mathbf{P}^{1/2} = \mathbf{P}^{-1/2}(\mathbf{P}^{1/2}\mathbf{H}^\infty\mathbf{P}^{1/2})\mathbf{P}^{1/2}$. This means that $\mathbf{H}^\infty\mathbf{P}$ and $\mathbf{P}^{1/2}\mathbf{H}^\infty\mathbf{P}^{1/2}$ are similar. Since the matrix $\mathbf{P}^{1/2}\mathbf{H}^\infty\mathbf{P}^{1/2}$ is symmetric positive definite, $\mathbf{H}^\infty\mathbf{P}$ has positive eigenvalues. We denote the eigenvalues of $\mathbf{H}^\infty\mathbf{P}$ by $\lambda_{n-1} \geq \cdots \geq \lambda_0 > 0$, which partially govern the frequency bias phenomena. It is worth mentioning that although $\mathbf{H}^\infty\mathbf{P}$ is not symmetric, it is self-adjoint in the inner product space induced by $\mathbf{P}$ because

$$\langle \mathbf{H}^\infty\mathbf{P}\boldsymbol{\xi}, \boldsymbol{\zeta}\rangle_\mathbf{P} = (\mathbf{H}^\infty\mathbf{P}\boldsymbol{\xi})^\top\mathbf{P}\boldsymbol{\zeta} = \boldsymbol{\xi}^\top\mathbf{P}(\mathbf{H}^\infty\mathbf{P}\boldsymbol{\zeta}) = \langle \boldsymbol{\xi}, \mathbf{H}^\infty\mathbf{P}\boldsymbol{\zeta}\rangle_\mathbf{P}. \tag{22}$$

It is convenient to analyze the eigenvalues and eigenvectors of $\mathbf{H}^\infty\mathbf{P}$ via the zonal kernel $K^\infty : \mathbb{S}^{d-1} \times \mathbb{S}^{d-1} \to \mathbb{R}$ given in eq. (11). The key is the Funk–Hecke formula (Seeley, 1966).

**Theorem 5** (Funk–Hecke). Suppose $K : [-1, 1] \to \mathbb{R}$ is measurable and $K(t)(1 - t^2)^{(d-3)/2}$ is integrable on $[-1, 1]$. Then, for any $h \in \mathcal{H}_\ell^d$, we have

$$\int_{\mathbb{S}^{d-1}} K(\langle\boldsymbol{\xi}, \boldsymbol{\zeta}\rangle)h(\boldsymbol{\xi})d\boldsymbol{\xi} = \left(A_d \int_{-1}^1 K(t)P_{\ell,d}(t)(1-t^2)^{(d-3)/2}dt\right)h(\boldsymbol{\zeta}), \qquad \boldsymbol{\zeta} \in \mathbb{S}^{d-1}, \tag{23}$$

where $P_{\ell,d}$ is the ultraspherical polynomial given by

$$P_{\ell,d}(t) = \frac{(-1)^\ell\Gamma((d-1)/2)}{2^\ell\Gamma(\ell+(d-1)/2)(1-t^2)^{(d-3)/2}}\frac{d^\ell}{dt^\ell}(1-t^2)^{\ell+(d-3)/2}. \tag{24}$$

Applying Theorem 5, we have that

$$\int_{\mathbb{S}^{d-1}} K^\infty(\mathbf{x}, \mathbf{y})h(\mathbf{y})d\sigma(\mathbf{y}) = \mu_\ell h(\mathbf{x}), \qquad h \in \mathcal{H}_\ell^d,$$

where $\mu_\ell > 0$, $\forall \ell$, given by (Basri et al., 2019)

$$
\mu_\ell = \begin{cases}
\frac{1}{2}C_1^d(0)\left(\frac{1}{(d-1)2^d}\binom{d-1}{\frac{d-1}{2}} + \frac{2^{d-2}}{(d-1)\binom{d-2}{\frac{d-1}{2}}} - \frac{1}{2}\sum_{p=0}^{\frac{d-3}{2}}(-1)^p\binom{\frac{d-3}{2}}{p}\frac{1}{2p+1}\right), & \ell = 0, \\[2ex]
\frac{1}{2}C_1^d(1)\sum_{p=\lceil\frac{\ell}{2}\rceil}^{\ell+\frac{d-3}{2}} C_2^d(p,1)\left(\frac{1}{2(2p+1)} + \frac{1}{4p}\left(1 - \frac{1}{2^{2p}}\binom{2p}{p}\right)\right), & \ell = 1, \\[2ex]
\frac{1}{2}C_1^d(\ell)\sum_{p=\lceil\frac{\ell}{2}\rceil}^{\ell+\frac{d-3}{2}} C_2^d(p,\ell)\left(\frac{-1}{2(2p-\ell+1)} + \frac{1}{2(2p-\ell+2)}\left(1 - \frac{1}{2^{2p-\ell+2}}\binom{2p-\ell+2}{\frac{2p-\ell+2}{2}}\right)\right), & \ell \geq 2 \text{ even}, \\[2ex]
\frac{1}{2}C_1^d(\ell)\sum_{p=\lceil\frac{\ell}{2}\rceil}^{\ell+\frac{d-3}{2}} C_2^d(p,\ell)\left(\frac{1}{2(2p-\ell+1)}\left(1 - \frac{1}{2^{2p-\ell+1}}\binom{2p-\ell+1}{\frac{2p-\ell+1}{2}}\right)\right), & \ell \geq 2 \text{ odd}
\end{cases}
$$

for $d \geq 3$, $d$ odd, where

$$
C_1^d(\ell) = \frac{(-1)^\ell 2\pi^{(d-1)/2}}{(d-1)2^\ell\Gamma(\ell + (d-1)/2)}, \qquad C_2^d(p,\ell) = (-1)^p\binom{\ell + \frac{d-3}{2}}{p}\frac{(2p)!}{(2p-\ell)!}.
$$

Here, the exclamation mark means a factorial and $\binom{\cdot}{\cdot}$ denotes the binomial coefficient.

Given $\mathbf{x}$, $\mathbf{w}_r$, and $b_r$ in eq. (5), we write $\tilde{\mathbf{x}} = \frac{1}{\sqrt{2}}(\mathbf{x}, 1) \in \mathbb{S}^d$ and $\tilde{\mathbf{w}}_r = (\mathbf{w}_r, b_r) \in \mathbb{R}^{d+1}$. Therefore, we have $\text{ReLU}(\mathbf{w}_r^\top \mathbf{x} + b_r) = \sqrt{2}\text{ReLU}(\tilde{\mathbf{w}}_r^\top \tilde{\mathbf{x}})$ and the NN function can be rewritten as

$$
\mathcal{N}(\mathbf{x}) = \frac{\sqrt{2}}{\sqrt{m}}\sum_{r=1}^{m} a_r \text{ReLU}(\tilde{\mathbf{w}}_r^\top \tilde{\mathbf{x}}).
$$

By replacing the expectation over random initialization of $\tilde{\mathbf{w}}$ by $\tilde{\mathbf{w}}(t)$, we define the instantiations of $\mathbf{H}^\infty$ at the $k$th iteration by $\mathbf{H}(k)$, where

$$
H_{ij}(k) = \frac{1}{m}\tilde{\mathbf{x}}_i^\top \tilde{\mathbf{x}}_j \sum_{r=1}^{m} \mathbb{1}_{\{\tilde{\mathbf{x}}_i^\top \tilde{\mathbf{w}}_r(k) \geq 0, \tilde{\mathbf{x}}_j^\top \tilde{\mathbf{w}}_r(k) \geq 0\}}, \tag{25}
$$

where $\mathbb{1}$ is an indicator function.

## B  THE MATRIX $\mathbf{H}^\infty$ IS SYMMETRIC AND POSITIVE DEFINITE

Proposition 1 states that the matrix $\mathbf{H}^\infty$ defined by eq. (9) is symmetric and positive definite. While the symmetry of $\mathbf{H}^\infty$ is immediate from its closed-form expression, the fact that it is positive definite requires a more detailed analysis. The proof idea is similar to that of Theorem 3.1 of (Du et al., 2018), in which the matrix $\mathbf{H}^\infty$ is associated with a 2-layer ReLU NN without biases. However, our $\mathbf{H}^\infty$ is associated with a 2-layer ReLU NN with biases. While (Du et al., 2018) requires that no two training data points are parallel, we allow the existence of $\mathbf{x}_{i_1} = -\mathbf{x}_{i_2}$ for some $i_1$ and $i_2$. To deal with this case, our proof employs a pair of nodes denoted by $\mathbf{x}_{i_1}, \mathbf{x}_{i_2}$.

*Proof of Proposition 1.* For a measurable function $f : \mathbb{R}^d \to \mathbb{R}^{d+1}$, we define a norm of $f$ as

$$
\|f\|_{\mathcal{H}}^2 = \mathbb{E}_{\mathbf{w}\sim\mathcal{N}(\mathbf{0},\kappa^2\mathbf{I})}\|f(\mathbf{w})\|_2^2,
$$

and let $\mathcal{H}$ be the space of measurable functions such that $\|f\|_{\mathcal{H}} < \infty$. It can be shown that $\mathcal{H}$ is a Hilbert space with respect to the inner product $\langle f, g\rangle_{\mathcal{H}} = \mathbb{E}_{\mathbf{w}\sim\mathcal{N}(\mathbf{0},\kappa^2\mathbf{I})}\left[f(\mathbf{w})^\top g(\mathbf{w})\right]$ (Du et al., 2018). For each $\mathbf{x}_i$, $1 \leq i \leq n$, we define the function $\phi_i$ by

$$
\phi_i(\mathbf{w}) = \tilde{\mathbf{x}}_i \mathbb{1}_{\{\mathbf{w}^\top \mathbf{x}_i \geq 0\}}, \qquad \mathbf{w} \in \mathbb{R}^d.
$$

Then, $\phi_i \in \mathcal{H}$ for all $i$, and $H_{ij}^\infty = \langle \phi_i, \phi_j \rangle_{\mathcal{H}}$. We prove that $\mathbf{H}^\infty$ is positive definite by showing that $\{\phi_i\}_{i=1}^n$ is a linearly independent set in $\mathcal{H}$.

To show $\{\phi_i\}_{i=1}^n$ is a linearly independent set, we show that

$$
\alpha_1\phi_1(\mathbf{w}) + \cdots + \alpha_n\phi_n(\mathbf{w}) = \mathbf{0} \qquad \text{for almost every } \mathbf{w} \in \mathbb{R}^d \tag{26}
$$

implies that $\alpha_i = 0$ for $1 \leq i \leq n$. We fix some $1 \leq i_1 \leq n$ and, without loss of generality, assume that $\mathbf{x}_{i_2} = -\mathbf{x}_{i_1}$.[2] Define the set $D_j = \{\mathbf{w} \in \mathbb{R}^d \mid \mathbf{w}^\top \mathbf{x}_j = 0\}$ for $1 \leq j \leq n$. As a

---

[2]Otherwise, if such $i_2$ does not exist, we can add an element $\phi_{n+1}$ associated with $-\mathbf{x}_{i_1}$ to $\{\phi_i\}_{i=1}^n$. If we can show $\{\phi_i\}_{i=1}^{n+1}$ is linearly independent, then so is $\{\phi_i\}_{i=1}^n$.

result, $D_{i_1} = D_{i_2}$. Since each $D_j$ is a hyperplane passing through the origin and $D_{i_1} \neq D_j$ for any $j \neq i_1, i_2$, $\exists \mathbf{z} \in D_{i_1}$ such that $\mathbf{z} \notin D_j$ for any $j \neq i_1, i_2$. For a positive radius $R > 0$, let $B_R = B(\mathbf{z}, R)$ be the ball centered at $\mathbf{z}$ of radius $R$. Define a partition of $B_R$ into two sets denoted by $B_R^+$ and $B_R^-$ (possibly missing a subset of $B_R$ that has zero Lebesgue measure), where

$$B_R^+ = \{\mathbf{w} \in B_R \mid \mathbf{w}^\top \mathbf{x}_{i_1} > 0\} = \{\mathbf{w} \in B_R \mid \mathbf{w}^\top \mathbf{x}_{i_2} < 0\},$$
$$B_R^- = \{\mathbf{w} \in B_R \mid \mathbf{w}^\top \mathbf{x}_{i_1} < 0\} = \{\mathbf{w} \in B_R \mid \mathbf{w}^\top \mathbf{x}_{i_2} > 0\}.$$

Since $D_j$ is closed for each $1 \leq j \leq n$, $B_R$ is eventually disjoint from $D_j$ as $R \to 0$. Hence, we have

$$\lim_{R \to 0} \sup_{\mathbf{w} \in B_R} |\phi_j(\mathbf{z}) - \phi_j(\mathbf{w})| = 0, \qquad j \neq i_1, i_2,$$

where $|\cdot|$ denotes the Euclidean distance. Then, for any $j \neq i_1, i_2$, we have

$$\lim_{R \to 0} \frac{1}{|B_R^+|} \int_{B_R^+} \phi_j(\mathbf{w}) d\mathbf{w} = \phi_j(\mathbf{z}), \quad \lim_{R \to 0} \frac{1}{|B_R^-|} \int_{B_R^-} \phi_j(\mathbf{w}) d\mathbf{w} = \phi_j(\mathbf{z}),$$

where $|\cdot|$ denotes the Lebesgue measure of a set. Consequently, we find that

$$\lim_{R \to 0} \left( \frac{1}{|B_R^+|} \int_{B_R^+} \phi_j(\mathbf{w}) d\mathbf{w} - \frac{1}{|B_R^-|} \int_{B_R^-} \phi_j(\mathbf{w}) d\mathbf{w} \right) = \mathbf{0}, \qquad j \neq i_1, i_2.$$

Now, consider the integral of $\phi_{i_1}$ and $\phi_{i_2}$. We have

$$\lim_{R \to 0} \left( \frac{1}{|B_R^+|} \int_{B_R^+} \phi_{i_1}(\mathbf{w}) d\mathbf{w} - \frac{1}{|B_R^-|} \int_{B_R^-} \phi_{i_1}(\mathbf{w}) d\mathbf{w} \right) = \lim_{R \to 0} \left( \frac{1}{|B_R^+|} \int_{B_R^+} \tilde{\mathbf{x}}_{i_1} d\mathbf{w} - \frac{1}{|B_R^-|} \int_{B_R^-} \mathbf{0} d\mathbf{w} \right) = \tilde{\mathbf{x}}_{i_1},$$

$$\lim_{R \to 0} \left( \frac{1}{|B_R^+|} \int_{B_R^+} \phi_{i_2}(\mathbf{w}) d\mathbf{w} - \frac{1}{|B_R^-|} \int_{B_R^-} \phi_{i_2}(\mathbf{w}) d\mathbf{w} \right) = \lim_{R \to 0} \left( \frac{1}{|B_R^+|} \int_{B_R^+} \mathbf{0} d\mathbf{w} - \frac{1}{|B_R^-|} \int_{B_R^-} \tilde{\mathbf{x}}_{i_2} d\mathbf{w} \right) = -\tilde{\mathbf{x}}_{i_2}.$$

By applying these limiting expressions to $\sum_{j=1}^n \alpha_j \phi_j(\mathbf{w}) = \mathbf{0}$, we find that $\alpha_{i_1} \tilde{\mathbf{x}}_{i_1} - \alpha_{i_2} \tilde{\mathbf{x}}_{i_2} = \mathbf{0}$. Since the last entries of both $\tilde{\mathbf{x}}_{i_1}$ and $\tilde{\mathbf{x}}_{i_2}$ are $1/\sqrt{2}$, we have $\alpha_{i_1} = \alpha_{i_2}$. Thus, we have $\alpha_{i_1} = 0$ because $\mathbf{x}_{i_1} \neq \mathbf{0}$. Since $i_1$ is arbitrary, we showed that $\alpha_j = 0$ for every $1 \leq j \leq n$, and the statement of the proposition follows. $\qquad\square$

It is clear that the proof of Proposition 1 is also true if we assume that each entry of $\mathbf{w}_r$ is initialized from an iid sub-Gaussian distribution with zero mean and whose support is the entire $\mathbb{R}$, and we update the definition of $\mathbf{H}^\infty$ according to eq. (9).

## C  THE CONVERGENCE OF NEURAL NETWORK TRAINING

In this section, we develop the theory for learning a NN with a general loss function $\Phi_{\mathbf{P}}$ defined by a positive definite matrix $\mathbf{P}$ in eq. (6). In particular, we prove Theorem 1, which states that provided the learning rate is sufficiently small, the weights are initialized without too much variance, and the NN is sufficiently wide, then the residual in the first few epochs can be described with the matrix $\mathbf{H}^\infty \mathbf{P}$.

While our proof is similar to that of (Su & Yang, 2019), the argument is distinct in three essential ways: (1) Our proof applies to any loss function defined by a positive definite matrix $\mathbf{P}$, which requires us to use a different Hilbert space $(\mathbb{R}^n, \langle \cdot, \cdot \rangle_{\mathbf{P}})$. (2) While the result in (Su & Yang, 2019) bounds the residual using the minimum eigenvalue of $\mathbf{H}^\infty$, we estimate the residual as a matrix-vector product of $(\mathbf{I} - 2\eta \mathbf{H}^\infty \mathbf{P})^k \mathbf{y}$, which allows us to analyze the training error using all eigenvalues of $\mathbf{H}^\infty \mathbf{P}$. (3) We use a different NN function that incorporates the bias terms and we do not assume that we initialize the weights in a way that makes $\mathcal{N}_0 = 0$.

Before we prove the theorem, we define some useful quantities. Let $\mathcal{A}$ be the set of indices such that the coefficients $a_r$ are initialized to $1$ and let $\mathcal{B}$ be the set initialized to $-1$. We then decompose $\mathbf{H}(k)$ into two parts, where $\mathbf{H}(k)$ is defined in eq. (25), so that $\mathbf{H}(k) = \mathbf{H}^+(k) + \mathbf{H}^-(k)$ with

$$H_{ij}^+(k) = \frac{1}{m} \tilde{\mathbf{x}}_i^\top \tilde{\mathbf{x}}_j \sum_{r \in \mathcal{A}} \mathbb{1}_{\left\{ \substack{\tilde{\mathbf{w}}_r(k)^\top \tilde{\mathbf{x}}_i \geq 0 \\ \tilde{\mathbf{w}}_r(k)^\top \tilde{\mathbf{x}}_j \geq 0} \right\}}, \qquad H_{ij}^-(k) = \frac{1}{m} \tilde{\mathbf{x}}_i^\top \tilde{\mathbf{x}}_j \sum_{r \in \mathcal{B}} \mathbb{1}_{\left\{ \substack{\tilde{\mathbf{w}}_r(k)^\top \tilde{\mathbf{x}}_i \geq 0 \\ \tilde{\mathbf{w}}_r(k)^\top \tilde{\mathbf{x}}_j \geq 0} \right\}}.$$

Similarly, we define two other matrices $\tilde{\mathbf{H}}^+(k)$ and $\tilde{\mathbf{H}}^-(k)$ as

$$\tilde{H}_{ij}^+(k) = \frac{1}{m}\tilde{\mathbf{x}}_i^\top \tilde{\mathbf{x}}_j \sum_{r \in \mathcal{A}} \mathbb{1}_{\left\{ \begin{array}{c} \tilde{\mathbf{w}}_r(k+1)^\top \tilde{\mathbf{x}}_i \geq 0 \\ \tilde{\mathbf{w}}_r(k)^\top \tilde{\mathbf{x}}_j \geq 0 \end{array} \right\}}, \quad \tilde{H}_{ij}^-(k) = \frac{1}{m}\tilde{\mathbf{x}}_i^\top \tilde{\mathbf{x}}_j \sum_{r \in \mathcal{B}} \mathbb{1}_{\left\{ \begin{array}{c} \tilde{\mathbf{w}}_r(k+1)^\top \tilde{\mathbf{x}}_i \geq 0 \\ \tilde{\mathbf{w}}_r(k)^\top \tilde{\mathbf{x}}_j \geq 0 \end{array} \right\}}.$$

Unfortunately, $\tilde{\mathbf{H}}^+(k)$ and $\tilde{\mathbf{H}}^-(k)$ are not necessarily symmetric and they differ from $\mathbf{H}^+(k)$ and $\mathbf{H}^-(k)$ up to sign flips. To simplify the notation later, we also define two auxiliary matrices $\mathbf{L}(k)$ and $\mathbf{M}(k)$ as

$$\mathbf{L}(k) = \tilde{\mathbf{H}}^+(k) - \mathbf{H}^+(k), \qquad \mathbf{M}(k) = \tilde{\mathbf{H}}^-(k) - \mathbf{H}^-(k).$$

We now prove that $\mathbf{I} - 2\eta \mathbf{H}(k)\mathbf{P}$ is close to the transition matrix for the residual, up to sign flips, i.e., $\mathbf{y} - \mathbf{u}(k+1) \approx (\mathbf{I} - 2\eta \mathbf{H}(k)\mathbf{P})(\mathbf{y} - \mathbf{u}(k))$.

**Lemma 1.** Let $\mathbf{z}(k) = \mathbf{y} - \mathbf{u}(k)$ be the residual after the $k$th iteration. For any $k \geq 0$ and $\eta > 0$, we have

$$\left(\mathbf{I} - 2\eta \left(\tilde{\mathbf{H}}^+(k) + \mathbf{H}^-(k)\right)\mathbf{P}\right)\mathbf{z}(k) \leq \mathbf{z}(k+1) \leq \left(\mathbf{I} - 2\eta \left(\mathbf{H}^+(k) + \tilde{\mathbf{H}}^-(k)\right)\mathbf{P}\right)\mathbf{z}(k),$$

where the inequalities are entry-wise.

*Proof.* First, by the gradient descent update rule, we have

$$\tilde{\mathbf{w}}_r(k+1) - \tilde{\mathbf{w}}_r(k) = -\eta \frac{\partial \Phi_{\mathbf{P}}(\tilde{\mathbf{w}}_1(k), \dots, \tilde{\mathbf{w}}_m(k))}{\partial \tilde{\mathbf{w}}_r} = -\eta \frac{\partial \mathbf{u}(k)}{\partial \tilde{\mathbf{w}}_r} \frac{\partial \Phi_{\mathbf{P}}(\mathbf{u})}{\partial \mathbf{u}},$$

where the $(d+1) \times n$ Jacobian matrix is given by

$$\frac{\partial \mathbf{u}(k)}{\partial \tilde{\mathbf{w}}_r} = \frac{\sqrt{2}a_r}{\sqrt{m}} \left[\tilde{\mathbf{x}}_1 \mathbb{1}_{\{\tilde{\mathbf{x}}_1^\top \tilde{\mathbf{w}}_r(k) \geq 0\}} \quad \cdots \quad \tilde{\mathbf{x}}_n \mathbb{1}_{\{\tilde{\mathbf{x}}_n^\top \tilde{\mathbf{w}}_r(k) \geq 0\},}\right] \tag{27}$$

and the gradient of the loss function $\Phi_{\mathbf{P}}$ defined in eq. (6) with respect to $\mathbf{u}$ is given by

$$\frac{\partial \Phi_{\mathbf{P}}(\mathbf{u})}{\partial \mathbf{u}} = -\mathbf{P}(\mathbf{y} - \mathbf{u}(k)) = -\mathbf{P}\mathbf{z}(k), \tag{28}$$

which is a vector of length $n$. Hence, it follows that

$$\tilde{\mathbf{w}}_r(k+1)^\top \tilde{\mathbf{x}}_i - \tilde{\mathbf{w}}_r(k)^\top \tilde{\mathbf{x}}_i = \frac{\sqrt{2}\eta a_r}{\sqrt{m}} \sum_{p=1}^n (\mathbf{P}\mathbf{z}(k))_p \, \tilde{\mathbf{x}}_i^\top \tilde{\mathbf{x}}_p \mathbb{1}_{\{\tilde{\mathbf{x}}_p^\top \tilde{\mathbf{w}}_r(k) \geq 0\}},$$

where $(\mathbf{P}\mathbf{z}(k))_p$ denotes the $p$th element of $\mathbf{P}\mathbf{z}(k)$. Using the property of ReLU that

$$(b-a)\mathbb{1}_{\{a>0\}} \leq \text{ReLU}(b) - \text{ReLU}(a) \leq (b-a)\mathbb{1}_{\{b>0\}}, \qquad a, b \in \mathbb{R},$$

we have

$$\text{ReLU}\left(\tilde{\mathbf{w}}_r(k+1)^\top \tilde{\mathbf{x}}_i\right) - \text{ReLU}\left(\tilde{\mathbf{w}}_r(k)^\top \tilde{\mathbf{x}}_i\right)$$
$$\leq \frac{\sqrt{2}\eta a_r}{\sqrt{m}} \sum_{p=1}^n (\mathbf{P}\mathbf{z}(k))_p \, \tilde{\mathbf{x}}_i^\top \tilde{\mathbf{x}}_p \mathbb{1}_{\{\tilde{\mathbf{x}}_p^\top \tilde{\mathbf{w}}_r(k) \geq 0\}} \mathbb{1}_{\{\tilde{\mathbf{x}}_i^\top \tilde{\mathbf{w}}_r(k+1) \geq 0\}},$$

and

$$\text{ReLU}\left(\tilde{\mathbf{w}}_r(k+1)^\top \tilde{\mathbf{x}}_i\right) - \text{ReLU}\left(\tilde{\mathbf{w}}_r(k)^\top \tilde{\mathbf{x}}_i\right)$$
$$\geq \frac{\sqrt{2}\eta a_r}{\sqrt{m}} \sum_{p=1}^n (\mathbf{P}\mathbf{z}(k))_p \, \tilde{\mathbf{x}}_i^\top \tilde{\mathbf{x}}_p \mathbb{1}_{\{\tilde{\mathbf{x}}_p^\top \tilde{\mathbf{w}}_r(k) \geq 0\}} \mathbb{1}_{\{\tilde{\mathbf{x}}_i^\top \tilde{\mathbf{w}}_r(k) \geq 0\}}.$$

Hence, we have

$$
\begin{aligned}
(\mathbf{u}(k+1))_i - (\mathbf{u}(k))_i &= \frac{\sqrt{2}}{\sqrt{m}} \sum_{r \in \mathcal{A}} \left( \mathrm{ReLU}(\tilde{\mathbf{w}}_r(k+1)^\top \tilde{\mathbf{x}}_i) - \mathrm{ReLU}(\tilde{\mathbf{w}}_r(k)^\top \tilde{\mathbf{x}}_i) \right) \\
&\quad - \frac{\sqrt{2}}{\sqrt{m}} \sum_{r \in \mathcal{B}} \left( \mathrm{ReLU}(\tilde{\mathbf{w}}_r(k+1)^\top \tilde{\mathbf{x}}_i) - \mathrm{ReLU}(\tilde{\mathbf{w}}_r(k)^\top \tilde{\mathbf{x}}_i) \right) \\
&\leq \frac{2\eta}{m} \sum_{r \in \mathcal{A}} \sum_{p=1}^n (\mathbf{P}\mathbf{z}(k))_p \, \tilde{\mathbf{x}}_i^\top \tilde{\mathbf{x}}_p \mathbb{1}_{\{\tilde{\mathbf{x}}_p^\top \tilde{\mathbf{w}}_r(k) \geq 0\}} \mathbb{1}_{\{\tilde{\mathbf{x}}_i^\top \tilde{\mathbf{w}}_r(k+1) \geq 0\}} \\
&\quad + \frac{2\eta}{m} \sum_{r \in \mathcal{B}} \sum_{p=1}^n (\mathbf{P}\mathbf{z}(k))_p \, \tilde{\mathbf{x}}_i^\top \tilde{\mathbf{x}}_p \mathbb{1}_{\{\tilde{\mathbf{x}}_p^\top \tilde{\mathbf{w}}_r(k) \geq 0\}} \mathbb{1}_{\{\tilde{\mathbf{x}}_i^\top \tilde{\mathbf{w}}_r(k) \geq 0\}} \\
&= 2\eta \sum_{p=1}^n \left( \tilde{H}_{ip}^+(k) + H_{ip}^-(k) \right) (\mathbf{P}\mathbf{z}(k))_p \, .
\end{aligned}
$$

This proves the first inequality. The second inequality can be shown with a similar argument. $\qquad\square$

In particular, if there is no sign flip of the weights, then $\tilde{\mathbf{H}}(k) = \tilde{\mathbf{H}}^+(k) + \tilde{\mathbf{H}}^-(k) = \mathbf{H}(k)$ and the inequalities in Lemma 1 are equalities. Next, using Lemma 1, we can derive an expression for $\mathbf{y} - \mathbf{u}(k)$ using $\mathbf{H}^\infty$, up to an error term.

**Lemma 2.** For any $0 < \eta < 1/(2M_{\mathbf{P}}^2 n)$ and any $k \geq 0$, we have that

$$
\mathbf{y} - \mathbf{u}(k) = (\mathbf{I} - 2\eta \mathbf{H}^\infty \mathbf{P})^k (\mathbf{y} - \mathbf{u}(0)) + \boldsymbol{\epsilon}(k), \tag{29}
$$

where

$$
\begin{aligned}
\|\boldsymbol{\epsilon}(k)\|_{\mathbf{P}} &\leq 2\eta \sum_{t=0}^{k-1} \|(\mathbf{H}^\infty - \mathbf{H}(t))\mathbf{P}\|_{\mathbf{P}} \left\| (\mathbf{I} - 2\eta \mathbf{H}^\infty \mathbf{P})^t (\mathbf{y} - \mathbf{u}(0)) \right\|_{\mathbf{P}} \\
&\quad + 2\eta \sum_{t=0}^{k-1} (\|\mathbf{M}(t)\mathbf{P}\|_{\mathbf{P}} + \|\mathbf{L}(t)\mathbf{P}\|_{\mathbf{P}}) \|\mathbf{y} - \mathbf{u}(t)\|_{\mathbf{P}} \, .
\end{aligned} \tag{30}
$$

*Proof.* For $k \geq 1$, we define $\mathbf{r}(k)$ by

$$
\mathbf{r}(k) = \mathbf{y} - \mathbf{u}(k) - (\mathbf{I} - 2\eta \mathbf{H}(k-1)\mathbf{P})(\mathbf{y} - \mathbf{u}(k-1)). \tag{31}
$$

Then, by Lemma 1 we have

$$
\|\mathbf{r}(k)\|_{\mathbf{P}} \leq 2\eta (\|\mathbf{M}(k-1)\mathbf{P}\|_{\mathbf{P}} + \|\mathbf{L}(k-1)\mathbf{P}\|_{\mathbf{P}}) \|\mathbf{y} - \mathbf{u}(k-1)\|_{\mathbf{P}} \, . \tag{32}
$$

Note that eq. (31) is a first-order non-homogeneous recurrence relation for $\mathbf{y} - \mathbf{u}(k)$, which has an analytic solution. Thus, we can expand $\mathbf{y} - \mathbf{u}(k)$ for $k \geq 1$ as

$$
\begin{aligned}
\mathbf{y} - \mathbf{u}(k) &= ((\mathbf{I} - 2\eta \mathbf{H}(k-1)\mathbf{P}) \cdots (\mathbf{I} - 2\eta \mathbf{H}(0)\mathbf{P}))(\mathbf{y} - \mathbf{u}(0)) \\
&\quad + \mathbf{r}(k) + \sum_{t=1}^{k-1} ((\mathbf{I} - 2\eta \mathbf{H}(k-1)\mathbf{P}) \cdots (\mathbf{I} - 2\eta \mathbf{H}(t)\mathbf{P})) \mathbf{r}(t).
\end{aligned} \tag{33}
$$

Moreover, we can write the product of the matrices as (Su & Yang, 2019)

$$
\begin{aligned}
&(\mathbf{I} - 2\eta \mathbf{H}(k-1)\mathbf{P}) \cdots (\mathbf{I} - 2\eta \mathbf{H}(0)\mathbf{P}) \\
&= (\mathbf{I} - 2\eta \mathbf{H}^\infty \mathbf{P})^k + 2\eta (\mathbf{H}^\infty \mathbf{P} - \mathbf{H}(k-1)\mathbf{P})(\mathbf{I} - 2\eta \mathbf{H}^\infty \mathbf{P})^{k-1} \\
&\quad + 2\eta \sum_{t=1}^{k-1} ((\mathbf{I} - 2\eta \mathbf{H}(k-1)\mathbf{P}) \cdots (\mathbf{I} - 2\eta \mathbf{H}(t)\mathbf{P}))(\mathbf{H}^\infty \mathbf{P} - \mathbf{H}(t-1)\mathbf{P})(\mathbf{I} - 2\eta \mathbf{H}^\infty \mathbf{P})^{t-1}.
\end{aligned} \tag{34}
$$

Combining eq. (33) and eq. (34), we obtain eq. (29) where

$$\boldsymbol{\epsilon}(k) = 2\eta(\mathbf{H}^\infty\mathbf{P} - \mathbf{H}(k-1)\mathbf{P})(\mathbf{I} - 2\eta\mathbf{H}^\infty\mathbf{P})^{k-1}(\mathbf{y} - \mathbf{u}(0))$$

$$+ 2\eta\sum_{t=1}^{k-1}(\mathbf{I} - 2\eta\mathbf{H}(k-1)\mathbf{P})\cdots(\mathbf{I} - 2\eta\mathbf{H}(t)\mathbf{P})(\mathbf{H}^\infty - \mathbf{H}(t-1))\mathbf{P}(\mathbf{I} - 2\eta\mathbf{H}^\infty\mathbf{P})^{t-1}(\mathbf{y} - \mathbf{u}(0))$$

$$+ \mathbf{r}(k) + \sum_{t=1}^{k-1}(\mathbf{I} - 2\eta\mathbf{H}(k-1)\mathbf{P})\cdots(\mathbf{I} - 2\eta\mathbf{H}(t)\mathbf{P})\mathbf{r}(t).$$

Finally, we note that

$$\lambda_{\max}(\mathbf{H}(t)\mathbf{P}) = \lambda_{\max}(\mathbf{P}^{1/2}\mathbf{H}(t)\mathbf{P}^{1/2}) = \left\|\mathbf{P}^{1/2}\mathbf{H}(t)\mathbf{P}^{1/2}\right\|_2 = \sup_{\boldsymbol{\xi}\in\mathbb{R}^n\setminus\{\mathbf{0}\}}\frac{\left\|\mathbf{P}^{1/2}\mathbf{H}(t)\mathbf{P}^{1/2}\boldsymbol{\xi}\right\|_2}{\|\boldsymbol{\xi}\|_2}$$

$$= \sup_{\boldsymbol{\zeta}\in\mathbb{R}^n\setminus\{\mathbf{0}\}}\frac{\left\|\mathbf{P}^{1/2}\mathbf{H}(t)\mathbf{P}\boldsymbol{\zeta}\right\|_2}{\left\|\mathbf{P}^{1/2}\boldsymbol{\zeta}\right\|_2} = \sup_{\boldsymbol{\zeta}\in\mathbb{R}^n\setminus\{\mathbf{0}\}}\frac{\|\mathbf{H}(t)\mathbf{P}\boldsymbol{\zeta}\|_\mathbf{P}}{\|\boldsymbol{\zeta}\|_\mathbf{P}} = \|\mathbf{H}(t)\mathbf{P}\|_\mathbf{P}\,,$$

$$(35)$$

and that

$$\|\mathbf{H}(t)\mathbf{P}\|_\mathbf{P} = \sup_{\boldsymbol{\xi}\in\mathbb{R}^n\setminus\{\mathbf{0}\}}\frac{\|\mathbf{H}(t)\mathbf{P}\boldsymbol{\xi}\|_\mathbf{P}}{\|\boldsymbol{\xi}\|_\mathbf{P}} = \sup_{\boldsymbol{\xi}\in\mathbb{R}^n\setminus\{\mathbf{0}\}}\frac{\|\mathbf{H}(t)\mathbf{P}\boldsymbol{\xi}\|_\mathbf{P}}{\|\mathbf{H}(t)\mathbf{P}\boldsymbol{\xi}\|_2}\frac{\|\mathbf{H}(t)\mathbf{P}\boldsymbol{\xi}\|_2}{\|\mathbf{P}\boldsymbol{\xi}\|_2}\frac{\|\mathbf{P}\boldsymbol{\xi}\|_2}{\|\boldsymbol{\xi}\|_\mathbf{P}}. \qquad (36)$$

We can then bound $\|\mathbf{H}(t)\mathbf{P}\|_\mathbf{P}$ using $M_\mathbf{P}$ defined in eq. (8) and $\|\mathbf{H}(t)\|_2$ as

$$\|\mathbf{H}(t)\mathbf{P}\|_\mathbf{P} \le M_\mathbf{P}\|\mathbf{H}(t)\|_2\,M_\mathbf{P} \le M_\mathbf{P}^2\sqrt{\|\mathbf{H}(t)\|_1\|\mathbf{H}(t)\|_\infty} \le M_\mathbf{P}^2 n.$$

By requiring that $\eta < 1/(2M_\mathbf{P}^2 n)$, we have $\lambda_{\min}(\mathbf{I} - 2\eta\mathbf{H}(t)\mathbf{P}) > 0$ for all $t$. Hence, $\mathbf{I} - 2\eta\mathbf{H}(t)\mathbf{P}$ is positive definite in $(\mathbb{R}^n, \langle\cdot,\cdot\rangle_\mathbf{P})$, and according to eq. (35), we have $\|\mathbf{I} - 2\eta\mathbf{H}(t)\mathbf{P}\|_\mathbf{P} = \lambda_{\max}(\mathbf{I} - 2\eta\mathbf{H}(t)\mathbf{P}) < 1$. The upper bound in eq. (30) follows from the triangle inequality and our estimate on $\mathbf{r}_k$ in eq. (32). $\qquad\square$

The residual terms in Lemma 2 can be made small by controlling $\|\mathbf{M}(t)\mathbf{P}\|_\mathbf{P} + \|\mathbf{L}(t)\mathbf{P}\|_\mathbf{P}$ and $\|(\mathbf{H}(0) - \mathbf{H}(t))\mathbf{P}\|_\mathbf{P}$. Their upper bounds are given in Lemma 3. First, we define

$$S_i(t) = \left\{1 \le r \le m \mid \mathbb{1}_{\{\tilde{\mathbf{w}}_r(t')^\top\tilde{\mathbf{x}}_i \ge 0\}} \ne \mathbb{1}_{\{\tilde{\mathbf{w}}_r(0)^\top\tilde{\mathbf{x}}_i \ge 0\}} \text{ for some } 0 \le t' \le t\right\}$$

to be the set of indices of the weights that have changed the sign at least once by the $k$th iteration.

**Lemma 3.** For all $t \ge 0$, we have

$$\max\left(\|\mathbf{M}(t)\mathbf{P}\|_\mathbf{P} + \|\mathbf{L}(t)\mathbf{P}\|_\mathbf{P}, \|(\mathbf{H}(0) - \mathbf{H}(t))\mathbf{P}\|_\mathbf{P}\right) \le \sqrt{\frac{4M_\mathbf{P}^4 n}{m^2}\sum_{i=1}^n|S_i(t)|^2}.$$

Moreover, for any $0 < \delta < 1$, with probability at least $1 - \delta$, we have

$$\|(\mathbf{H}^\infty - \mathbf{H}(0))\mathbf{P}\|_\mathbf{P} \le 2M_\mathbf{P}^2 n\sqrt{\frac{\log(2n/\delta)}{m}}.$$

*Proof.* First, we have

$$\|\mathbf{M}(t)\mathbf{P}\|_\mathbf{P}^2 \le M_\mathbf{P}^4\|\mathbf{M}(t)\|_2^2 \le M_\mathbf{P}^4\|\mathbf{M}(t)\|_F^2$$

$$\le \frac{M_\mathbf{P}^4}{m^2}\sum_{i=1}^n\sum_{p=1}^n\left(\sum_{r\in\mathcal{A}}\left|\mathbb{1}_{\{\tilde{\mathbf{w}}_r(t)^\top\tilde{\mathbf{x}}_i \ge 0, \tilde{\mathbf{w}}_r(t)^\top\tilde{\mathbf{x}}_p \ge 0\}} - \mathbb{1}_{\{\tilde{\mathbf{w}}_r(t+1)^\top\tilde{\mathbf{x}}_i \ge 0, \tilde{\mathbf{w}}_r(t)^\top\tilde{\mathbf{x}}_p \ge 0\}}\right|\right)^2$$

$$\le \frac{M_\mathbf{P}^4 n}{m^2}\sum_{i=1}^n|S_i(t)|^2.$$

The estimate for $\|\mathbf{L}(t)\mathbf{P}\|_{\mathbf{P}}^2$ is exactly the same and obtained by replacing $\mathcal{A}$ with $\mathcal{B}$. We also have

$$
\begin{aligned}
\|(\mathbf{H}(0) - \mathbf{H}(t))\mathbf{P}\|_{\mathbf{P}}^2 &\leq M_{\mathbf{P}}^4 \|\mathbf{H}(0) - \mathbf{H}(t)\|_2^2 \leq M_{\mathbf{P}}^4 \|\mathbf{H}(0) - \mathbf{H}(t)\|_F^2 \\
&\leq \frac{M_{\mathbf{P}}^4}{m^2} \sum_{i=1}^{n} \sum_{p=1}^{n} \left( \sum_{r=1}^{m} |\mathbb{1}_{\{\tilde{\mathbf{w}}_r(0)^\top \tilde{\mathbf{x}}_i \geq 0, \tilde{\mathbf{w}}_r(0)^\top \tilde{\mathbf{x}}_p \geq 0\}} - \mathbb{1}_{\{\tilde{\mathbf{w}}_r(t)^\top \tilde{\mathbf{x}}_i \geq 0, \tilde{\mathbf{w}}_r(t)^\top \tilde{\mathbf{x}}_p \geq 0\}} | \right)^2 \\
&\leq \frac{M_{\mathbf{P}}^4}{m^2} \sum_{i=1}^{n} \sum_{p=1}^{n} (|S_i(t)| + |S_p(t)|)^2 \leq \frac{M_{\mathbf{P}}^4}{m^2} \sum_{i=1}^{n} \sum_{p=1}^{n} \left( 2|S_i(t)|^2 + 2|S_p(t)|^2 \right) \\
&= \frac{4 M_{\mathbf{P}}^4 n}{m^2} \sum_{i=1}^{n} |S_i(t)|^2 .
\end{aligned}
$$

This proves the first inequality. Since $H_{ij}(0)$ is the average of $m$ iid random variables bounded in $[0, 1]$, by Hoeffding's inequality (Hoeffding, 1963), for any $t > 0$ and any $1 \leq i, j \leq n$, we have

$$
\mathbb{P}\left( m \left| H_{ij}(0) - H_{ij}^\infty \right| \geq t \right) \leq 2\exp\left( -\frac{2t^2}{m} \right) \leq 2\exp\left( -\frac{t^2}{m} \right). \tag{37}
$$

Set $t = \sqrt{m \log(2n^2/\delta)}$. With probability at least $1 - \delta/n^2$, we have

$$
\left| H_{ij}(0) - H_{ij}^\infty \right| \leq \sqrt{\frac{\log(2n^2/\delta)}{m}} \leq 2\sqrt{\frac{\log(2n/\delta)}{m}}. \tag{38}
$$

Hence, by a union bound, we know that with probability at least $1 - \delta$, we have

$$
\|\mathbf{H}^\infty - \mathbf{H}(0)\|_2 \leq \|\mathbf{H}^\infty - \mathbf{H}(0)\|_F \leq 2n\sqrt{\frac{\log(2n/\delta)}{m}}.
$$

The last estimate follows from the definitions of $M_{\mathbf{P}}$ (see eq. (8) and eq. (36)). $\qquad \square$

Now, we state and prove our initial control of the decay of the residual.

**Lemma 4.** *Let $\epsilon > 0, \kappa > 0, 0 < \delta < 1$ and $T > 0$ be given. There exist constants $C_m, C_m' > 0$ such that if $0 \leq \eta \leq 1/(2M_{\mathbf{P}}^2 n)$ and $m$ satisfies*

$$
m \geq C_m \frac{M_{\mathbf{P}}^6 n^3}{\kappa^2 \epsilon^2} \left( \lambda_0^{-4} \left( 1 + \frac{\kappa^2 M_{\mathbf{P}}^2 n}{\delta} \right)^2 + \eta^4 T^4 \epsilon^4 \right)
$$

*and*

$$
m \geq C_m' \frac{M_{\mathbf{P}}^4 n^2 \log(n/\delta)}{\epsilon^2} \left( \lambda_0^{-2} \left( 1 + \frac{\kappa^2 M_{\mathbf{P}}^2 n}{\delta} \right) + \eta^2 T^2 \epsilon^2 \right),
$$

*then with probability at least $1 - \delta$, we have the following for all $0 \leq k \leq T$:*

$$
\mathbf{y} - \mathbf{u}(k) = (\mathbf{I} - 2\eta \mathbf{H}^\infty \mathbf{P})^k (\mathbf{y} - \mathbf{u}(0)) + \boldsymbol{\epsilon}(k), \qquad \|\boldsymbol{\epsilon}(k)\|_{\mathbf{P}} \leq \epsilon. \tag{39}
$$

*Proof.* Set $\delta' = \delta/3$. For any $R > 0$ and $r = 1, \ldots, m$, since $\mathbf{w}_r(0)^\top \mathbf{x}_i \sim \mathcal{N}(0, \kappa^2)$, we have

$$
\mathbb{P}\left( \left| \tilde{\mathbf{w}}_r(0)^\top \tilde{\mathbf{x}}_i \right| \leq R \right) = \mathbb{E}\left[ \mathbb{1}_{\{|\tilde{\mathbf{w}}_r(0)^\top \tilde{\mathbf{x}}_i| \leq R\}} \right] < \frac{2R}{\sqrt{\pi}\kappa}.
$$

By Hoeffding's inequality (Hoeffding, 1963), for any $t > 0$ we have

$$
\mathbb{P}\left( \sum_{r=1}^{m} \mathbb{1}_{\{|\tilde{\mathbf{w}}_r(0)^\top \tilde{\mathbf{x}}_i| \leq R\}} \geq \frac{2mR}{\sqrt{\pi}\kappa} + t \right) \leq \exp\left( -\frac{2t^2}{m} \right) \leq \exp\left( -\frac{t^2}{m} \right), \qquad 1 \leq i \leq n. \tag{40}
$$

Thus, if we set $t = \sqrt{m \log(n/\delta')}$ then we find that with probability at least $1 - \delta'/n$ we have

$$
\sum_{r=1}^{m} \mathbb{1}_{\{|\tilde{\mathbf{w}}_r(0)^\top \tilde{\mathbf{x}}_i| \leq R\}} \leq \frac{2mR}{\sqrt{\pi}\kappa} + \sqrt{m \log(n/\delta')} \leq 2m \left( \frac{R}{\sqrt{\pi}\kappa} + \sqrt{\frac{\log(n/\delta')}{m}} \right).
$$

By a union bound, we have with probability at least $1 - \delta'$,

$$\sum_{i=1}^{n} \left( \sum_{r=1}^{m} \mathbb{1}_{\{|\tilde{\mathbf{w}}_r(0)^\top \tilde{\mathbf{x}}_i| \le R\}} \right)^2 \le 4m^2 n \left( \frac{R}{\sqrt{\pi}\kappa} + \sqrt{\frac{\log(n/\delta')}{m}} \right)^2 .$$

By combining this with Lemma 3, we have that with probability at least $1 - 2\delta'$,

$$\sqrt{\frac{4M_{\mathbf{P}}^4 n}{m^2} \sum_{i=1}^{n} \left( \sum_{r=1}^{m} \mathbb{1}_{\{|\tilde{\mathbf{w}}_r(0)^\top \tilde{\mathbf{x}}_i| \le R\}} \right)^2} \le 4M_{\mathbf{P}}^2 n \left( \frac{R}{\sqrt{\pi}\kappa} + \sqrt{\frac{\log(n/\delta')}{m}} \right), \tag{41}$$

and

$$\|(\mathbf{H}^\infty - \mathbf{H}(0))\mathbf{P}\|_{\mathbf{P}} \le 2M_{\mathbf{P}}^2 n \sqrt{\frac{\log(2n/\delta')}{m}}. \tag{42}$$

Since the $i$th entry of $\mathbf{u}(0)$ has mean 0 and variance $\le \kappa^2$, we have $\mathbb{E}[(\mathbf{u}(0))_i^2] \le \kappa^2$, where $(\mathbf{u}(0))_i$ is the $i$th entry of $\mathbf{u}(0)$. Hence, we have $\mathbb{E}[\|\mathbf{u}(0)\|_{\mathbf{P}}^2] \le M_{\mathbf{P}}^2 n\kappa^2$. By Markov's inequality, with probability at least $1 - \delta'$, we have

$$\|\mathbf{u}(0)\|_{\mathbf{P}} \le \kappa M_{\mathbf{P}} \sqrt{n/\delta'}, \qquad \|\mathbf{y} - \mathbf{u}(0)\|_{\mathbf{P}} \le \|\mathbf{y}\|_{\mathbf{P}} + \kappa M_{\mathbf{P}} \sqrt{n/\delta'}. \tag{43}$$

By a union bound, we know that eqs. (41) to (43) hold with probability of at least $1 - 3\delta'$. The theorem now follows using induction, where the base case when $k = 0$ is obvious. Assume eq. (39) holds for $t = 0, \ldots, k-1$, where $1 \le k \le T$. Then, we have

$$2\eta \sum_{t=0}^{k-1} \|\mathbf{y} - \mathbf{u}(t)\|_{\mathbf{P}} \le 2\eta \sum_{t=0}^{k-1} \left[ (1 - 2\eta\lambda_0)^t \|\mathbf{y} - \mathbf{u}(0)\|_{\mathbf{P}} + \epsilon \right] \le \lambda_0^{-1} \|\mathbf{y} - \mathbf{u}(0)\|_{\mathbf{P}} + 2\eta T\epsilon, \tag{44}$$

where the first inequality follows from the fact that $\mathbf{I} - 2\eta\mathbf{H}^\infty\mathbf{P}$ is positive semidefinite in $(\mathbb{R}^n, \langle \cdot, \cdot \rangle_{\mathbf{P}})$ with the maximum eigenvalue being $1 - 2\eta\lambda_0$, and the second inequality follows by bounding the power series. By the definition of $\lambda_0$, we have

$$2\eta \sum_{t=0}^{k-1} \left\| (\mathbf{I} - 2\eta\mathbf{H}^\infty\mathbf{P})^t (\mathbf{y} - \mathbf{u}(0)) \right\|_{\mathbf{P}} \le 2\eta \sum_{t=0}^{k-1} (1 - 2\eta\lambda_0)^t \|\mathbf{y} - \mathbf{u}(0)\|_{\mathbf{P}} \le \lambda_0^{-1} \|\mathbf{y} - \mathbf{u}(0)\|_{\mathbf{P}}. \tag{45}$$

By Lemma 2 and 3, we have

$$\mathbf{y} - \mathbf{u}(k) = (\mathbf{I} - 2\eta\mathbf{H}^\infty\mathbf{P})^k (\mathbf{y} - \mathbf{u}(0)) + \boldsymbol{\epsilon}(k)$$

with

$$\|\boldsymbol{\epsilon}(k)\|_{\mathbf{P}} \le \lambda_0^{-1} \|(\mathbf{H}(0) - \mathbf{H}^\infty)\mathbf{P}\|_{\mathbf{P}} \|\mathbf{y} - \mathbf{u}(0)\|_{\mathbf{P}}$$
$$+ 2\sqrt{\frac{4M_{\mathbf{P}}^4 n}{m^2} \sum_{i=1}^{n} |S_i(k)|^2} \left( \lambda_0^{-1} \|\mathbf{y} - \mathbf{u}(0)\|_{\mathbf{P}} + \eta T\epsilon \right), \tag{46}$$

where we used the fact that $|S_i(t)|$ is a nondecreasing function of $t$ and the triangle inequality $\|(\mathbf{H}^\infty - \mathbf{H}(t))\mathbf{P}\|_{\mathbf{P}} \le \|(\mathbf{H}(0) - \mathbf{H}^\infty)\mathbf{P}\|_{\mathbf{P}} + \|(\mathbf{H}(t) - \mathbf{H}(0))\mathbf{P}\|_{\mathbf{P}}$. Here, we also combined $\lambda_0^{-1} \|(\mathbf{H}(t) - \mathbf{H}(0))\mathbf{P}\|_{\mathbf{P}} \|\mathbf{y} - \mathbf{u}(0)\|_{\mathbf{P}}$ with the last term on the right-hand side of eq. (30) and applied Lemma 3, eq. (44), and eq. (45) to obtain the last term on the right-hand side of eq. (46). To control $|S_i(k)|$, we first bound the change of the weights. For any $0 \le t \le k-1$ and $1 \le r \le m$, the change of weights in one iteration can be bounded by

$$\|\tilde{\mathbf{w}}_r(t+1) - \tilde{\mathbf{w}}_r(t)\|_2 = \eta \left\| \frac{\partial \mathbf{u}(t)}{\partial \tilde{\mathbf{w}}_r} \frac{\partial \Phi_{\mathbf{P}}(\mathbf{u})}{\partial \mathbf{u}} \right\|_2$$
$$\le \eta \left\| \frac{\partial \mathbf{u}(t)}{\partial \tilde{\mathbf{w}}_r} \right\|_F \|\mathbf{P}(\mathbf{y} - \mathbf{u}(t))\|_2 \le \eta \sqrt{\frac{2n}{m}} M_{\mathbf{P}} \|\mathbf{y} - \mathbf{u}(t)\|_{\mathbf{P}},$$

where the inequalities follow from eq. (27) and eq. (28). Hence, the total change of the weights can be bounded by

$$\|\tilde{\mathbf{w}}_r(t) - \tilde{\mathbf{w}}_r(0)\|_2 \le \eta M_{\mathbf{P}} \sqrt{\frac{2n}{m}} \sum_{t'=0}^{t-1} \|\mathbf{y} - \mathbf{u}(t')\|_{\mathbf{P}} \le R_T, \tag{47}$$

where $R_T = M_{\mathbf{P}} \sqrt{\frac{n}{2m}} \left( \lambda_0^{-1} \|\mathbf{y} - \mathbf{u}(0)\|_{\mathbf{P}} + 2\eta T \epsilon \right)$. Recall that $S_i(k)$ is the set of indices of weights that have gone through at least one sign flip by iteration $k$. Thus, if $r \in S_i(k)$, we have $\|\tilde{\mathbf{w}}_r(t) - \tilde{\mathbf{w}}_r(0)\|_2 \geq \left| \tilde{\mathbf{w}}_r(0)^\top \tilde{\mathbf{x}}_i \right|$ for some $0 \leq t \leq k$ as the sign flip leads to $\left| \tilde{\mathbf{w}}_r(0)^\top \tilde{\mathbf{x}}_i \right| \leq \left| \tilde{\mathbf{w}}_r(0)^\top \tilde{\mathbf{x}}_i - \tilde{\mathbf{w}}_r(t)^\top \tilde{\mathbf{x}}_i \right|$. This gives us

$$
\begin{aligned}
|S_i(k)| &\leq \left| \left\{ r \in [m] : \left| \tilde{\mathbf{w}}_r(0)^\top \tilde{\mathbf{x}}_i \right| \leq \|\tilde{\mathbf{w}}_r(t) - \tilde{\mathbf{w}}_r(0)\|_2 \text{ for some } 0 \leq t \leq k \right\} \right| \\
&\leq \left| \left\{ r \in [m] : \left| \tilde{\mathbf{w}}_r(0)^\top \tilde{\mathbf{x}}_i \right| \leq R_T \right\} \right|,
\end{aligned}
\tag{48}
$$

where $[m] = \{1, \ldots, m\}$. Hence, there exists a constant $C > 0$ such that

$$
\begin{aligned}
\|\boldsymbol{\epsilon}(k)\|_{\mathbf{P}} &\leq 2M_{\mathbf{P}}^2 n \sqrt{\frac{\log(2n/\delta')}{m}} \lambda_0^{-1} \|\mathbf{y} - \mathbf{u}(0)\|_{\mathbf{P}} \\
&\quad + 8M_{\mathbf{P}}^2 n \left( \frac{R_T}{\sqrt{\pi}\kappa} + \sqrt{\frac{\log(n/\delta')}{m}} \right) \left( \lambda_0^{-1} \|\mathbf{y} - \mathbf{u}(0)\|_{\mathbf{P}} + \eta T \epsilon \right) \\
&\leq 2M_{\mathbf{P}}^2 n \underbrace{\sqrt{\frac{\log(6n/\delta)}{m}} \lambda_0^{-1} C \left( 1 + \kappa M_{\mathbf{P}} \sqrt{\frac{3n}{\delta}} \right)}_{A_1} \\
&\quad + 8M_{\mathbf{P}}^2 n \left( \underbrace{\frac{1}{\sqrt{\pi}\kappa} M_{\mathbf{P}} \sqrt{\frac{n}{2m}} \left( \lambda_0^{-1} C \left( 1 + \kappa M_{\mathbf{P}} \sqrt{\frac{3n}{\delta}} \right) + 2\eta T \epsilon \right)}_{A_2} + \underbrace{\sqrt{\frac{\log(3n/\delta)}{m}}}_{A_3} \right) \\
&\quad \times \underbrace{\left( \lambda_0^{-1} C \left( 1 + \kappa M_{\mathbf{P}} \sqrt{\frac{3n}{\delta}} \right) + \eta T \epsilon \right)}_{A_4},
\end{aligned}
\tag{49}
$$

where the first inequality follows from eq. (41), eq. (46), and eq. (48), and the second inequality follows from eq. (43) and eq. (47). Finally, eq. (39) follows from the way we define $m$. By taking $C_m$ large enough, we guarantee that $2M_{\mathbf{P}}^2 n A_1, 8M_{\mathbf{P}}^2 n A_2 A_4 < \epsilon/3$. By taking $C_m'$ large enough, we guarantee that $8M_{\mathbf{P}}^2 n A_3 A_4 < \epsilon/3$. Hence, eq. (39) follows. $\qquad \square$

Lemma 4 gives us an estimate of the residual $\mathbf{y} - \mathbf{u}(k)$ in terms of the initial residual $\mathbf{y} - \mathbf{u}(0)$. However, in analyzing the frequency bias, we hope to express the residual in terms of $\mathbf{y}$ only. This can be done by controlling the size of $\mathbf{u}(0)$. First, we note that the proof of eq. (43) does not rely on the assumptions on $m$ in Lemma 4. Hence, it holds for any $n, m \geq 1, \kappa > 0, 0 < \delta < 1$, and positive definite matrix $\mathbf{P}$. Now we are ready to prove our first main result Theorem 1.

*Proof of Theorem 1.* By eq. (43), with probability at least $1 - \delta/2$, we have

$$
\left\| (\mathbf{I} - \eta \mathbf{H}^\infty \mathbf{P})^k \mathbf{u}(0) \right\|_{\mathbf{P}} \leq \|\mathbf{u}(0)\|_{\mathbf{P}} \leq \kappa M_{\mathbf{P}} \sqrt{2n/\delta}.
\tag{50}
$$

By taking $\kappa \leq \epsilon \sqrt{\delta/2n}/(2M_{\mathbf{P}})$, we guarantee that

$$
\left\| (\mathbf{I} - \eta \mathbf{H}^\infty \mathbf{P})^k \mathbf{u}(0) \right\|_{\mathbf{P}} \leq \epsilon/2.
$$

By the way we pick $\kappa$ and $m$, for some constant $C' > 0$ that only depends on $d$, we have

$$
1 + \frac{\kappa^2 M_{\mathbf{P}}^2 n}{\delta} \leq C'.
$$

Hence, by taking $C_2$ in eq. (13) large enough, we guarantee that $m$ satisfies the assumptions in Lemma 4 with $\epsilon, \kappa, T$, and $\delta$ to be $\epsilon/2, \kappa, T$, and $\delta/2$, respectively. Then, we have eq. (39) is true with probability of at least $1 - \delta/2$, for which $\|\boldsymbol{\epsilon}(k)\|_{\mathbf{P}} \leq \epsilon/2$. The result follows from the triangle inequality and union bound. $\qquad \square$

Notably, there are other initialization schemes that allow us to avoid using $\kappa$. One of the examples is to initialize the weights at odd indices $\mathbf{w}_{2p+1}$ and $a_{2p+1}$ randomly and set $\mathbf{w}_{2p+2} = \mathbf{w}_{2p+1}$, $a_{2p+2} = -a_{2r+1}$, assuming $m$ is even (Su & Yang, 2019). This initialization scheme guarantees that $\mathbf{u}(0) = \mathbf{0}$ and hence we do not need to introduce $\kappa$ to control the initialization size $\|\mathbf{u}(0)\|_{\mathbf{P}}$. In addition, if we assume that each entry of $\mathbf{w}_r$ is initialized from an iid sub-Gaussian distribution with zero mean and whose support is the entire $\mathbb{R}$, then since $\mathbf{H}^\infty$ is still SPD (see the remark at the end of Appendix B) and the Hoeffding's inequality holds, the proof does not break down and a result similar to Theorem 1 can be shown.

Following the same steps of proof, we can study the case when the gradient descent steps are slightly perturbed. That is, Suppose we perturb the output of the NN by $\delta \mathbf{u}_j$ at the $j$th iteration. Then, we expect that the residual of the NN at the $k$th iteration is approximately

$$\mathbf{y} - \mathbf{u}(k) \approx (\mathbf{I} - 2\eta \mathbf{H}^\infty \mathbf{P}) \cdots ((\mathbf{I} - 2\eta \mathbf{H}^\infty \mathbf{P}) ((\mathbf{I} - 2\eta \mathbf{H}^\infty \mathbf{P}) \mathbf{y} + \delta \mathbf{u}_1) + \delta \mathbf{u}_2) + \cdots + \delta \mathbf{u}_k$$

$$= (\mathbf{I} - 2\eta \mathbf{H}^\infty \mathbf{P})^k \mathbf{y} + \sum_{j=1}^{k} (\mathbf{I} - 2\eta \mathbf{H}^\infty \mathbf{P})^{k-j} \delta \mathbf{u}_j.$$

Since the maximum eigenvalue of $\mathbf{I} - 2\eta \mathbf{H}^\infty \mathbf{P}$ is less than one, we can then control the errors in this approximation using arguments similar to previous lemmas.

In Theorem 1, we showed how the parameters of the NN should depend on the desired maximum error $\epsilon$. Sometimes, it is also very useful to understand the dependence of $\epsilon$ on the parameters $\eta, T, n, m$, etc. Therefore, we present the following result to show this dependency.

**Theorem 1'.** *In eq. (5), suppose that $\mathbf{w}_1, \ldots, \mathbf{w}_m$ are initialized iid from Gaussian random variables with covariance matrix $\kappa^2 \mathbf{I}$, $b_1, \ldots, b_m$ are initialized to zero, and $a_1, \ldots, a_m$ are initialized iid as $+1$ with probability $1/2$ and $-1$ otherwise. Suppose the NN is trained with training data $(\mathbf{x}_i, y_i)$ for $1 \leq i \leq n$, loss function $\Phi_{\mathbf{P}}$ in eq. (6) for a symmetric positive definite matrix $\mathbf{P}$, and the training procedure is the gradient descent update rule eq. (7) with step size $\eta \leq 1/(2M_{\mathbf{P}}^2 n)$. Let $\mathcal{N}_k$ be the NN function after the $k$th iteration and $\mathbf{u}(k) = (\mathcal{N}_k(\mathbf{x}_1), \ldots, \mathcal{N}_k(\mathbf{x}_n))$, where $\mathcal{N}_0$ is the initial NN function. Let a probability of failure $0 < \delta < 1$ be given. Then, there exists a constant $C > 0$ that depends only on the dimension $d$ such that with probability $\geq 1 - \delta$, the following statement holds: for any $k \geq 1$, if we define $\epsilon_k$ by*

$$\epsilon_k = \max_{1 \leq t \leq k-1} \left\| \mathbf{y} - \mathbf{u}(t) - (\mathbf{I} - 2\eta \mathbf{H}^\infty \mathbf{P})^t \mathbf{y} \right\|_{\mathbf{P}},$$

*then we have*

$$\left\| \mathbf{y} - \mathbf{u}(k) - (\mathbf{I} - 2\eta \mathbf{H}^\infty \mathbf{P})^k \mathbf{y} \right\|_{\mathbf{P}} \leq 2M_{\mathbf{P}}^2 n \sqrt{\frac{\log(6n/\delta)}{m}} \lambda_0^{-1} C \left( 1 + \kappa M_{\mathbf{P}} \sqrt{\frac{3n}{\delta}} \right)$$

$$+ 8M_{\mathbf{P}}^2 n \left( \frac{1}{\sqrt{\pi}\kappa} M_{\mathbf{P}} \sqrt{\frac{n}{2m}} \left( \lambda_0^{-1} C \left( 1 + \kappa M_{\mathbf{P}} \sqrt{\frac{3n}{\delta}} \right) + 2\eta k \epsilon_k \right) + \sqrt{\frac{\log(3n/\delta)}{m}} \right) \quad (51)$$

$$\times \left( \lambda_0^{-1} C \left( 1 + \kappa M_{\mathbf{P}} \sqrt{\frac{3n}{\delta}} \right) + \eta k \epsilon_k \right) + \kappa M_{\mathbf{P}} \sqrt{\frac{2n}{\delta}}.$$

*Proof.* The result follows immediately from eq. (49) and eq. (50). $\qquad\square$

While this paper is primarily concerned with learning a continuous function using loss functions that are adapted from MSE, we briefly discuss the changes that are needed to study classifiers trained by cross-entropy. If the classification task has $p$ classes, then our NN has $p$ outputs, each of which is then passed through a softmax layer and represents an estimate of the likelihood that the input belongs to the corresponding class. More information on the NN architecture and the cross-entropy loss function can be found in (Kurbiel, 2021). Let $\mathcal{N}_j(\mathbf{x}; \mathbf{W})$ be the $j$th entry of the outputs of the NN and let $u_j(\mathbf{x}; \mathbf{W}) = \text{softmax}(\mathcal{N}_j(\mathbf{x}; \mathbf{W}))$, $1 \leq j \leq p$. Let $g_j(\mathbf{x})$ be the ground-truth of the $j$th entry of the outputs and let $z_j(\mathbf{x}; \mathbf{W}) = g_j(\mathbf{x}) - u_j(\mathbf{x}; \mathbf{W})$ be the residual. Assume we use the gradient flow and have access to data on the entire domain. Then, to derive a formula analogous

to eq. (2), for each fixed $\mathbf{x}$, we have

$$\frac{dz_j(\mathbf{x};\mathbf{W})}{dt} = -\frac{du_j(\mathbf{x};\mathbf{W})}{dt} = -\sum_{i=1}^{p} \frac{du_j(\mathbf{x};\mathbf{W})}{d\mathcal{N}_i} \frac{d\mathcal{N}_i(\mathbf{x};\mathbf{W})}{dt}$$

$$= -\sum_{i=1}^{p} \frac{du_j(\mathbf{x};\mathbf{W})}{d\mathcal{N}_i} \frac{\partial \mathcal{N}_i(\mathbf{x};\mathbf{W})}{\partial \mathbf{W}} \frac{d\mathbf{W}}{dt} = \sum_{i=1}^{p} \frac{du_j(\mathbf{x};\mathbf{W})}{d\mathcal{N}_i} \frac{\partial \mathcal{N}_i(\mathbf{x};\mathbf{W})}{\partial \mathbf{W}} \left( \frac{\partial \mathcal{L}(\mathbf{W})}{\partial \mathbf{W}} \right)^\top$$

$$= \sum_{i=1}^{p} \left( \frac{du_j(\mathbf{x};\mathbf{W})}{d\mathcal{N}_i} \frac{\partial \mathcal{N}_i(\mathbf{x};\mathbf{W})}{\partial \mathbf{W}} \sum_{i'=1}^{p} \int_{\mathbb{S}^{d-1}} \left( -z_{i'}(\mathbf{x}';\mathbf{W}) \frac{\partial \mathcal{N}_{i'}(\mathbf{x}';\mathbf{W})}{\partial \mathbf{W}} \right)^\top d\mu(\mathbf{x}') \right),$$

where $\mathcal{L}$ is the cross-entropy loss function computed as an integral over $\mathbb{S}^{d-1}$ and in the last step we used the fact that $(d/d\mathcal{N}_{i'})\mathcal{L}(\mathbf{x}';\mathbf{W}) = u_{i'}(\mathbf{x}';\mathbf{W}) - g_{i'}(\mathbf{x}') = -z_{i'}(\mathbf{x}';\mathbf{W})$ (Kurbiel, 2021). Hence, eq. (2) now becomes

$$\frac{dz_j(\mathbf{x};\mathbf{W})}{dt} = -\sum_{i=1}^{p} \left( \frac{du_j(\mathbf{x};\mathbf{W})}{d\mathcal{N}_i} \sum_{i'=1}^{p} \int_{\mathbb{S}^{d-1}} \underbrace{\left\langle \frac{\partial \mathcal{N}_i}{\partial \mathbf{W}}(\mathbf{x};\mathbf{W}), \frac{\partial \mathcal{N}_{i'}}{\partial \mathbf{W}}(\mathbf{x}';\mathbf{W}) \right\rangle}_{=K_{i,i'}(\mathbf{x},\mathbf{x}';\mathbf{W})} z_{i'}(\mathbf{x}';\mathbf{W}) d\mu(\mathbf{x}') \right),$$

where we also know that $(d/d\mathcal{N}_i)u_j(\mathbf{x};\mathbf{W}) = u_i(\mathbf{x};\mathbf{W})(\mathbb{1}_{\{i=j\}} - u_j(\mathbf{x};\mathbf{W}))$. Again, we define $\mathbf{H}_{i,i'}^\infty$ as the discretization of $K_{i,i'}$ in expectation over random initialization of $\mathbf{W}$. Note that $\mathbf{H}_{i,i}^\infty$ coincides with $\mathbf{H}^\infty$ that we used extensively in this paper. However, the entries of $\mathbf{H}_{i,i'}^\infty$ might differ with $\mathbf{H}^\infty$ in signs. This potentially causes difficulties in analyzing the spectrum of $\mathbf{H}_{i,i'}^\infty$. Now, by the formula above, we expect that the residual can approximately be written as

$$\mathbf{z}_j(k+1) - \mathbf{z}_j(k) = [g_j(\mathbf{x}) - \mathbf{u}_j(k+1)] - [g_j(\mathbf{x}) - \mathbf{u}_j(k)]$$

$$\approx -\eta \sum_{i=1}^{p} \left( \mathbf{u}_i(k) \circ ([\mathbb{1}_{\{i=j\}}, \cdots, \mathbb{1}_{\{i=j\}}]^\top - \mathbf{u}_j(k)) \circ \sum_{i'=1}^{p} \left( \mathbf{H}_{i,i'}^\infty \mathbf{P} \mathbf{z}_{i'}(k) \right) \right), \tag{52}$$

where '$\circ$' is the Hadamard product, $\mathbf{u}_j(k) = [u_j(\mathbf{x}_1;\mathbf{W}(k)), \cdots, u_j(\mathbf{x}_n;\mathbf{W}(k))]^\top$, and $\mathbf{z}_j(k) = [z_j(\mathbf{x}_1;\mathbf{W}(k)), \cdots, z_j(\mathbf{x}_n;\mathbf{W}(k))]^\top$. Suppose we define the vectorized residual $\mathbf{z} = [\mathbf{z}_1^\top, \ldots, \mathbf{z}_p^\top]^\top$ and define the $np \times np$ block matrix $\mathbf{J}(k)$ by $\mathbf{J}(k)_{(in+1):(i+1)n,(jn+1):(j+1)n} = \text{diag}\left( \mathbf{u}_i(k) \circ ([\mathbb{1}_{\{i=j\}}, \cdots, \mathbb{1}_{\{i=j\}}]^\top - \mathbf{u}_j(k)) \right)$ for $i, j = 0, \ldots, p-1$. Let $\mathcal{H}^\infty$ be the $np \times np$ block matrix such that $\mathcal{H}_{(in+1):(i+1)n,(i'n+1):(i'+1)n}^\infty = \mathbf{H}_{i,i'}^\infty$ for $i, i' = 0, \ldots, p-1$. Then, eq. (52) can be written compactly as

$$\mathbf{z}(k+1) - \mathbf{z}(k) \approx -\eta \mathbf{J}(k) \mathcal{H}^\infty (\mathbf{I}_p \otimes \mathbf{P}) \mathbf{z}(k), \tag{53}$$

where '$\otimes$' is the Kronecker product. Frequency bias can be analyzed by studying the dynamics of $\mathbf{z}(k)$ based on eq. (53). However, the fact that $\mathbf{J}$ depends on $k$ is expected to add complication to the analysis.

## D  THE THEORY OF FREQUENCY BIAS WITH AN $\mathbf{L}^2$-BASED LOSS FUNCTION

In this section, we prove the results stated in Section 4, where we are concerned with the frequency bias behavior of NN training when using the squared $L^2$ norm as the loss function. We theoretically show the frequency bias phenomena in this setting, up to a quadrature error.

### D.1  A CONSEQUENCE OF THEOREM 1

Given a bandlimited function $g : \mathbb{S}^{d-1} \to \mathbb{R}$ with bandlimit $L$, we can uniquely decompose $g$ into a spherical harmonic expansion as $g(\mathbf{x}) = \sum_{\ell=0}^{L} g_\ell(\mathbf{x})$, where $g_\ell(\mathbf{x}) \in \mathcal{H}_\ell^d$. Here, $\mathcal{H}_\ell^d$ is the space of the restriction of (real) homogeneous harmonic polynomials of degree $\ell$ on $\mathbb{S}^{d-1}$. That is,

$$g_\ell(\mathbf{x}) = \sum_{p=1}^{N(d,\ell)} \hat{g}_{\ell,p} Y_{\ell,p}, \qquad \hat{g}_{\ell,p} = \int_{\mathbb{S}^{d-1}} g(\mathbf{x}) Y_{\ell,p}(\mathbf{x}) d\mathbf{x}.$$

where $N(d, \ell)$ is given in section 2 and $Y_{\ell,p}$ are the spherical harmonic function of degree $\ell$ and order $p$. As a consequence of Theorem 1, we can consider the NN training error with the squared $L^2$-based loss function.

*Proof of Theorem 2.* By Theorem 1, for every $k = 0, \ldots, T$, we can write

$$\|\mathbf{u}(k) - \mathbf{y}\|_{\mathbf{c}} = \left\|(\mathbf{I} - 2\eta\mathbf{H}^\infty\mathbf{D_c})^k\mathbf{y}\right\|_{\mathbf{c}} + \varepsilon_3(k), \qquad |\varepsilon_3(k)| \leq \|\boldsymbol{\epsilon}(k)\|_{\mathbf{c}} \leq \epsilon. \qquad (54)$$

To estimate the first term, we first note that the matrix $\mathbf{I} - 2\eta\mathbf{H}^\infty\mathbf{D_c}$ is positive semidefinite in $(\mathbb{R}^n, \langle \cdot, \cdot \rangle_{\mathbf{c}})$ and $\|\mathbf{I} - 2\eta\mathbf{H}^\infty\mathbf{D_c}\|_{\mathbf{c}} \leq 1 - 2\eta\lambda_0$ (see Theorem 1). Since $g(\mathbf{x}) = \sum_{\ell=0}^L g_\ell(\mathbf{x})$, we have

$$(\mathbf{I} - 2\eta\mathbf{H}^\infty\mathbf{D_c})^k\mathbf{y} = \sum_{\ell=0}^L (\mathbf{I} - 2\eta\mathbf{H}^\infty\mathbf{D_c})^k\mathbf{y}^\ell, \qquad \mathbf{y} = \sum_{\ell=0}^L \mathbf{y}^\ell,$$

where $\mathbf{y}^\ell = [g_\ell(\mathbf{x}_1), \ldots, g_\ell(\mathbf{x}_n)]^\top \in \mathbb{R}^n$. By the Funk–Hecke formula and the quadrature rule, we have

$$\left(\mathbf{H}^\infty\mathbf{D_c}\mathbf{y}^\ell\right)_p = \sum_{i=1}^n c_i K^\infty(\mathbf{x}_p, \mathbf{x}_i)g_\ell(\mathbf{x}_i) = \int_{\mathbb{S}^{d-1}} K^\infty(\mathbf{x}_p, \boldsymbol{\xi})g_\ell(\boldsymbol{\xi})d\boldsymbol{\xi} + e_{p,\ell}^d = \mu_\ell g_\ell(\mathbf{x}_p) + e_{p,\ell}^d,$$

where $e_{p,\ell}^d$ is a quadrature error (see eq. (16)). Therefore, in vectorized form, we have $\mathbf{H}^\infty\mathbf{D_c}\mathbf{y}^\ell = \mu_\ell\mathbf{y}^\ell + \mathbf{e}_\ell^d$ or, equivalently, $(\mathbf{I} - 2\eta\mathbf{H}^\infty\mathbf{D_c})\mathbf{y}^\ell = (1 - 2\eta\mu_\ell)\mathbf{y}^\ell - 2\eta\mathbf{e}_\ell^d$, where $\mathbf{e}_\ell^d = (e_{1,\ell}^d, \ldots, e_{n,\ell}^d)^\top$. By applying $\mathbf{I} - 2\eta\mathbf{H}^\infty\mathbf{D_c}$ to $\mathbf{y}^\ell$ for $k$ times, we find that

$$(\mathbf{I} - 2\eta\mathbf{H}^\infty\mathbf{D_c})^k\mathbf{y}^\ell = (1 - 2\eta\mu_\ell)^k\mathbf{y}^\ell - \underbrace{2\eta\sum_{t=0}^{k-1}(1 - 2\eta\mu_\ell)^t (\mathbf{I} - 2\eta\mathbf{H}^\infty\mathbf{D_c})^{k-t-1}\mathbf{e}_\ell^d}_{\boldsymbol{\varepsilon}_2^\ell}.$$

The second term, $\boldsymbol{\varepsilon}_2^\ell$, can be easily bounded from above to obtain

$$\left\|\boldsymbol{\varepsilon}_2^\ell\right\|_{\mathbf{c}} \leq 2\eta\left\|\mathbf{e}_\ell^d\right\|_{\mathbf{c}}\sum_{t=0}^\infty (1 - 2\eta\mu_\ell)^t = \frac{1}{\mu_\ell}\left\|\mathbf{e}_\ell^d\right\|_{\mathbf{c}}.$$

The inequality above shows that $(\mathbf{I} - 2\eta\mathbf{H}^\infty\mathbf{D_c})^k\mathbf{y}^\ell$ is close to $(1 - 2\eta\mu_\ell)^k\mathbf{y}^\ell$. We define

$$\varepsilon_2 = \left\|\sum_{\ell=0}^L (\mathbf{I} - 2\eta\mathbf{H}^\infty\mathbf{D_c})^k\mathbf{y}^\ell\right\|_{\mathbf{c}} - \left\|\sum_{\ell=0}^L (1 - 2\eta\mu_\ell)^k\mathbf{y}^\ell\right\|_{\mathbf{c}}$$

to be the quantity in the statement of Theorem 2, which measures the accuracy of approximating the eigenvalues and eigenvectors of $\mathbf{H}^\infty\mathbf{D_c}$ using the eigenvalues and eigenfunctions of the continuous kernel $K^\infty$. Hence, we have

$$\left\|(\mathbf{I} - 2\eta\mathbf{H}^\infty\mathbf{D_c})^k\mathbf{y}\right\|_{\mathbf{c}} = \left\|\sum_{\ell=0}^L (1 - 2\eta\mu_\ell)^k\mathbf{y}^\ell\right\|_{\mathbf{c}} + \varepsilon_2. \qquad (55)$$

Using the triangle inequality, we have

$$|\varepsilon_2| \leq \sum_{\ell=0}^L \left\|\boldsymbol{\varepsilon}_2^\ell\right\|_{\mathbf{c}} \leq \sum_{\ell=0}^L \frac{1}{\mu_\ell}\left\|\mathbf{e}_\ell^d\right\|_{\mathbf{c}} = \sum_{\ell=0}^L \frac{1}{\mu_\ell}\sqrt{\sum_{i=1}^n c_i(e_{i,\ell}^d)^2} \leq \sum_{\ell=0}^L \frac{\sqrt{A_d}}{\mu_\ell}\max_{1 \leq i \leq n}\left|e_{i,\ell}^d\right|.$$

Recall that $\sum_{i=1}^{n} c_i = A_d$, the surface area of $\mathbb{S}^{d-1}$. Next, we can write

$$\left\| \sum_{\ell=0}^{L} (1-2\eta\mu_\ell)^k \, \mathbf{y}^\ell \right\|_{\mathbf{c}}^2 = \left( \sum_{\ell=0}^{L} (1-2\eta\mu_\ell)^k \, \mathbf{y}^\ell \right)^\top \mathbf{D_c} \left( \sum_{\ell=0}^{L} (1-2\eta\mu_\ell)^k \, \mathbf{y}^\ell \right)$$

$$= \sum_{j=0}^{L} \sum_{\ell=0}^{L} (1-2\eta\mu_j)^k (1-2\eta\mu_\ell)^k \, \mathbf{y}^{j^\top} \mathbf{D_c} \mathbf{y}^\ell$$

$$= \sum_{j=0}^{L} \sum_{\ell=0}^{L} (1-2\eta\mu_j)^k (1-2\eta\mu_\ell)^k \left( \int_{\mathbb{S}^{d-1}} g_j(\boldsymbol{\xi}) g_\ell(\boldsymbol{\xi}) d\boldsymbol{\xi} + e_{j,\ell}^c \right)$$

$$= \sum_{\ell=0}^{L} (1-2\eta\mu_\ell)^{2k} \|g_\ell\|_{L^2}^2 + \underbrace{\sum_{j=0}^{L} \sum_{\ell=0}^{L} (1-2\eta\mu_j)^k (1-2\eta\mu_\ell)^k e_{j,\ell}^c}_{\varepsilon_1(k)}, \quad (56)$$

where we used the fact that $g_j$ and $g_\ell$ are orthogonal in $L^2(\mathbb{S}^{d-1})$ for $j \neq \ell$. Combining eq. (55) and eq. (56), we can write

$$\left\| (\mathbf{I} - 2\eta \mathbf{H}^\infty \mathbf{D_c})^k \mathbf{y} \right\|_{\mathbf{c}} = \sqrt{ \sum_{j=0}^{L} (1-2\eta\mu_j)^{2k} \|g_j\|_{L^2}^2 + \varepsilon_1(k) + \varepsilon_2 }, \quad (57)$$

where

$$|\varepsilon_1(k)| \leq \left| \sum_{j=0}^{L} \sum_{\ell=0}^{L} (1-2\eta\mu_j)^k (1-2\eta\mu_\ell)^k e_{j,\ell}^c \right|.$$

The result follows from eq. (54) and eq. (57). $\qquad \square$

Similarly, suppose we use Theorem 1′, we can write down Theorem 2 in a form such that $\epsilon_3(k)$ depends on the parameters of the NN.

**Theorem 2′.** *Under the same setup and assumptions of Theorem 1′, let $\mathbf{P} = \mathbf{D_c}$ and $M_{\mathbf{P}} = \sqrt{c_{max}}$. If $g : \mathbb{S}^{d-1} \to \mathbb{R}$ is a bandlimited function with bandlimit $L$ and $1 - 2\eta\mu_\ell > 0$ for all $0 \leq \ell \leq L$ (see eq. (12)), then with probability $\geq 1 - \delta$ we have*

$$\|\mathbf{y} - \mathbf{u}(k)\|_{\mathbf{c}} = \sqrt{ \sum_{\ell=0}^{L} (1-2\eta\mu_\ell)^{2k} \|g_\ell\|_{L^2}^2 + \varepsilon_1(k) + \varepsilon_2 + \varepsilon_3(k) }, \quad 0 \leq k \leq T, \quad (58)$$

*where $|\varepsilon_3(k)|$ is bounded by eq. (51), and $\varepsilon_1(k)$ and $\varepsilon_2$ satisfy*

$$|\varepsilon_1(k)| \leq \left| \sum_{j=0}^{L} \sum_{\ell=0}^{L} (1-2\eta\mu_j)^k (1-2\eta\mu_\ell)^k e_{j,\ell}^c \right|, \qquad |\varepsilon_2| \leq \sum_{\ell=0}^{L} \frac{\sqrt{A_d}}{\mu_\ell} \max_{1 \leq i \leq n} \left| e_{i,\ell}^d \right|.$$

We remark that the left-hand side of eq. (17) can be rewritten as

$$\|\mathbf{y} - \mathbf{u}(k)\|_{\mathbf{c}} = \sqrt{ \sum_{i=1}^{n} c_i (y_i - u_i(k))^2 } = \sqrt{ \sum_{i=1}^{n} c_i (g(\mathbf{x}_i) - \mathcal{N}_k(\mathbf{x}_i))^2 }$$

$$= \sqrt{ \|g - \mathcal{N}_k\|_{L^2}^2 - E_{\mathbf{c}}((g - \mathcal{N}_k)^2) },$$

where $\mathcal{N}_k$ is the neural network at the $k$th iteration. Hence, eq. (17) can also be written as

$$\|g - \mathcal{N}_k\|_{L^2} = \sqrt{ \sum_{\ell=0}^{L} (1-2\eta\mu_\ell)^{2k} \|g_\ell\|_{L^2}^2 + \varepsilon_1(k) + \varepsilon_2 + \varepsilon_3(k) + \varepsilon_4(k) }, \quad (59)$$

where $\varepsilon_1, \varepsilon_2, \varepsilon_3$ are as in Theorem 2, and $|\varepsilon_4(k)| \leq \sqrt{|E_{\mathbf{c}}((g - \mathcal{N}_k)^2)|}$. Moreover, we assumed $g$ is bandlimited, and as we see in Proposition 2, the spherical harmonic coefficients of $\mathcal{N}_k$ decay fast as the frequency $\ell \to \infty$. This shows that the size of $|\varepsilon_4(k)|$ can also be controlled by $\gamma_{n,\ell}$ in eq. (18). Therefore, we derived a frequency bias statement of the generalization error with respect to the uniform distribution on $\mathbb{S}^{d-1}$.

### D.2 Frequency bias up to an approximation error

Theorem 2 shows that we theoretically have frequency bias up to the level of quadrature errors $\varepsilon_1(k)$ and $\varepsilon_2$. If the quadrature errors are large, then we may not observe frequency bias in practice. Here, we show that the quadrature errors can be made arbitrarily small by taking enough samples in the training data. Recall that our quadrature rule satisfies eq. (18).

Next, we pass the quadrature error to the approximation error, which may not necessarily be tight. However, this allows us to use the existing theory of spherical harmonics approximation to show the decay of quadrature errors.

**Lemma 5.** Let $f : \mathbb{S}^{d-1} \to \mathbb{R}$ be a function and $\text{dist}(f, \Pi_\ell^d) = \min_{h \in \Pi_\ell^d} \|f - h\|_{L^\infty}$. We have

$$\left| \int_{\mathbb{S}^{d-1}} f(\boldsymbol{\xi}) d\boldsymbol{\xi} - \sum_{i=1}^n c_i f(\mathbf{x}_i) \right| \leq 2\gamma_{n,\ell} \|f\|_{L^\infty} + 2A_d \text{dist}(f, \Pi_\ell^d) \tag{60}$$

for any integer $\ell \geq 0$.

*Proof.* Let $h_\ell = \arg\min_{h \in \Pi_\ell^d} \|f - h\|_{L^\infty}$. By the triangle inequality, we have

$$\left| \int_{\mathbb{S}^{d-1}} f(\boldsymbol{\xi}) d\boldsymbol{\xi} - \sum_{i=1}^n c_i f(\mathbf{x}_i) \right|$$

$$\leq \left| \int_{\mathbb{S}^{d-1}} f(\boldsymbol{\xi}) d\boldsymbol{\xi} - \int_{\mathbb{S}^{d-1}} h_\ell(\boldsymbol{\xi}) d\boldsymbol{\xi} \right| + \left| \int_{\mathbb{S}^{d-1}} h_\ell(\boldsymbol{\xi}) d\boldsymbol{\xi} - \sum_{i=1}^n c_i h_\ell(\mathbf{x}_i) \right| + \left| \sum_{i=1}^n c_i \left( h_\ell(\mathbf{x}_i) - f(\mathbf{x}_i) \right) \right|$$

$$\leq A_d \text{dist}(f, \Pi_\ell^d) + \gamma_{n,\ell} \|h_\ell\|_{L^\infty} + A_d \text{dist}(f, \Pi_\ell^d) \leq 2\gamma_{n,\ell} \|f\|_{L^\infty} + 2A_d \text{dist}(f, \Pi_\ell^d),$$

where we used the fact that $c_i > 0$ for $i = 1, \ldots, n$ and the fact that $\|h_\ell\|_{L^\infty} \leq 2\|f\|_{L^\infty}$. $\qquad\square$

Next, we focus on controlling the minimum approximation error $\text{dist}(f, \Pi_\ell^d)$ on the right-hand side of eq. (60). To do so, we prove the following lemma.

**Lemma 6.** Let $f_{ij}(\boldsymbol{\xi}) = K^\infty(\mathbf{x}_i, \boldsymbol{\xi}) g_j(\boldsymbol{\xi})$ and $g_{jp}(\boldsymbol{\xi}) = g_j(\boldsymbol{\xi}) g_p(\boldsymbol{\xi})$. Then, for a constant $C > 0$ that only depends on $d$, we have

$$\text{dist}(f_{ij}, \Pi_\ell^d) \leq C \frac{j+1}{\ell} \|g_j\|_{L^\infty}, \qquad 1 \leq i \leq n, 0 \leq j \leq L,$$

$$\text{dist}(g_{jp}, \Pi_\ell^d) \leq C \frac{j+p}{\ell} \|g_j\|_{L^\infty} \|g_p\|_{L^\infty}, \qquad 0 \leq j, p \leq L,$$

for all $\ell \geq 1$.

*Proof.* For $1 \leq a < b \leq d$ where $a, b \in \mathbb{N}$, and $t \in [-\pi, \pi)$, let $Q_{a,b,t}$ denote the action on $\mathbb{S}^{d-1}$ of rotation by the angle $t$ in the $(x_a, x_b)$-plane. For an integer $\alpha \geq 1$, we define the operator on functions on $\mathbb{S}^{d-1}$ by

$$\Delta_{a,b,t}^\alpha = (I - T(Q_{a,b,t}))^\alpha,$$

where $T(Q)f(\mathbf{x}) = f(Q^{-1}\mathbf{x})$. If $f \in C(\mathbb{S}^{d-1})$, then we define for $t > 0$ that

$$\omega_\alpha(f; t) = \max_{1 \leq a < b \leq d} \sup_{|\theta| \leq t} \left\| \Delta_{a,b,\theta}^\alpha f \right\|_{L^\infty}.$$

By (Dai & Xu, 2013, Thm. 4.4.2), we have

$$\text{dist}(f, \Pi_\ell^d) \leq c_1 \omega_\alpha(f; \ell^{-1}), \qquad \ell \geq 1, \alpha \geq 1,$$

where $c_1 > 0$ is some constant that only depends on $\alpha$. Then it is sufficient to bound $\omega_\alpha(f_{ij}; \ell^{-1})$ and $\omega_\alpha(g_{jp}; \ell^{-1})$ to finish the proof.

First, we aim to bound the term $\text{dist}(f_{ij}, \Pi_\ell^d)$ where $f_{ij}(\boldsymbol{\xi}) = K^\infty(\mathbf{x}_i, \boldsymbol{\xi}) g_j(\boldsymbol{\xi})$. We fix $1 \leq i \leq n$, $1 \leq a < b \leq d$ where $i, a, b \in \mathbb{N}$, and choose $\theta$ such that $|\theta| \leq \ell^{-1}$. We have

$$\left\| \Delta_{a,b,\theta}^1 f_{ij} \right\|_{L^\infty} = \left\| f_{ij}(\boldsymbol{\xi}) - f_{ij}(Q_{a,b,\theta}^{-1}\boldsymbol{\xi}) \right\|_{L^\infty}.$$

We then define
$$\delta K_i^\infty(\boldsymbol{\xi}) := K^\infty(\mathbf{x}_i, Q_{a,b,\theta}^{-1}\boldsymbol{\xi}) - K^\infty(\mathbf{x}_i, \boldsymbol{\xi}), \qquad \delta g_j(\boldsymbol{\xi}) := g_j(Q_{a,b,\theta}^{-1}\boldsymbol{\xi}) - g_j(\boldsymbol{\xi}).$$

We then have
$$\left\| f_{ij}(\boldsymbol{\xi}) - f_{ij}(Q_{a,b,\theta}^{-1}\boldsymbol{\xi}) \right\|_{L^\infty} = \| K^\infty(\mathbf{x}_i,\boldsymbol{\xi})g_j(\boldsymbol{\xi}) - (K^\infty(\mathbf{x}_i,\boldsymbol{\xi}) + \delta K_i^\infty(\boldsymbol{\xi}))(g_j(\boldsymbol{\xi}) + \delta g_j(\boldsymbol{\xi})) \|_{L^\infty}$$
$$\leq \| \delta K_i^\infty(\boldsymbol{\xi})(g_j(\boldsymbol{\xi}) + \delta g_j(\boldsymbol{\xi})) \|_{L^\infty} + \| K^\infty(\mathbf{x}_i,\boldsymbol{\xi})\delta g_j(\boldsymbol{\xi}) \|_{L^\infty}. \tag{61}$$

We control the two terms separately. First, to control the second term, we write
$$\| K^\infty(\mathbf{x}_i,\boldsymbol{\xi})\delta g_j(\boldsymbol{\xi}) \|_{L^\infty} \leq \| K^\infty(\mathbf{x}_i,\boldsymbol{\xi}) \|_{L^\infty} \| \delta g_j(\boldsymbol{\xi}) \|_{L^\infty} \leq \frac{1}{2} \| \delta g_j(\boldsymbol{\xi}) \|_{L^\infty}, \tag{62}$$

based on the definition of $K^\infty$ in eq. (11). For some constant $c_2 > 0$ that depends only on $d$, we have
$$\| \delta g_j(\boldsymbol{\xi}) \|_{L^\infty} = \left\| \Delta_{a,b,\theta}^1 g_j(\boldsymbol{\xi}) \right\|_{L^\infty} \leq c_2 \ell^{-1} \| D_{a,b} g_j(\boldsymbol{\xi}) \|_{L^\infty} \leq c_2 \frac{j}{\ell} \| g_j \|_{L^\infty}, \tag{63}$$

where $D_{a,b} := x_a \partial_b - x_b \partial_a$ and the two inequalities follow from (Dai & Xu, 2013, Lem. 4.2.2 (iii)) and (Dai & Xu, 2013, Lem. 4.2.4), respectively. Next, we control the first term of eq. (61) by writing
$$\| \delta K_i^\infty(\boldsymbol{\xi})(g_j(\boldsymbol{\xi}) + \delta g_j(\boldsymbol{\xi})) \|_{L^\infty} \leq \| \delta K_i^\infty(\boldsymbol{\xi}) \|_{L^\infty} \left\| g_j(Q_{a,b,\theta}^{-1}\boldsymbol{\xi}) \right\|_{L^\infty} = \| \delta K_i^\infty(\boldsymbol{\xi}) \|_{L^\infty} \| g_j \|_{L^\infty}. \tag{64}$$

We fix $\boldsymbol{\xi}$ and define $\boldsymbol{\xi}' = Q_{a,b,\theta}^{-1}\boldsymbol{\xi}$. It follows that
$$\delta K_i^\infty(\boldsymbol{\xi}) = K^\infty(\mathbf{x}_i, \boldsymbol{\xi}') - K^\infty(\mathbf{x}_i, \boldsymbol{\xi})$$
$$= \frac{1}{4\pi} \left[ (\mathbf{x}_i^\top \boldsymbol{\xi}' + 1)(\pi - \arccos(\mathbf{x}_i^\top \boldsymbol{\xi}')) - (\mathbf{x}_i^\top \boldsymbol{\xi} + 1)(\pi - \arccos(\mathbf{x}_i^\top \boldsymbol{\xi})) \right]$$
$$= \frac{1}{4\pi} \left[ \mathbf{x}_i^\top(\boldsymbol{\xi}' - \boldsymbol{\xi})(\pi - \arccos(\mathbf{x}_i^\top \boldsymbol{\xi})) - (\mathbf{x}_i^\top \boldsymbol{\xi}' + 1)(\arccos(\mathbf{x}_i^\top \boldsymbol{\xi}) - \arccos(\mathbf{x}_i^\top \boldsymbol{\xi}')) \right]$$

Next, by the triangle inequality for angles, we have
$$\left| \arccos(\mathbf{x}_i^\top \boldsymbol{\xi}) - \arccos(\mathbf{x}_i^\top \boldsymbol{\xi}') \right| \leq |\theta| \leq \ell^{-1}.$$

Since $\arccos$ is monotone and $|(d/dt)\arccos(t)| > 1$ for all $t$, by the fundamental theorem of calculus, we must have
$$\left| \mathbf{x}_i^\top \boldsymbol{\xi}' - \mathbf{x}_i^\top \boldsymbol{\xi} \right| \leq \left| \arccos(\mathbf{x}_i^\top \boldsymbol{\xi}) - \arccos(\mathbf{x}_i^\top \boldsymbol{\xi}') \right| \leq \ell^{-1}.$$

As a result, we have
$$|\delta K_i^\infty(\boldsymbol{\xi})| \leq \frac{1}{4\pi} \left[ \ell^{-1}\pi + 2\ell^{-1} \right] = \frac{\pi + 2}{4\pi} \ell^{-1}. \tag{65}$$

By eq. (64) and eq. (65), we find that
$$\| \delta K_i^\infty(\boldsymbol{\xi})(g_j(\boldsymbol{\xi}) + \delta g_j(\boldsymbol{\xi})) \|_{L^\infty} \leq \frac{\pi + 2}{4\pi} \ell^{-1} \| g_j \|_{L^\infty}. \tag{66}$$

Putting eqs. (61) to (63) and (66) together, we obtain
$$\left\| \Delta_{a,b,\theta}^1 f_{ij} \right\|_{L^\infty} = \left\| f_{ij}(\boldsymbol{\xi}) - f_{ij}(Q_{a,b,\theta}^{-1}\boldsymbol{\xi}) \right\|_{L^\infty} \leq \left( c_2 j + \frac{\pi + 2}{4\pi} \right) \ell^{-1} \| g_j \|_{L^\infty}.$$

Since $a$, $b$, and $\theta$ are chosen arbitrarily, this bound also holds for $\omega_1(f_{ij}; \ell^{-1})$.

Second, we aim to bound the term $\text{dist}(g_{jp}, \Pi_\ell^d)$ where $g_{jp}(\boldsymbol{\xi}) = g_j(\boldsymbol{\xi})g_p(\boldsymbol{\xi})$. As before, we fix the indices $a, b \in \mathbb{N}$ where $1 \leq a < b \leq d$ and $\theta$ such that $|\theta| \leq \ell^{-1}$. Similarly, we define
$$\delta g_j(\boldsymbol{\xi}) = g_j(Q_{a,b,\theta}^{-1}\boldsymbol{\xi}) - g_j(\boldsymbol{\xi}), \quad \delta g_p(\boldsymbol{\xi}) = g_p(Q_{a,b,\theta}^{-1}\boldsymbol{\xi}) - g_p(\boldsymbol{\xi}).$$

By eq. (63), we have
$$\left\| g_j g_p(\boldsymbol{\xi}) - g_j g_p(Q_{a,b,\theta}^{-1}\boldsymbol{\xi}) \right\|_{L^\infty} = \| g_j(\boldsymbol{\xi})g_p(\boldsymbol{\xi}) - (g_j(\boldsymbol{\xi}) + \delta g_j(\boldsymbol{\xi}))(g_p(\boldsymbol{\xi}) + \delta g_p(\boldsymbol{\xi})) \|_{L^\infty}$$
$$\leq \| \delta g_j(\boldsymbol{\xi})(g_p(\boldsymbol{\xi}) + \delta g_p(\boldsymbol{\xi})) \|_{L^\infty} + \| g_j(\boldsymbol{\xi})\delta g_p(\boldsymbol{\xi}) \|_{L^\infty} \tag{67}$$
$$\leq c_2 \frac{j + p}{\ell} \| g_j \|_{L^\infty} \| g_p \|_{L^\infty}.$$

Since $a$, $b$, and $\theta$ are arbitrary numbers, this bound also holds for $\omega_1(g_{jp}; \ell^{-1})$. $\qquad\square$

Using these lemmas, we can prove that Theorem 3 asymptotically controls the quadrature errors.

*Proof of Theorem 3.* First, we control $\varepsilon_1(k)$ as

$$
|\varepsilon_1(k)| \leq \left| \sum_{j=0}^{L} \sum_{p=0}^{L} (1-2\eta\mu_j)^k (1-2\eta\mu_p)^k e_{jp}^c \right| \leq \left| \sum_{j=0}^{L} \sum_{p=0}^{L} e_{jp}^c \right|
$$

$$
\leq \sum_{j=0}^{L} \sum_{p=0}^{L} \left| 2\gamma_{n,\ell} \|g_{jp}\|_{L^\infty} + 2A_d \mathrm{dist}(g_{jp}, \Pi_\ell^d) \right|
$$

$$
\leq \sum_{j=0}^{L} \sum_{p=0}^{L} \left( 2\gamma_{n,\ell} \|g_j\|_{L^\infty} \|g_p\|_{L^\infty} + 2A_d C \frac{j+p}{\ell} \|g_j\|_{L^\infty} \|g_p\|_{L^\infty} \right),
$$

where the third and the forth inequalities follow from Lemma 5 and Lemma 6, respectively. The final upper bound for $|\varepsilon_1(k)|$ holds since $\sum_{j=0}^{L} \sum_{p=0}^{L} (j+p) = \mathcal{O}(L^3)$. Next, there exists $C_2 > 0$ such that we can control $\varepsilon_2$ by writing

$$
|\varepsilon_2| \leq \sum_{j=0}^{L} \frac{\sqrt{A_d}}{\mu_j} \max_{1 \leq i \leq n} |e_{ij}^b| \leq \sum_{j=0}^{L} \frac{\sqrt{A_d}}{\mu_j} \left( 2\gamma_{n,\ell} \|g_j\|_{L^\infty} + 2A_d C \frac{j+1}{\ell} \|g_j\|_{L^\infty} \right)
$$

$$
\leq C_2 \left( \sum_{j=0}^{L} \mu_j^{-1} \right) \left( \frac{L^2}{\ell} + L\gamma_{n,\ell} \right) \max_j \|g_j\|_{L^\infty},
$$

where the second and the third inequalities follow from Lemma 5 and Lemma 6, respectively. The proof is complete. □

*Proof of Corollary 1.* By Theorem 3, it suffices to show that $\max_{0 \leq j \leq L} \|g_j\|_{L^\infty} \leq C \|g\|_{L^2}$ for some constant $C > 0$ that does not depend on $g$. Since $\mathcal{H}_i^d \perp \mathcal{H}_j^d$ in $L^2$ for $i \neq j$, we have $\|g_j\|_{L^2} \leq \|g\|_{L^2}$ for $0 \leq j \leq L$. The claim follows from the fact that $\|\cdot\|_{L^2}$ and $\|\cdot\|_{L^\infty}$ are equivalent in $\Pi_L^d$. □

# E  THE THEORY OF FREQUENCY BIAS WITH A $\mathbf{H^s}$-BASED LOSS FUNCTION

This section presents detailed proofs for Proposition 2 and Theorem 4 in Section 5, which concerns the frequency bias behavior of NN training using the $H^s$ loss function.

## E.1  A 2-LAYER RELU NEURAL NETWORK IS IN $\mathbf{H^s}(\mathbb{S}^{d-1})$ FOR $\mathbf{s < 3/2}$

We prove that a 2-layer ReLU NN map is contained in $H^s(\mathbb{S}^{d-1})$ for any $s < 3/2$ (see Proposition 2).

*Proof of Proposition 2.* Since $\mathcal{N}$ can be written as

$$
\mathcal{N}(\mathbf{x}) = \sum_{r=1}^{m} a_r \mathrm{ReLU}(\mathbf{w}_r^\top \mathbf{x} + b_r),
$$

it suffices to prove that $f(\mathbf{x}) = \mathrm{ReLU}(\mathbf{w}^\top \mathbf{x} + b)$ is in $H^s$ for all $\mathbf{w} \in \mathbb{R}^d, b \in \mathbb{R}$. Since $\mathrm{ReLU}(a\mathbf{x}) = a\mathrm{ReLU}(\mathbf{x})$ for any $a > 0$, we can assume that $\|\mathbf{w}\|_2 = 1$. Moreover, since the Sobolev spaces are rotationally invariant, we can assume that $\mathbf{w} = (1, 0, \ldots, 0)^\top$. Then, $f$ can be written as

$$
f(\mathbf{x}) = \mathrm{ReLU}(x_1 + b).
$$

If $b \leq -1$ or $b \geq 1$, then $f(\mathbf{x})$ is a constant, and thus $f \in H^s(\mathbb{S}^{d-1})$ for all $s \in \mathbb{R}$. We assume $-1 < b < 1$. Then, we have

$$
f(\mathbf{x}) = \begin{cases} x_1 + b, & x_1 > -b, \\ 0, & x_1 \leq -b. \end{cases}
$$

We define the function

$$\mathcal{S}_s(f)(\mathbf{x}) = \int_0^\pi \frac{|\mathcal{I}_t f(\mathbf{x}) - f(\mathbf{x})|^2}{t^{2s+1}} dt,$$

where

$$\mathcal{I}_t f(\mathbf{x}) = \fint_{C(\mathbf{x},t)} f(\boldsymbol{\xi}) d\boldsymbol{\xi}, \qquad C(\mathbf{x},t) = \{\boldsymbol{\xi} \in \mathbb{S}^{d-1} \mid \arccos(\boldsymbol{\xi} \cdot \mathbf{x}) \leq t\}.$$

Here, $\fint_{C(\mathbf{x},t)} f(\boldsymbol{\xi}) d\boldsymbol{\xi} = |C(\mathbf{x},t)|^{-1} \int_{C(\mathbf{x},t)} f(\boldsymbol{\xi}) d\boldsymbol{\xi}$ is the averaged integral, where $|C(\mathbf{x},t)|$ is the Lebesgue measure of $C(\mathbf{x},t)$. Then, by (Barceló et al., 2020, Thm. 1.1), we have $f \in H^s(\mathbb{S}^{d-1})$ if and only if $\mathcal{S}_s(f)$ is integrable. We now show that $\mathcal{S}_s(f)$ is integrable on both $E_1 = \{\mathbf{x} \in \mathbb{S}^{d-1} \mid x_1 > -b\}$ and $E_2 = \{\mathbf{x} \in \mathbb{S}^{d-1} \mid x_1 < -b\}$ if $s < 3/2$.

First, we define the function

$$h(\mathbf{x}) = x_1 + b.$$

Then, $h \in H^s(\mathbb{S}^{d-1})$. If we can show that

$$\mathcal{S}_s(h - f)(\mathbf{x}) = \int_0^\pi \frac{|\mathcal{I}_t(h - f)(\mathbf{x}) - (h - f)(\mathbf{x})|^2}{t^{2s+1}} dt$$

is integrable on $E_1$, then we have

$$
\begin{aligned}
\int_{E_1} \mathcal{S}_s(f)(\mathbf{x}) d\mathbf{x} &= \int_{E_1} \int_0^\pi \frac{|\mathcal{I}_t f(\mathbf{x}) - f(\mathbf{x})|^2}{t^{2s+1}} dt d\mathbf{x} \\
&\leq 2 \int_{E_1} \int_0^\pi \frac{|\mathcal{I}_t h(\mathbf{x}) - h(\mathbf{x})|^2 + |\mathcal{I}_t(h - f)(\mathbf{x}) - (h - f)(\mathbf{x})|^2}{t^{2s+1}} dt d\mathbf{x} \quad (68) \\
&= 2 \int_{E_1} \mathcal{S}_s(h)(\mathbf{x}) d\mathbf{x} + 2 \int_{E_1} \mathcal{S}_s(h - f)(\mathbf{x}) d\mathbf{x} < \infty.
\end{aligned}
$$

Assume $\mathbf{x} \in E_1$, and let $\rho$ be the minimum angular distance between $\mathbf{x}$ and any point in the set $S = \{\boldsymbol{\xi} \in \mathbb{S}^{d-1} \mid \xi_1 = -b\}$, i.e., $\rho = \min_{\boldsymbol{\xi} \in S} \arccos(\boldsymbol{\xi} \cdot \mathbf{x})$. Then, for $0 < t \leq \rho$, we clearly have $\mathcal{I}_t(h - f)(\mathbf{x}) = 0 = (h - f)(\mathbf{x})$. Assume $\rho < t < \pi$. We divide $C(\mathbf{x},t)$ into two parts $C_1$ and $C_2$ up to a Lebesgue null set, where $C_i = C(\mathbf{x},t) \cap E_i$. Then, $h - f = 0$ on $C_1$. Next, by Li (2011), we know that the measure of $C_2$ satisfies[3]

$$\frac{|C_2|}{|C(\mathbf{x},t)|} = \Theta\left(I\left(\frac{t^2 - \rho^2}{t^2}; \frac{d}{2}, \frac{1}{2}\right)\right) = \Theta\left(B\left(\frac{t^2 - \rho^2}{t^2}; \frac{d}{2}, \frac{1}{2}\right)\right), \qquad t \to \rho^+,$$

where $I$ is the regularized incomplete beta function and $B$ is the incomplete beta function. Here and throughout the proof, the constants in the big-$\Theta$ notations are independent of $\rho$ or $t$, but possibly depend on $b$ and $d$, which are fixed in the proof. Moreover, we have the formula

$$B\left(\frac{t^2 - \rho^2}{t^2}; \frac{d}{2}, \frac{1}{2}\right) = \frac{[(t^2 - \rho^2)/t^2]^{d/2}}{d/2} F\left(\frac{d}{2}, \frac{1}{2}, \frac{d+2}{2}; \frac{t^2 - \rho^2}{t^2}\right),$$

where $F$ is the hypergeometric function that converges to 1 as $t \to \rho^+$ (Olver et al., 2010, sect. 8.17, sect. 15.2). Hence, we have

$$\frac{|C_2|}{|C(\mathbf{x},t)|} = \Theta\left(\frac{(t - \rho)^{d/2}}{t^{d/2}}\right), \qquad t \to \rho^+. \quad (69)$$

Now, by the way we defined $h - f$, there exist constants $R_1, R_2 > 0$ such that

$$|h - f|(\boldsymbol{\xi}) \leq R_1(t - \rho), \qquad \boldsymbol{\xi} \in C_2,$$

$$|h - f|(\boldsymbol{\xi}) \geq R_2(t - \rho), \qquad \boldsymbol{\xi} \in \left\{\boldsymbol{\zeta} \in C_2 \,\Big|\, \min_{\boldsymbol{\theta} \in S} \arccos(\boldsymbol{\zeta} \cdot \boldsymbol{\theta}) \geq \frac{t - \rho}{2}\right\}.$$

---

[3]For two functions $\alpha(t)$ and $\beta(t)$, we say $\alpha(t) = \Theta(\beta(t))$ as $t \to \rho^+$ if there exist positive constants $C_l, C_r$ and radius $r > 0$ such that $C_l \beta(t) \leq \alpha(t) \leq C_r \beta(t)$ for all $0 < t - \rho < r$.

This gives us

$$\left| \fint_{C(\mathbf{x},t)} (h-f)(\boldsymbol{\xi})d\boldsymbol{\xi} \right| = \Theta\left(\frac{(t-\rho)^{(d+2)/2}}{t^{d/2}}\right), \qquad t \to \rho^+. \tag{70}$$

Now, we have

$$\mathcal{S}_s(h-f)(\mathbf{x}) = \int_\rho^\pi t^{-2s-1}\mathcal{I}_t(h-f)(\mathbf{x})^2 dt = \int_\rho^\pi t^{-2s-1}\left(\fint_{C(\mathbf{x},t)} (h-f)(\boldsymbol{\xi})d\sigma(\boldsymbol{\xi})\right)^2 dt$$

$$= \Theta\left(\int_\rho^\pi t^{-2s-1-d}(t-\rho)^{d+2}dt\right) = \Theta(\rho^{2-2s}+1).$$

To integrate $\mathcal{S}_s(h-f)$ over $E_1$, we first change the coordinates and integrate over $\mathbb{S}^{d-2}$ by fixing a $\rho$. The resulting integral is still in $\Theta(\rho^{2-2s}+1)$. We then integrate over $\rho$ and the result follows from the fact that a function in $\Theta(\rho^{2-2s}+1)$ is integrable near $\rho = 0$ if and only if $s < 3/2$. This proves $\mathcal{S}_s(h-f)$ is integrable on $E_1$ if and only if $s < 3/2$. Note that if $\mathcal{S}_s(h-f)$ is not integrable on $E_1$, then $\mathcal{S}_s(f)$ is neither integrable. This proves the proposition when $s \geq 3/2$.

To see $\mathcal{S}_s(f)$ is integrable over $E_2$ when $s < 3/2$, we note that $f$ can be rewritten as $\tilde{f} - \tilde{h}$, where $\tilde{f}(\mathbf{x}) = \mathrm{ReLU}(-x_1-b)$ and $\tilde{h}(\mathbf{x}) = -x_1-b$. By the same argument, we have that $\mathcal{S}_s(f) = \mathcal{S}_s(\tilde{f}-\tilde{h})$ is integrable on $E_2$, which completes the proof. $\qquad\square$

### E.2 Frequency bias with a squared Sobolev norm as the loss function

In this section, we prove Theorem 4 on Sobolev training. Recall that we compute the $H^s$-based loss based on eq. (20).

*Proof of Theorem 4.* Fix some $0 \leq \ell \leq L$. We can write

$$\mathbf{H}^\infty \mathbf{P}_s \mathbf{y}^\ell = \sum_{j=0}^{\ell_{\max}} \sum_{p=1}^{N(d,j)} \mathbf{H}^\infty \omega_j \mathbf{a}_{j,p} \mathbf{a}_{j,p}^\top \mathbf{y}^\ell = \sum_{p=1}^{N(d,\ell)} \mathbf{H}^\infty \omega_\ell \mathbf{a}_{\ell,p} \hat{g}_{\ell,p} + \sum_{j=0}^{\ell_{\max}} \sum_{p=1}^{N(d,j)} \mathbf{H}^\infty \omega_j \mathbf{a}_{j,p} e_{\ell,j,p}^a, \tag{71}$$

where $\omega_\ell = (1+\ell)^{2s}$ and we used the fact that

$$\mathbf{a}_{j,p}^\top \mathbf{y}^\ell = \int_{\mathbb{S}^{d-1}} Y_{j,p}(\boldsymbol{\xi})g_\ell(\boldsymbol{\xi})d\boldsymbol{\xi} + e_{\ell,j,p}^a = \begin{cases} \hat{g}_{\ell,p} + e_{\ell,j,p}^a, & \text{if } \ell = j, \\ e_{\ell,j,p}^a, & \text{otherwise.} \end{cases}$$

Next, we have

$$(\mathbf{H}^\infty \mathbf{a}_{j,p})_i = \int_{\mathbb{S}^{d-1}} K^\infty(\mathbf{x}_i,\boldsymbol{\xi})Y_{j,p}(\boldsymbol{\xi})d\boldsymbol{\xi} + e_{i,j,p}^b = \mu_j Y_{j,p}(\mathbf{x}_i) + e_{i,j,p}^b,$$

where the last equality follows from the Funk–Hecke formula. Hence, the first term in eq. (71) can be written as

$$\left(\sum_{p=1}^{N(d,\ell)} \mathbf{H}^\infty \omega_\ell \mathbf{a}_{\ell,p} \hat{g}_{\ell,p}\right)_i = \sum_{p=1}^{N(d,\ell)} (\mu_j Y_{\ell,p}(\mathbf{x}_i) + e_{i,\ell,p}^b)\omega_\ell \hat{g}_{\ell,p} = \mu_j \omega_\ell g_j(\mathbf{x}_i) + \sum_{p=1}^{N(d,\ell)} e_{i,\ell,p}^b \omega_\ell \hat{g}_{\ell,p}.$$

Moreover, the second term in eq. (71) can be written as

$$\left(\sum_{j=0}^{\ell_{\max}} \sum_{p=1}^{N(d,j)} \mathbf{H}^\infty \omega_j \mathbf{a}_{j,p} e_{\ell,j,p}^a\right)_i = \sum_{j=0}^{\ell_{\max}} \sum_{p=1}^{N(d,j)} \omega_j e_{\ell,j,p}^a \left(\mu_j Y_{j,p}(\mathbf{x}_i) + e_{i,j,p}^b\right).$$

Therefore, we have

$$\mathbf{H}^\infty \mathbf{P}_s \mathbf{y}^\ell = \mu_\ell \omega_\ell \mathbf{y}^\ell + \omega_\ell \boldsymbol{\varepsilon}_1^\ell, \tag{72}$$

where

$$(\boldsymbol{\varepsilon}_1^\ell)_i = \sum_{p=1}^{N(d,\ell)} e_{i,\ell,p}^b \hat{g}_{\ell,p} + \sum_{j=0}^{\ell_{\max}} \sum_{p=1}^{N(d,j)} \frac{\omega_j}{\omega_\ell} e_{\ell,j,p}^a \left(\mu_j Y_{j,p}(\mathbf{x}_i) + e_{i,j,p}^b\right). \tag{73}$$

Applying eq. (72) recursively, we have

$$(\mathbf{I}-2\eta\mathbf{H}^\infty\mathbf{P}_s)^k\mathbf{y} = \sum_{\ell=0}^{L}(1-2\eta\mu_\ell\omega_\ell)^k\,\mathbf{y}^\ell \underbrace{-2\eta\sum_{\ell=0}^{L}\sum_{t=0}^{k-1}(1-2\eta\mu_\ell\omega_\ell)^t(\mathbf{I}-2\eta\mathbf{H}^\infty\mathbf{P}_s)^{k-t-1}\omega_\ell\boldsymbol{\varepsilon}_1^\ell}_{\boldsymbol{\varepsilon}_1}.$$

Now, since $\mathbf{H}^\infty\mathbf{P}_s$ is self-adjoint and positive definite in $(\mathbb{R}^n,\langle\cdot,\cdot\rangle_{\mathbf{P}_s})$, by the way we pick $\eta$, we guarantee that $\mathbf{I}-2\eta\mathbf{H}^\infty\mathbf{P}_s$ is positive definite in $(\mathbb{R}^n,\langle\cdot,\cdot\rangle_{\mathbf{P}_s})$ and hence

$$\|\mathbf{I}-2\eta\mathbf{H}^\infty\mathbf{P}_s\|_{\mathbf{P}_s}\leq 1-2\eta\lambda_0.$$

This gives us $\|\boldsymbol{\varepsilon}_1\|_{\mathbf{P}_s}\leq\sum_{\ell=0}^{L}\mu_\ell^{-1}\|\varepsilon_1^\ell\|_{\mathbf{P}_s}$, and the result follows from Theorem 1. $\qquad\square$

While Theorem 4 captures the frequency bias in squared $H^s$ loss training up to quadrature errors, analyzing the quadrature errors can be task-specific. Therefore, studying the quadrature rules could be a direction of future research.

Suppose we use Theorem 1$'$ in the proof, we can write down Theorem 4 in a form such that $\epsilon_2(k)$ depends on the parameters of the NN.

**Theorem 4$'$.** *Suppose $g\in\Pi_L^d$ and $\Phi_s$ is the loss function in eq. (20), where $\mathbf{P}_s$ is positive definite and $\ell_{max}\geq L$. Under the assumptions of Theorem 1$'$, if $1-2\eta\mu_\ell(1+\ell)^{2s}>0$ for all $0\leq\ell\leq L$, then with probability $\geq 1-\delta$ over the random initialization, we have*

$$\mathbf{y}-\mathbf{u}(k) = \sum_{\ell=0}^{L}\left(1-2\eta\mu_\ell(1+\ell)^{2s}\right)^k\mathbf{y}^\ell+\boldsymbol{\varepsilon}_1+\boldsymbol{\varepsilon}_2(k), \qquad 0\leq k\leq T, \tag{74}$$

*where $\|\boldsymbol{\varepsilon}_2(k)\|_{\mathbf{P}_s}$ is bounded by eq. (51) with $M_{\mathbf{P}}=M_{\mathbf{P}_s}$, and $\mathbf{y}^\ell=(g_\ell(\mathbf{x}_1),\ldots,g_\ell(\mathbf{x}_n))^\top$ and $\boldsymbol{\varepsilon}_1$ satisfies*

$$\|\boldsymbol{\varepsilon}_1\|_{\mathbf{P}_s}\leq\sum_{\ell=0}^{L}\mu_\ell^{-1}\|\varepsilon_1^\ell\|_{\mathbf{P}_s},\quad (\varepsilon_1^\ell)_i=e_{i,\ell}^d+\sum_{j=0}^{\ell_{max}}\frac{(1+j)^{2s}}{(1+\ell)^{2s}}\sum_{p=1}^{N(d,j)}e_{\ell,j,p}^a\left(\mu_j Y_{j,p}(\mathbf{x}_i)+e_{i,j,p}^b\right).$$

Moreover, we note that the remark after the proof of Theorem 2 applies here as well. In particular, using the relation that

$$\|\mathbf{y}-\mathbf{u}(k)\|_{\mathbf{c}}=\sqrt{\|g-\mathcal{N}_k\|_{L^2}^2-E_{\mathbf{c}}((g-\mathcal{N}_k)^2)},$$

we can rewrite eq. (21) into

$$\|g-\mathcal{N}_k\|_{L^2}=\left\|\sum_{\ell=0}^{L}\left(1-2\eta\mu_\ell(1+\ell)^{2s}\right)^k\mathbf{y}^\ell\right\|_{\mathbf{c}}+\varepsilon_1+\varepsilon_2(k)+\varepsilon_3(k), \tag{75}$$

where $|\varepsilon_1|\leq\|\boldsymbol{\varepsilon}_1\|_{\mathbf{c}}, |\varepsilon_2(k)|\leq\|\boldsymbol{\varepsilon}_2(k)\|_{\mathbf{c}}$ are as in Theorem 4, and $|\varepsilon_3(k)|\leq\sqrt{|E_{\mathbf{c}}(g-\mathcal{N}_k)^2|}$. By eq. (57), we can directly relate $\left\|\sum_{\ell=0}^{L}\left(1-2\eta\mu_\ell(1+\ell)^{2s}\right)^k\mathbf{y}^\ell\right\|_{\mathbf{c}}$ to $\sqrt{\sum_{\ell=0}^{L}(1-2\eta\mu_\ell(1+\ell)^{2s})^{2k}\|g_\ell\|_{L^2}^2}$ up to some quadrature errors. Therefore, the generalization error with respect to the uniform distribution can be written as

$$\|g-\mathcal{N}_k\|_{L^2}=\sqrt{\sum_{\ell=0}^{L}(1-2\eta\mu_j(1+\ell)^{2s})^{2k}\|g_\ell\|_{L^2}^2+E_{\mathrm{NTK}}+E_{\mathrm{quad}}},$$

where $E_{\mathrm{NTK}}$ is an error that can be made arbitrarily small by taking $\kappa$ small enough and $m$ large enough and $E_{\mathrm{quad}}$ involves only quadrature errors and can be made arbitrarily small if we make the quadrature rule accurate enough.

## F EXPERIMENTAL DETAILS

We now present the details of the three experiments in Section 6.

### F.1 LEARNING TRIGONOMETRIC POLYNOMIALS ON THE UNIT CIRCLE

In section 6.1, we train a NN with data derived from sampling a trigonometric polynomial at nonuniform points on the unit circle. The $n = 1140$ nonuniform data points $\{\mathbf{x}_i\}$ for this test are generated by taking the union of three sets of equally spaced points, as shown in Figure 1. The data set $\{(\cos(\theta_i), \sin(\theta_i))\}$ contains 100 equally spaced nodes sampled from $\theta \in (0, 2\pi]$, superimposed with 40 equally spaced nodes sampled from $\theta \in [0, 0.3\pi]$ and 1000 equally spaced nodes sampled from $\theta \in [1.4\pi, 1.8\pi]$ (see Figure 1, left). We construct the quadrature weights $c_j$ by minimizing $\sum_{j=1}^{n} c_j^2$, under the constraints that the $c_i$'s are positive and the quadrature rule is exact on $\Pi_{55}^2$, i.e., $E_{\mathbf{c}}(f) = 0$ for all $f \in \Pi_{55}^2$. Note that due to the linearity of the quadrature rule, it suffices to check a finite set of linear constraints

$$E_{\mathbf{c}}(Y_{\ell,p}) = 0, \qquad 0 \le \ell \le 55, 1 \le p \le N(2, \ell).$$

This computation method is proposed in (Mhaskar et al., 2000). Here, 55 is selected to be an integer close to the maximum degree $\ell$ such that there exists a positive quadrature rule exact on $\Pi_{\ell}^d$. In the upper-bounds of the quadrature errors in Theorem 3, there exist terms

$$\frac{L^3}{\ell} + L^2 \gamma_{n,\ell}, \qquad \frac{L^2}{\ell} + L \gamma_{n,\ell}, \tag{76}$$

where $L$ is a constant bandlimit of the target function. By requiring the quadrature rule to be exact on $\Pi_{\ell_0}^2$, we guarantee that $\gamma_{n,\ell_0} = 0$. Hence, by requiring $\ell_0$ to be large, we can heuristically make $\gamma_{n,\ell}$ smaller for a moderate $\ell$. This allows us to show that the upper-bounds in eq. (76) are small by balancing the term that involves $\ell^{-1}$, which vanishes as $\ell \to \infty$, and the term that involves $\gamma_{n,\ell}$, which increases as $\ell \to \infty$. This justifies why we compute the quadrature rule by requiring it to be exact on spherical harmonics of degree as large as possible.

The experiment consists of two parts. First, we compare the effects of training with the loss function $\Phi$ based on eq. (1) against the squared $L^2$ loss function $\widetilde{\Phi}$ in eq. (3), where

$$\Phi(\mathbf{W}) = \frac{A_d}{2n} \sum_{i=1}^{n} |\mathcal{N}(\mathbf{x}_i) - g(\mathbf{x}_i)|^2, \qquad \widetilde{\Phi}(\mathbf{W}) = \frac{1}{2} \sum_{i=1}^{n} c_i |\mathcal{N}(\mathbf{x}_i) - g(\mathbf{x}_i)|^2.$$

We define the target function to be

$$g(\mathbf{x}) = \tilde{g}(\theta) = \sum_{\ell=1}^{9} \sin(\ell\theta), \qquad \mathbf{x} = \mathbf{x}(\theta) = (\cos\theta, \sin\theta) \in \mathbb{S}^1,$$

where $\tilde{g}(\theta) = g(\mathbf{x}(\theta))$. We set up two 2-layer ReLU-activated NNs with $5 \times 10^4$ hidden neurons in each layer and train them using the same training data and gradient descent procedure, except with different loss functions $\Phi$ and $\widetilde{\Phi}$. In Figure 5, we showed the evolution of the NNs trained using $\Phi$ and $\widetilde{\Phi}$, respectively. While the NN trained with $\Phi$ approximates the function very well in the region where the training data are dense (i.e., where $\theta \in [1.4\pi, 1.8\pi] \approx [4.40, 5.65]$), the NN trained with $\widetilde{\Phi}$ provides a better overall approximation on the entire domain and demonstrates frequency bias much more clearly.

To evaluate the frequency loss, we collect 100 uniform samples from $\mathcal{N}(\mathbf{x})$ and $g(\mathbf{x})$ and compute the Fourier coefficients $\widehat{\mathcal{N}}(\ell)$ and $\hat{g}(\ell)$ such that functions

$$\mathcal{N}(\mathbf{x}) = \widetilde{\mathcal{N}}(\theta) \approx \sum_{\ell=0}^{30} \widehat{\mathcal{N}}(\ell) e^{i\ell\theta}, \qquad g(\mathbf{x}) = \tilde{g}(\theta) \approx \sum_{\ell=0}^{30} \hat{g}(\ell) e^{i\ell\theta},$$

where $\widetilde{\mathcal{N}}(\theta) = \mathcal{N}(\mathbf{x}(\theta))$. The frequency loss $|\widehat{\mathcal{N}}(\ell) - \hat{g}(\ell)|$ estimates how $g(\mathbf{x})$ is approximated by $\mathcal{N}(\mathbf{x})$ at the frequency $\ell$ when training with the different loss functions (see Figure 1, middle). In addition, we also train the NN to learn each individual frequency with the training data coming from $g_\ell = \sin(\ell\theta)$ and count the number of iterations it takes to obtain $\widetilde{\Phi}(\mathbf{W}) < 1.0 \times 10^{-3}$ (see Figure 1, right).

The second part of the experiment focuses on NN training with a discretized squared Sobolev norm as the loss function. More precisely, we fix some $s \in \mathbb{R}$ and consider the Sobolev loss function eq. (20). For $\mathbb{S}^1$, we have $N(d, \ell) = 2$ if $\ell > 0$. We take $Y_{\ell,1}(\mathbf{x}) = \sin(\ell\theta)/\sqrt{2\pi}$ and $Y_{\ell,2}(\mathbf{x}) = \cos(\ell\theta)/\sqrt{2\pi}$, where the $\sqrt{2\pi}$ factor is a normalization factor.

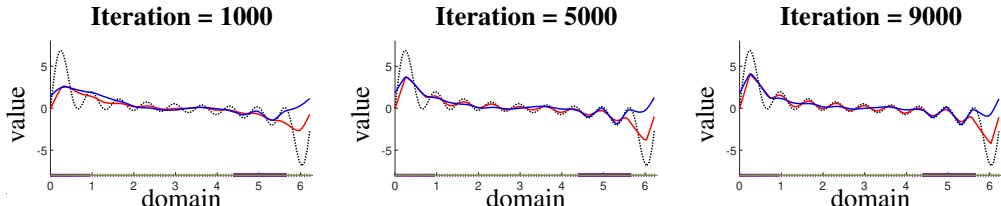

Figure 5: The target function (black, dotted), the NN trained with $\Phi$ (blue, solid), and the NN trained with $\tilde{\Phi}$ (red, solid). The purple bars on the horizontal axes show the positions of the training data.

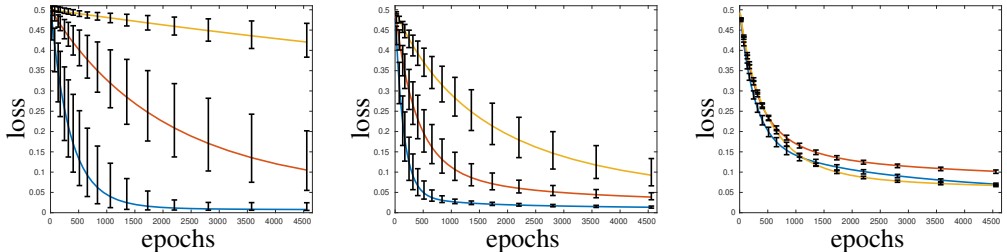

Figure 6: Frequency loss for $\ell = 3$ (blue), $\ell = 5$ (red), and $\ell = 9$ (yellow) based on the squared $H^s$ norm as the loss function. Left: $s = -1$; Middle: $s = 0$; Right: $s = 1$. The error bars are generated using the results obtained by executions with thirty different random seeds. In the left figure, the result of one of the thirty executions is omitted because the NN is trapped by a local minimizer that is not a global one, making the NN not converge to the target function.

We set $\ell_{\max} = 30$ in eq. (20) and learn $g$ with different $s$ values ranging from $-1$ to $4$. For each $s$, we compute the frequency loss $|\widehat{\mathcal{N}}(\ell) - \hat{g}(\ell)|$ after different numbers of epochs. As $s$ increases, the frequency loss for higher frequencies decays faster (see Figure 6). In particular, we see in Figure 2, the "rainbow" plot, that when $s = -1$, the lower-frequency losses are much smaller than the higher ones after 5000 iterations, while when $s = 3$ the higher frequencies are learned faster than the lower ones.

## F.2 LEARNING SPHERICAL HARMONICS ON THE UNIT SPHERE

In section 6.2, we train a NN with data derived from sampling a function defined on $\mathbb{S}^2$ at nonuniform points. The training data is a maximum determinant set of 2500 points that comes from the so-called "spherepts" dataset (Wright & Michaels, 2015). In this experiment, the target function is

$$g(\mathbf{x}) = \sum_{\ell=1}^{15} Y_{2\ell,0}(\mathbf{x}),$$

where $Y_{2\ell,0}$ is the normalized zonal spherical harmonic function of degree $2\ell$. Therefore, the spherical harmonic coefficients of $g$ defined in eq. (4) satisfy $\hat{g}_{\ell,p} = 1$ if $p = 0$ and $\ell = 2, 4, \ldots, 30$, and $\hat{g}_{\ell,p} = 0$ otherwise. We then train an NN with the squared $H^s$ norm as the loss function (see eq. (20)) for $s = -1, 0, 2.5$. By Theorem 4, we need $\ell_{\max} \geq L = 30$. We set $\ell_{\max} = 40$ in eq. (20), assuming that the bandwidth $L$ is not known a priori. We observe frequency bias in this experiment by considering $|\widehat{\mathcal{N}}_{\ell,p} - \hat{g}_{\ell,p}|$ after each epoch for $\ell = 4, 10, 20$. We confirm that low frequencies of $g$ are captured earlier in training than high frequencies when $s = -1$ and $s = 0$ (see Figure 3, left and middle), and that this frequency bias phenomena can be counterbalanced by taking $s = 2.5$ (see Figure 3, right).

## F.3 TEST ON AUTOENCODER

Autoencoders can be used as a generative model to randomly generate new data that is similar to the training data. In this experiment, we use Sobolev-based norms to improve NN training for producing new images of digits that match the MNIST dataset. In our final experiment, we use the same autoencoder architecture as in (Chollet, 2016), except we train the autoencoder with a different loss function (see section 6.3).

Here, for training images $\{\mathbf{x}_i\}$, a standard loss function can be

$$\Phi(\mathbf{W}) = \frac{1}{2} \int \text{dist}(\mathcal{N}(\mathbf{x}), \mathbf{x}) d\mu(\mathbf{x}) \approx \frac{1}{2n} \sum_{i=1}^{n} \text{dist}(\mathcal{N}(\mathbf{x}_i), \mathbf{x}_i),$$

where $\mu$ is the distribution of the training images and $\text{dist}(\mathcal{N}(\mathbf{x}_i), \mathbf{x}_i)$ is a distance between the output of the NN given by $\mathcal{N}(\mathbf{x})$ and the image $\mathbf{x}$. We select the distance metric to measure the difference between $\mathcal{N}(\mathbf{x})$ and $\mathbf{x}$ as

$$\Phi(\mathbf{W}) = \frac{1}{2n} \sum_{i=1}^{n} \|\mathcal{N}(\mathbf{x}_i) - \mathbf{x}_i\|_F^2, \tag{77}$$

where $\|\cdot\|_F$ denotes the matrix Frobenius norm. The distance metric in eq. (77) can be viewed as a discretization of the continuous $L^2$ norm. That is, if one imagines generating a continuous function $x : [0,1]^2 \to [0,\infty)$ that interpolates an image as well as a function that interpolates the NN, then

$$\frac{1}{N_{\text{pixel}}} \|\mathcal{N}(\mathbf{x}) - \mathbf{x}\|_F^2 \approx \iint |\mathcal{N}(x)(y_1, y_2) - x(y_1, y_2)|^2 dy_1 dy_2 = \|\mathcal{N}(x) - x\|_{L^2}^2,$$

where $N_{\text{pixel}}$ is the total number of pixels of the image $\mathbf{x}$. In this continuous viewpoint, the $H^s$ norm is given by (assuming that the continuous interpolating functions $x$ and $\mathcal{N}(x)$ are constructed with periodic boundary conditions)

$$\|\mathcal{N}(x) - x\|_{H^s}^2 = \int (1 + |\xi|^2)^s |\widehat{\mathcal{N}(x)}(\xi) - \hat{x}(\xi)|^2 d\xi \approx \|\mathbf{S}_s \circ (\mathbf{F}_l (\mathcal{N}(\mathbf{x}) - \mathbf{x}) \mathbf{F}_r^\top)\|_F^2, \tag{78}$$

where $\mathbf{F}_l, \mathbf{F}_r$ are the left and right 2D-DFT matrices, respectively, $(S_s)_{j\ell} = (1 + j^2 + \ell^2)^{s/2}$, and '$\circ$' is the Hadamard product. Hence, if we define $\text{vec}(\mathbf{A})$ to be the vector obtained by reshaping a matrix $\mathbf{A}$ using the column-major order, then we have

$$\|\mathbf{S}_s \circ (\mathbf{F}_l (\mathcal{N}(\mathbf{x}) - \mathbf{x}) \mathbf{F}_r^\top)\|_F^2 = \|\text{diag}(\text{vec}(\mathbf{S}_s))(\mathbf{F}_r \otimes \mathbf{F}_l)\text{vec}(\mathcal{N}(\mathbf{x}) - \mathbf{x})\|_2^2,$$

where $\otimes$ is the Kronecker product of two matrices. Setting $\mathbf{J}_s = \text{diag}(\text{vec}(\mathbf{S}_s))(\mathbf{F}_r \otimes \mathbf{F}_l)$, the loss function in our NN training can be written as

$$\Phi_s(\mathbf{W}) = \frac{1}{2n} \sum_{i=1}^{n} \|\mathbf{J}_s \text{vec}(\mathcal{N}(\mathbf{x}_i) - \mathbf{x}_i)\|_2^2 = \frac{1}{2n}(\mathbf{u} - \mathbf{y})^\top (\mathbf{I} \otimes \mathbf{J}_s^\top \mathbf{J}_s)(\mathbf{u} - \mathbf{y}), \tag{79}$$

where $\mathbf{I}$ is the $n$-by-$n$ identity matrix and $\mathbf{u}, \mathbf{y}$ are the vectors of length $n \times N_{\text{pixel}}$ given by $\mathbf{u} = (\text{vec}(\mathcal{N}(\mathbf{x}_1))^\top, \ldots, \text{vec}(\mathcal{N}(\mathbf{x}_n))^\top)^\top$ and $\mathbf{y} = (\text{vec}(\mathbf{x}_1)^\top, \ldots, \text{vec}(\mathbf{x}_n)^\top)^\top$. Hence, the discrete NTK matrix is given by $n^{-1}\mathbf{H}^\infty(\mathbf{I} \otimes \mathbf{J}_s^\top \mathbf{J}_s)$, where the $(i,j)$th sub-block is $\mathbf{H}_{ij}^\infty = \left\langle \frac{\partial \text{vec}(\mathcal{N}(\mathbf{x}_i; \mathbf{W}))}{\partial \mathbf{W}}, \frac{\partial \text{vec}(\mathcal{N}(\mathbf{x}_j; \mathbf{W}))}{\partial \mathbf{W}} \right\rangle$ for $i, j = 1, \ldots, n$. Here, $\left\langle \frac{\partial \text{vec}(\mathcal{N}(\mathbf{x}_i; \mathbf{W}))}{\partial \mathbf{W}}, \frac{\partial \text{vec}(\mathcal{N}(\mathbf{x}_j; \mathbf{W}))}{\partial \mathbf{W}} \right\rangle$ is interpreted as the $N_{\text{pixel}}$-by-$N_{\text{pixel}}$ matrix whose $(i', j')$th entry is $\left\langle \frac{\partial [\text{vec}(\mathcal{N}(\mathbf{x}_i; \mathbf{W}))_{i'}]}{\partial \mathbf{W}}, \frac{\partial [\text{vec}(\mathcal{N}(\mathbf{x}_j; \mathbf{W}))_{j'}]}{\partial \mathbf{W}} \right\rangle$. This means that the frequency bias behavior during NN training is directly affected by the choice of $s$. Alternatively, one can consider this problem more abstractly from an operator learning perspective, which is presented in appendix G. We remark that while eq. (79) is a mathematically equivalent expression for the loss function that allows us to easily express the NTK, in practice, we implement the loss function based on eq. (78) using a 2D FFT protocol.

We use the same autoencoder architecture as in (Chollet, 2016), except with the loss function in eq. (79) for $s = -1, 0, 1$. We train the autoencoder using mini-batch gradient descent with batch size equal to 256. We first pollute the training images with low-frequency noise and train the NN, hoping that the trained NN will act as a filter for the noise. We see that training the NN with eq. (79) for $s = 1$ gives us the best results due to the high-frequency bias induced by choice of the loss function. Although $\mathbf{H}^\infty$ is low-frequency bias, the high-frequency bias of $\mathbf{J}_s^\top \mathbf{J}_s$ dominates for sufficiently large $s$. In that case, the low-frequency noise barely changes the training in the earlier epochs as the low-frequency components of the residual correspond to small eigenvalues of $\mathbf{H}^\infty(\mathbf{I} \otimes \mathbf{J}_s^\top \mathbf{J}_s)$. Similar results are discussed in (Engquist et al., 2020) and (Zhu et al., 2021) in the inverse problem and image processing contexts, respectively.

The opposite phenomenon occurs when we add high-frequency noise (see Figure 4, bottom row). Since $\mathbf{H}^\infty$ by itself makes the NN training procedure bias towards low-frequencies, the output for $s = 0$ does already filter high-frequency noise. Since $\mathbf{J}_s^\top \mathbf{J}_s$ for $s < 0$ further biases towards low-frequencies, one can obtain better high-frequency filters. We observe that the best denoising results for the autoencoder come from selecting $s = -1$ (see Figure 4, bottom row).

## G  NTK AND FREQUENCY BIAS IN OPERATOR LEARNING

We saw how using the $H^s$-based losses can help us tune frequency bias in training the autoencoder (see section 6.3). In fact, in training the autoencoder, we are learning the identity operator on the space of images. In this section, we briefly mention the NTK associated with operator learning and its consequences. Let $\mathcal{L}$ be a linear operator on $L^2(\mathbb{S}^{d-1})$. Let $\mathcal{D} = \{f_1, \ldots, f_N\}$ be a finite subset of $L^2(\mathbb{S}^{d-1})$, and $\mathbf{x}_1, \ldots, \mathbf{x}_M$ be distinct "samplers" in $\mathbb{S}^{d-1}$. Since $\mathcal{L}$ is linear, without loss of generality, we assume $\left\| (f_i(\mathbf{x}_1), \ldots, f_i(\mathbf{x}_M))^\top \right\|_2 = 1$ for $i = 1, \ldots, N$. Given a function $f$, we use $f(\mathbf{x})$ to denote the vector $(f(\mathbf{x}_1), \ldots, f(\mathbf{x}_M))^\top$. We consider a fully-connected two-layer ReLU NN, $\mathcal{N} = (\mathcal{N}_1, \ldots, \mathcal{N}_M)$, that takes $M$ inputs and produces $M$ outputs. The goal of training is to learn the linear operator $\mathcal{L}$ on $\mathcal{D}$. That is, we want $\mathcal{N}_j(f(\mathbf{x}_1), \ldots, f(\mathbf{x}_M)) \approx (\mathcal{L}f)(\mathbf{x}_j)$. In other words, given the samples $f(\mathbf{x})$ of a function $f$, we want the NN to output the values of $\mathcal{L}f$ at $\mathbf{x}_1, \ldots, \mathbf{x}_M$, i.e., $(\mathcal{L}f)(\mathbf{x})$. As before, we let $\mathbf{P}$ be an $M \times M$ symmetric positive definite matrix that measures the distance between two $L^2$ functions given their samples at $\mathbf{x}_1, \ldots, \mathbf{x}_M$. That is,

$$D(f(\mathbf{x}), \ g(\mathbf{x})) = (f(\mathbf{x}) - g(\mathbf{x}))^\top \mathbf{P} (f(\mathbf{x}) - g(\mathbf{x})).$$

We then consider a loss function of the operator NN given by

$$\Phi(\mathbf{W}) = \frac{1}{N} \sum_{i=1}^{N} D((\mathcal{L}f_i)(\mathbf{x}), \ \mathcal{N}(f_i)), \tag{80}$$

where $\mathcal{N}(f_i) := (\mathcal{N}_1(f_i), \ldots, \mathcal{N}_M(f_i))^\top$ in which $\mathcal{N}_j(f_i)$ denotes the $j$th output of the NN when we input $f_i$. That is, $\mathcal{N}_j(f_i) = \mathcal{N}_j(f_i(\mathbf{x}))$. Let $w_{i,j}^{(1)}$ be the $i$th weight in the $j$th neuron of the first hidden layer, $1 \le i \le M$ and $1 \le j \le m$, where $m$ is the number of hidden neurons. Let $b_j$ be the bias term of the $j$th neuron in the hidden layer. Let $w_{i,j}^{(2)}$ be the $i$th weight in the $j$th neuron of the output layer, so $1 \le i \le m$ and $1 \le j \le M$. We assume the same initialization scheme. That is, $w_{i,j}^{(1)}$ are initialized from iid Gaussian and $b_j$ are initialized to $0$, and they are updated during training, whereas $w_{i,j}^{(2)}$ are initialized from iid Rademacher random variables and are not updated during training. In the derivation of the NTK below, we let $\mathbf{W}$ to be the vector of all trainable weights and biases, $w_{i,j}^{(1)}$ and $b_j$, enumerated in an (arbitrary) fixed order. This allows us to write $\mathbf{W} = (w_1, \ldots, w_K)^\top$, where $K = m(M + 1)$.

We assume that the gradient flow algorithm is used to train the NN, i.e., $\frac{d\mathbf{W}}{dt} = -\frac{\partial \Phi}{\partial \mathbf{W}}$. We define the vector of labels by

$$\mathbf{y}_i = (\mathcal{L}f_i(\mathbf{x}_1), \ldots, \mathcal{L}f_i(\mathbf{x}_M))^\top, \qquad \mathbf{y} = (\mathbf{y}_1^\top, \ldots, \mathbf{y}_N^\top)^\top.$$

Similarly, we define the vector of the NN outputs by

$$\mathbf{u}_i(\mathbf{W}) = (\mathcal{N}_1(f_i; \mathbf{W}), \ldots, \mathcal{N}_M(f_i; \mathbf{W}))^\top, \qquad \mathbf{u} = (\mathbf{u}_1^\top, \ldots, \mathbf{u}_N^\top)^\top.$$

Our goal is to understand $\frac{d}{dt}(\mathbf{y} - \mathbf{u})$ and write it in terms of $\mathbf{y} - \mathbf{u}$. To this end, we first consider

$$\frac{d(\mathbf{y}_i - \mathbf{u}_i(\mathbf{W}))}{dt} = -\frac{d\mathbf{u}_i(\mathbf{W})}{dt} = -\frac{d\mathbf{u}_i(\mathbf{W})}{d\mathbf{W}} \frac{d\mathbf{W}}{dt} = \frac{d\mathbf{u}_i(\mathbf{W})}{d\mathbf{W}} \frac{\partial \Phi(\mathbf{W})}{\partial \mathbf{W}}. \tag{81}$$

Denote by $w_k$ the $k$th entry of $\mathbf{W}$, where $1 \le k \le K$. Then, by the chain rule, the $k$th entry of $\frac{\partial \Phi(\mathbf{W})}{\partial \mathbf{W}}$ can be written as

$$\left( \frac{\partial \Phi(\mathbf{W})}{\partial \mathbf{W}} \right)_k = \frac{\partial \Phi(\mathbf{W})}{\partial w_k} = \frac{1}{N} \sum_{i'=1}^{N} \frac{\partial}{\partial w_k} D\left( (\mathcal{L}f_{i'})(\mathbf{x}), \ \mathcal{N}(f_{i'})(\mathbf{x}) \right)$$

$$= \frac{1}{N} \sum_{i'=1}^{N} \left[ \frac{\partial D\left( (\mathcal{L}f_{i'})(\mathbf{x}), \ \mathcal{N}(f_{i'})(\mathbf{x}) \right)}{\partial \mathcal{N}(f_{i'}; \mathbf{W})} \right] \left[ \frac{\partial \mathcal{N}(f_{i'}; \mathbf{W})}{\partial w_k} \right]$$

$$= -\frac{1}{N} \sum_{i'=1}^{N} \left[ \frac{\partial \mathcal{N}_1(f_{i'})}{\partial w_k} \quad \cdots \quad \frac{\partial \mathcal{N}_M(f_{i'})}{\partial w_k} \right] \mathbf{P} \begin{bmatrix} \mathcal{L}f_{i'}(\mathbf{x}_1) - \mathcal{N}_1(f_{i'}) \\ \vdots \\ \mathcal{L}f_{i'}(\mathbf{x}_M) - \mathcal{N}_M(f_{i'}) \end{bmatrix}.$$

Hence, we have

$$\frac{\partial \Phi(\mathbf{W})}{\partial \mathbf{W}} = -\frac{1}{N} \sum_{i'=1}^{N} \begin{bmatrix} \frac{\partial \mathcal{N}_1(f_{i'})}{\partial w_1} & \cdots & \frac{\partial \mathcal{N}_M(f_{i'})}{\partial w_1} \\ \vdots & \ddots & \vdots \\ \frac{\partial \mathcal{N}_1(f_{i'})}{\partial w_K} & \cdots & \frac{\partial \mathcal{N}_M(f_{i'})}{\partial w_K} \end{bmatrix} \mathbf{P} \begin{bmatrix} \mathcal{L}f_{i'}(\mathbf{x}_1) - \mathcal{N}_1(f_{i'}) \\ \vdots \\ \mathcal{L}f_{i'}(\mathbf{x}_M) - \mathcal{N}_M(f_{i'}) \end{bmatrix}. \tag{82}$$

Now, combining eq. (81) with eq. (82), we have

$$-\frac{d\mathbf{u}_i(\mathbf{W})}{dt} = -\frac{1}{N} \begin{bmatrix} \frac{\partial \mathcal{N}_1(f_i)}{\partial w_1} & \cdots & \frac{\partial \mathcal{N}_1(f_i)}{\partial w_K} \\ \vdots & \ddots & \vdots \\ \frac{\partial \mathcal{N}_M(f_i)}{\partial w_1} & \cdots & \frac{\partial \mathcal{N}_M(f_i)}{\partial w_K} \end{bmatrix} \times$$

$$\left( \sum_{i'=1}^{N} \underbrace{\begin{bmatrix} \frac{\partial \mathcal{N}_1(f_{i'})}{\partial w_1} & \cdots & \frac{\partial \mathcal{N}_M(f_{i'})}{\partial w_1} \\ \vdots & \ddots & \vdots \\ \frac{\partial \mathcal{N}_1(f_{i'})}{\partial w_K} & \cdots & \frac{\partial \mathcal{N}_M(f_{i'})}{\partial w_K} \end{bmatrix}}_{\mathbf{J}_{i'}} \mathbf{P} \begin{bmatrix} \mathcal{L}f_{i'}(\mathbf{x}_1) - \mathcal{N}_1(f_{i'}) \\ \vdots \\ \mathcal{L}f_{i'}(\mathbf{x}_M) - \mathcal{N}_M(f_{i'}) \end{bmatrix} \right),$$

which gives us

$$\frac{d(\mathbf{y} - \mathbf{u}(\mathbf{W}))}{dt} = -\frac{d\mathbf{u}(\mathbf{W})}{dt} = -\frac{1}{N} \begin{bmatrix} \mathbf{J}_1^\top \mathbf{J}_1 & \cdots & \mathbf{J}_1^\top \mathbf{J}_N \\ \vdots & \ddots & \vdots \\ \mathbf{J}_N^\top \mathbf{J}_1 & \cdots & \mathbf{J}_N^\top \mathbf{J}_N \end{bmatrix} (\mathbf{I}_N \otimes \mathbf{P})(\mathbf{y} - \mathbf{u}). \tag{83}$$

The derivation so far does not rely on the architecture of the NN. In fact, it holds for the loss function $\Phi$ defined in eq. (80) with any abstract function $\mathcal{N}(f_i; \mathbf{W})$. Now, we exploit the NN architecture to study $\mathbf{J}_i^\top \mathbf{J}_{i'}$. Consider the $(j, j')$th entry of $\mathbf{J}_i^\top \mathbf{J}_{i'}$. We have

$$(\mathbf{J}_i^\top \mathbf{J}_{i'})_{(j,j')} = \frac{1}{m} \sum_{k=1}^{m} w_{k,j}^{(2)} w_{k,j'}^{(2)} \frac{f_i(\mathbf{x})^\top f_{i'}(\mathbf{x}) + 1}{2} \mathbb{1}_{\{f_i(\mathbf{x})^\top \mathbf{w}_k^{(1)} + b_k^{(1)} \geq 0, f_{i'}(\mathbf{x})^\top \mathbf{w}_k^{(1)} + b_k^{(1)} \geq 0\}}, \tag{84}$$

where $\mathbf{w}_k^{(1)} = (w_{1,k}^{(1)}, \ldots, w_{M,k}^{(1)})$ is the collection of weights on the $k$th hidden neuron. Now, note that $w_{k,j}^{(2)} w_{k,j'}^{(2)} = 1$ if $j = j'$ and is a Rademacher random variable when $j \neq j'$. Hence, by the way we initialize $w_{i,j}^{(1)}$, we have

$$\mathbf{J}_i^\top \mathbf{J}_{i'} \xrightarrow{m \to \infty} \left[ \frac{(f_i(\mathbf{x})^\top f_{i'}(\mathbf{x}) + 1)(\pi - \arccos(f_i(\mathbf{x})^\top f_{i'}(\mathbf{x})))}{4\pi} \right] \mathbf{I}_M, \tag{85}$$

where the convergence is entry-wise. Now, suppose we define the $N \times N$ matrix $\mathbf{H}^\infty$ as

$$H_{i,i'}^\infty = \frac{(f_i(\mathbf{x})^\top f_{i'}(\mathbf{x}) + 1)(\pi - \arccos(f_i(\mathbf{x})^\top f_{i'}(\mathbf{x})))}{4\pi}. \tag{86}$$

That is, we define $\mathbf{H}^\infty$ as in the function learning case, except that the inputs are $\{f_i(\mathbf{x})\}_{i=1}^N$ instead of $\{\mathbf{x}_i\}_{i=1}^n$. By eq. (85), we have

$$\begin{bmatrix} \mathbf{J}_1^\top \mathbf{J}_1 & \cdots & \mathbf{J}_1^\top \mathbf{J}_N \\ \vdots & \ddots & \vdots \\ \mathbf{J}_N^\top \mathbf{J}_1 & \cdots & \mathbf{J}_N^\top \mathbf{J}_N \end{bmatrix} \xrightarrow{m \to \infty} \mathbf{H}^\infty \otimes \mathbf{I}_M, \tag{87}$$

where the convergence is entrywise. Hence, if we discretize the gradient flow algorithm and apply gradient descent with step size $\eta$, then by eq. (83) and eq. (87), we expect that

$$\mathbf{y}_i - \mathbf{u}_i(k) \approx (\mathbf{y}_i - \mathbf{u}_i(k-1)) - \frac{\eta}{N} \sum_{i'=1}^{N} H_{i,i'}^\infty \mathbf{P}(\mathbf{y}_{i'} - \mathbf{u}_{i'}(k-1)), \tag{88}$$

or otherwise written more compactly,

$$\mathbf{y} - \mathbf{u}(k) \approx \left(\mathbf{I}_{MN} - \frac{\eta}{N}(\mathbf{H}^\infty \otimes \mathbf{I}_M)(\mathbf{I}_N \otimes \mathbf{P})\right)^k \mathbf{y} \tag{89}$$

if the variance in the initialization is small enough.

Given this characterization of the residual equation 89, we can see how the usage of the Sobolev loss plays a role in tuning frequency bias. Suppose we decompose the residual vector into

$$\mathbf{y}_i - \mathbf{u}_i(k) = \sum_{\ell=0}^{\infty} \sum_{p=1}^{N(d,\ell)} \hat{\gamma}_i^{\ell,p} \mathbf{Y}_{\ell,p},$$

where $\mathbf{Y}_{\ell,p} = (Y_{\ell,p}(\mathbf{x}_1), \ldots, Y_{\ell,p}(\mathbf{x}_M))^\top$ is the evaluation of the spherical harmonic on $\mathbb{S}^{d-1}$. Then, $\hat{\gamma}_i^{\ell,p}$ measures the amount of frequency loss associated with the $(\ell, p)$th frequency in learning the function $f_i$. Let $\hat{\boldsymbol{\gamma}}^{\ell,p}(k) = (\hat{\gamma}_i^{\ell,p}(k), \ldots, \hat{\gamma}_N^{\ell,p}(k))^\top$. The size of $\hat{\boldsymbol{\gamma}}^{\ell,p}(k)$ measures the amount of overall frequency-$\ell$ components (in the $p$th direction) in the residuals after the $k$th iteration. Assume $\omega_\ell$ is an eigenvalue of $\mathbf{P}$ associated with the eigenspace that is approximately spanned by $\{\mathbf{Y}_{\ell,p}\}_{p=1}^{N(d,\ell)}$. Note that this is the case when $\mathbf{P} = \mathbf{P}_s$ is associated with the squared-$H^s$ loss on $\mathbb{S}^{d-1}$. Then, by eq. (88), we have

$$\hat{\boldsymbol{\gamma}}^{\ell,p}(k) \approx \left(\mathbf{I}_N - \frac{\eta}{N}\omega_\ell \mathbf{H}^\infty\right)^k \hat{\mathbf{y}}^{\ell,p}, \tag{90}$$

where $\hat{\mathbf{y}}^{\ell,p} = (\hat{y}_1^{\ell,p}, \ldots, \hat{y}_N^{\ell,p})$ is defined by the decomposition

$$\mathbf{y}_i = \sum_{\ell=0}^{\infty} \sum_{p=1}^{N(d,\ell)} \hat{y}_i^{\ell,p} \mathbf{Y}_{\ell,p}.$$

Equation (90) demonstrates that if $\omega_\ell$ is relatively large for bigger $\ell$ (e.g., when we use the squared-$H^s$ loss with a large $s > 0$), then we expect that $\boldsymbol{\gamma}^{\ell,p}(k)$ decays much faster for high frequencies than the low ones, and vice versa. This connects frequency bias in operator learning to the spectral properties of $\mathbf{P}_s$. We studied and justified the phenomena in the autoencoder experiment in Sec. 6.3.

## H  COMPUTATION OF QUADRATURE WEIGHTS

In practice, the training dataset usually does not come with a carefully designed quadrature rule. Hence, we inevitably need to compute a set of quadrature weights before training the NN. In this section, we briefly discuss methods for computing positive quadrature weights.

Given a set of points $\{\mathbf{x}_i\}_{i=1}^n$ on $\mathbb{S}^{d-1}$, we wish to construct a quadrature rule so that

$$I_n(f) := \sum_{i=1}^{n} c_i f(\mathbf{x}_i) \approx \int_{\mathbb{S}^{d-1}} f(\mathbf{x})d\mathbf{x}$$

for sufficiently smooth $f$, where $c_i > 0$ are positive quadrature weights. One approach that could give us a very accurate quadrature rule is to guarantee that

$$I_n(f) = \int_{\mathbb{S}^{d-1}} f(\mathbf{x})d\mathbf{x}, \qquad f \in \Pi_\ell^d, \tag{91}$$

where $\dim(\Pi_\ell^d) \leq n$. The one-dimensional case of such quadrature rules was studied in (Austin & Trefethen, 2017; Yu & Townsend, 2022) and the general higher-dimensional case was analyzed in (Mhaskar et al., 2000; Dai & Xu, 2013). Given any dataset $\{\mathbf{x}_i\}_{i=1}^n$ and $\ell$ so that $\dim(\Pi_\ell^d) \leq n$, one cannot guarantee the existence of a positive quadrature rule satisfying eq. (91) (Mhaskar et al., 2000; Dai & Xu, 2013), even when the distribution of $\{\mathbf{x}_i\}_{i=1}^n$ is very regular (Yu & Townsend, 2022). On the other hand, by choosing $\ell$ to be small, we can eventually find an $\ell$ for which eq. (91) holds. When such a positive quadrature rule exists, Mhaskar et al. (2000) proposed to solve the

following feasible constrained quadratic program

$$
\begin{aligned}
\min_{c_i} \quad & \sum_{i=1}^{n} c_i^2 \\
\text{s.t.} \quad & c_i > 0 \quad \forall 1 \le i \le n, \\
& \sum_{i=1}^{n} c_i Y_{j,p}(\mathbf{x}_i) = \int_{\mathbb{S}^{d-1}} Y_{j,p}(\mathbf{x}) d\mathbf{x} \quad \forall 0 \le j \le \ell, 1 \le p \le N(d,j).
\end{aligned}
$$

While eq. (91) gives us a guarantee on the accuracy of the quadrature rule (provided $\ell$ is not too small), it is not always practical to compute the quadrature weights in this way. Indeed, if $d$ is large, then we need a tremendous amount of points to guarantee that $\dim(\Pi_\ell^d) \le n$ even for a small $\ell$. Also, if we have too many data points, then the quadratic program can get infeasible to solve. Hence, we need some other methods for computing quadrature weights that, albeit less accurate, can be applied more cheaply to general datasets. One of the many possible approaches is to do kernel density estimation (Rosenblatt, 1956; Parzen, 1962). To do so, we fix a positive kernel $\mathcal{K}(\mathbf{x}, \mathbf{y}) = \mathcal{K}(\arccos(\mathbf{x}^\top \mathbf{y}))$ defined on $\mathbb{S}^{d-1} \times \mathbb{S}^{d-1}$. A common choice of $\mathcal{K}$ can be the Gaussian density function of standard deviation 1 centered at 0. For each $h > 0$, we then define a function $p_h$ on $\mathbb{S}^{d-1}$ by

$$
p_h(\mathbf{x}) = \sum_{i=1}^{n} \mathcal{K}\left( \frac{\arccos(\mathbf{x}^\top \mathbf{x}_i)}{h} \right).
$$

The bandwidth $h$ is a hyperparameter, and with an appropriate $h$, the function $p_h$ is an (unnormalized) estimate of the density function of the distribution of nodes. Hence, by setting

$$
c_i = A_d \frac{p_h^{-1}(\mathbf{x}_i)}{\sum_{j=1}^{n} p_h^{-1}(\mathbf{x}_j)}, \tag{92}
$$

we obtain a positive quadrature rule that approximates the integral of smooth functions on $\mathbb{S}^{d-1}$.

