# OpenReview forum: "Tuning Frequency Bias in Neural Network Training with Nonuniform Data"
_ICLR.cc/2023/Conference — ICLR 2023 poster_

### Official Review · Reviewer_g5Zh · 2022-10-21

**Confidence:** 3
**Clarity, Quality, Novelty And Reproducibility:** The results is hard to read.  See wea…
**Correctness:** 4
**Technical Novelty And Significance:** 4
**Empirical Novelty And Significance:** 4
**Recommendation:** 6

**Strength And Weaknesses:**

Strength:
The link between the data-dependent quadrature rule and non-uniform data is super interesting. I also believe this is the right way to conduct the research.

Weakness:

However, the paper presents a pretty hard-to-read bond, and I can't understand how different components of the assumptions are involved in the final results. I suggest the authors use a simple model to illustrate how different factors contribute to the bounds. The starting point I suggest is [1-5], which considers kernel gradient descent under Sobolev/L2 losses.

At the same time, in the abstract and main context, the author claims
> Small generalization errors of over-parameterized neural networks (NNs) can be partially explained by the frequency biasing phenomenon
However, as I understand, this paper deals with optimization but not the generalization error. I think the author can follow that variance analysis in [1-5] to see how this relates to the generalization properties. I think it would be interesting to see whether the error introduced by the quadrature rule will ruin the generalization error. (Or this can be revealed by the theorem in this version's paper)

My third concern is about the algorithm, the algorithm is not a gradient descent to original loss but needs to know the P. For a random dataset sampled from an unknown distribution, Is there any way to estimate the P?

At the same time,  I would like to ask is $H^\infty P$ diagonalizable? To me this formulation is strange for the matrix here should be the gram matrix, which is symmetry. I guess the matrix should be $\nabla f^\top P \nabla f$.

[1] Pillaud-Vivien L, Rudi A, Bach F. Statistical optimality of stochastic gradient descent on hard learning problems through multiple passes. Advances in Neural Information Processing Systems, 2018, 31.

[2] Lin J, Rosasco L. Optimal rates for multi-pass stochastic gradient methods[J]. The Journal of Machine Learning Research, 2017, 18(1): 3375-3421.

[3] Nitanda A, Suzuki T. Optimal rates for averaged stochastic gradient descent under neural tangent kernel regime[J]. arXiv preprint arXiv:2006.12297, 2020.

[4] Mücke N, Reiss E. Stochastic gradient descent in Hilbert scales: smoothness, preconditioning and earlier stopping. arXiv preprint arXiv:2006.10840, 2020.

[5] Lu Y, Blanchet J, Ying L. Sobolev Acceleration and Statistical Optimality for Learning Elliptic Equations via Gradient Descent. arXiv preprint arXiv:2205.07331, 2022.

**Summary Of The Paper:**

This paper deal with training a NN with the nonuniform data distribution under the NTK regime. This paper assumes a data-dependent quadrature rule to build bound for non-uniform data.

**Summary Of The Review:**

The paper is super interesting, but the results are hard to interpret. Although I have criticized this paper a lot int he weakness part, I still think  the quadrature rule  idea here is super interesting and is worth accepting.

---

> ### Author Response · Authors · 2022-11-12
> **Response to the Reviewer**
>
> We thank the reviewer for the careful review and comments. Below we address the concerns and questions raised in this review.
>
> * "However, the paper presents a pretty hard-to-read bond...":
>
> We thank the referee for the suggestion, to which we agree.  To make the dependence of the error bound to different parameters of the NN clearer, we have added Theorem 1' in Appendix C, which is in a similar style to the results in [1]-[5]. This new statement is equivalent to Theorem 1, but it bounds the error in approximating the residual $\mathbf{y} - \mathbf{u}(k)$ by $(\mathbf{I} - 2\eta\mathbf{H}^\infty \mathbf{P})^k \mathbf{y}$ explicitly using the parameters of the NN and the algorithm, e.g., $m, n, \eta$, etc. Still, the bound is a bit complicated, but we hope it clarifies how the error depends on each parameter. Accordingly, we added Theorem 2' and Theorem 4', which can be proved based on Theorem 1' instead of Theorem 1.
>
> * "At the same time, in the abstract and main context...":
>
> Thank you for bringing up this issue, which was not explicitly discussed in the paper. Generalization error is an interesting topic and we have added two remarks after the proofs of Theorem 2 and Theorem 4 on the $L^2$ generalization error with respect to the uniform distribution on $\mathbb{S}^{d-1}$, i.e., $\\|{g - \mathcal{N}}\\|\_{L^2}$, where $g$ is the target function and $\mathcal{N}$ is the NN function. This is a natural error to look at because the "frequency" studied in our paper is a notion that is based on the Lebesgue measure on $\mathbb{S}^{d-1}$. Nevertheless, it is clear that the generalization error with respect to a (possibly nonuniform) distribution on $\mathbb{S}^{d-1}$ can be bounded by a constant factor of $\\|{g - \mathcal{N}}\\|\_{L^2}$, as long as the density is bounded on $\mathbb{S}^{d-1}$.
>
> In our remarks after the proofs of Theorem 2 (in Appendix D.1) and Theorem 4 (in Appendix E.2), we showed that the residual vectors that appear in Theorem 2 and Theorem 4 could be related to the generalization error $\\|{g - \mathcal{N}}\\|\_{L^2}$ by introducing yet another quadrature error. While more details can be found in the appendices, the high-level conclusion is that when using the squared-$H^s$ loss, we have
> \\[
>     \\|{g-\mathcal{N}_k}\\|\_{L^2} = \sqrt{\sum\_{\ell=0}^L \left(1 - 2\eta\mu\_j(1+\ell)^{2s}\right)^{2k} \\|{g\_\ell}\\|^2\_{L^2}} + E\_{\text{NTK}} + E\_{\text{quad}},
> \\]
> where $E\_{\text{NTK}}$ is an error that can be made arbitrarily small by taking $\kappa$ small enough and $m$ large enough and $E\_{\text{quad}}$ involves only quadrature errors and can be made arbitrarily small if we make the quadrature rule accurate enough.
>
> * "My third concern is about the algorithm...":
>
> There are many techniques for computing the quadrature rule or estimating the density. In Appendix H, we discuss kernel density estimation and other common algorithms for computing the quadrature weights. These techniques, in general, perform better when the dimension of the input space is relatively small. When the dimension is very large, e.g., in image processing tasks, computing a good quadrature is more difficult due to the curse of dimensionality. Nevertheless, as shown in the MNIST experiment, for image processing tasks, the more interesting notion of frequency is the frequency on the 2D grid, and we have added Appendix G to formally formulate this problem in our framework and justify the use of the $H^s$-based losses. We hope this example eases the worries about the lack of a good quadrature rule in high-dimensional tasks.
>
> * "At the same time, I would like to...":
>
> Yes, the matrix $\mathbf{H}^\infty \mathbf{P}$ is diagonalizable, as is now shown in equation (10). The intuition is that although $\mathbf{H}^\infty \mathbf{P}$ is not symmetric, in the inner product space induced by $\mathbf{P}$ (thus in a weighted Hilbert space), $\mathbf{H}^\infty \mathbf{P}$ is a self-adjoint operator (see equation (21)). Hence, $\mathbf{H}^\infty \mathbf{P}$ on $(\mathbb{R}^n, \langle \cdot,\cdot \rangle\_{\mathbf{P}})$ is a natural analogue of $\mathbf{H}^\infty$ on $(\mathbb{R}^n, \langle \cdot,\cdot \rangle)$, where $\langle \cdot,\cdot \rangle$ is the standard $\ell^2$ inner product.

---

> ### Author Response · Authors · 2022-12-06
> **Follow up**
>
> Thank you for your comments and questions. We hope that we have answered all your questions. Please let us know if you have any more remarks.

---

### Official Review · Reviewer_JXdi · 2022-10-21

**Confidence:** 3
**Correctness:** 3
**Technical Novelty And Significance:** 2
**Empirical Novelty And Significance:** 2
**Recommendation:** 5

**Clarity, Quality, Novelty And Reproducibility:**

The theoretical techniques are presented clearly, but the motivations or reasoning for the assumptions and formulations are a bit lacking. I think the central concern is about the quadrature weights $c_i$.

The idea of using quadrature and the Sobolev-norm seems novel. Besides that, I believe the NTK analysis along with the frequency bias are from previous works.


Some questions I wish the authors can answer:
1. In particular, I find assuming (17) to hold extremely unrealistic. Can the authors provide a general approach to compute the quadrature weights $c_i$? Can the authors also comment on the difficulty of obtaining accurate quadrature weights, especially the dependence on the input dimension $d$?

2. In Section F.1, the procedure of computing $c_i$ seems very ad hoc. Can the authors provide further explanation on why optimizing over $\sum c_i^2$ and what does it mean by `the quadrature rule is exact on $\Pi_{55}^2$', and why $55$ is the meaningful choice?





**Strength And Weaknesses:**

Strength:
1. The introduction of the Sobolev-norm loss function is new and interesting. It can explicitly guide NTK training to favor higher frequencies or lower frequencies.

Weakness:
1. The idea of using quadrature to approximate the loss function seems not well supported. My main concern is that it seems not easy to find good quadrature when given only the training data, namely, identifying proper $c_i$. Theorem 2 bounds the loss by the quadrature approximation error, and Theorem 3 assumes there is a quadrature weight $c_i$ ready to use. None of these indicate that $c_i$ can be determined properly. This heavily decreases the significance of this work.

**Summary Of The Paper:**

This paper proposes to use quadratures to approximate the $L_2$ error of NTK training. By doing so, the spectral analysis of NTK training can be extended to non-uniform data. Further, this paper proposes to use a Sobolev norm to modify the loss function so that the convergence under the modified loss function will exhibit different spectral behavior. By doing so the authors were able to tune the spectral bias of NN training.

**Summary Of The Review:**

Given the current state of this paper, I recommend weak reject due to the lack of support on the quadrature weights assumption, Eqn (17).

---

> ### Author Response · Authors · 2022-11-12
> **Response to the Reviewer**
>
> We thank the reviewer for the careful review and comments. Below we address the concerns and questions raised in this review.
>
> * "The idea of using quadrature..." + "In particular, I find assuming (17) to hold...":
>
> The reviewer has pointed out the difficulty in computing a good quadrature rule, which indeed causes an issue when the input space dimension is large. In Appendix H, we have discussed a couple of general approaches to computing the quadrature weights. While the quadrature rule might be coarse when $d$ is large, and the number of training data is only moderate, we do have some important applications in mind for a small $d$ in which we have certain control of the position of training data $\mathbf{x}\_1, \ldots, \mathbf{x}\_n$. One of the examples is PDE learning using physics-informed neural networks (PINNs), in which the samples come from a low-dimensional domain, and it is often desirable to place more samples in the regions where the solution has a singularity or is oscillatory [Mao et al., 2020]. In this example, if we have a reasonable amount of training data, we are expected to find a good quadrature rule, allowing us to study frequency bias with nonuniform training data.
>
> In the case when $d$ is large, we have added a comment after equation (18) (formerly equation (17)) to admit the difficulty in obtaining a good quadrature rule. However, it is also worth to be pointed out that when the dimension $d$ is large and the number of training data $n$ is not enormous, the aliasing error makes the high frequencies indistinguishable from the low ones. This shows that in high-dimensional spaces, frequency bias is a theoretically interesting question only if we assume a large amount of data, in which case finding a good quadrature rule is possible.
>
> * "In Section F.1, the procedure of computing $c\_i$ seems very ad hoc...":
>
> Thank you for pointing out the need for more explanation for the choices of $\{c\_i\}$. To justify the method more rigorously, we have refined our discussion in Appendix F.1 (and referred to it in the main text). While more details can be found in the appendix, we now briefly answer the questions raised by the reviewer. By saying a quadrature rule $\\{c_i\\}\_{i=1}^n$ is exact on $\Pi^d\_\ell$, we mean
> \\[
>     \sum\_{i=1}^n c\_i g(\mathbf{x}\_i) = \int\_{\mathbb{S}^{d-1}} g(\mathbf{x}) d\mathbf{x}, \qquad g \in \Pi\_\ell^d.
> \\]
> The idea of solving the quadratic constraint program comes from the celebrated paper [Mhaskar et al., 2000]. As also discussed in this paper, it is not always possible to find a positive quadrature rule that is exact on $\Pi^d\_\ell$, even if $\text{dim} \Pi^d\_\ell \ll n$. In our experiment, $55$ is chosen to be the largest degree $L$ for which it is possible to efficiently find a positive quadrature rule that is exact on $\Pi^d\_\ell$. In addition, we want $L$ to be large because according to Theorem 3, $L$ being large is a heuristic for $|\epsilon\_1|$ and $|\epsilon\_2|$ to be small. A more detailed discussion of this can be found in Appendix F.1.
>
>
> Z. Mao, A.D. Jagtap, and G.E. Karniadakis. Physics-informed neural networks for high-speed flows. _Computer Methods in Applied Mechanics and Engineering_, 2020.
>
> H. N. Mhaskar, F. J. Narcowich, and J. D. Ward. Spherical Marcinkiewicz–Zygmund inequalities and positive quadrature. _Math. Comp._, 70(235):1113–1130, 2000.

---

> ### Author Response · Authors · 2022-12-06
> **Follow up**
>
> Thank you for your comments and questions. We hope that we have answered all your questions. Please let us know if you have any more remarks.

---

### Official Review · Reviewer_XAwZ · 2022-10-22

**Confidence:** 3
**Correctness:** 4
**Technical Novelty And Significance:** 3
**Empirical Novelty And Significance:** 3
**Recommendation:** 8

**Clarity, Quality, Novelty And Reproducibility:**

**Clarity**

The work is clear and well written.  I do have some comments and questions in the "Comments and Questions for the authors" section.

**Quality**

The results are of high quality.

**Originality**

This work is original in that it studies general quadratic losses and introduces the Soblev-type loss functions for the modulation of spectral bias.

**Reproducibility**

Since this is primarily a theoretical work with proofs I do not view reproducibility as applicable here.


**Strength And Weaknesses:**

**Strengths:**

The paper is well written and offers a new addition to the spectral bias literature by studying general quadratic losses.  By choosing $P$ appropriately one can compensate for bias's of the NTK.  Furthermore the Sobolev-based loss functions allow one to modify or reverse the spectral bias which can be useful when the low-frequency bias is a difficulty such as in value function approximation [1].

**Weaknesses**

This work shares the weaknesses of other works in this area, such as being limited to the NTK regime, fixing the outer layer weights, specific initialization, limited to shallow feedforward networks, etc.  However since this weakness is present in other works and the community has not figured out yet how to fully overcome these limitations it is reasonable that this work also operates in this regime.



**Summary Of The Paper:**

Motivated by quadrature rules, this paper studies the convergence of shallow ReLU networks trained on a quadratic loss defined by a symmetric positive definite matrix $P$.  They study the convergence rate along different components of the matrix determined by the NTK and the matrix $P$ for general target functions.  When $P$ is diagonal and the target is bandlimited, the convergence can be understood in terms of the eigenvalues of the NTK integral operator and the spherical harmonic components of the target.  They also demonstrate that by using a Sobolev-based loss function, one can modulate or even reverse the spectral bias by punishing low/high frequencies at different rates.

**Summary Of The Review:**

**Summary**

This work illustrates how one can modulate the bias arising from the NTK by optimizing a quadratic loss determined by an appropriately chosen matrix $P$.  Furthermore they introduce Sobolev-style loss functions to enhance or reverse the low frequency bias of feedforward networks.  I believe these are contributions that the ICLR community will be interested in, and thus I recommend to accept this paper.

**Comments and Questions for the authors**

Note on page 2 that equation (2) is only true when one is performing gradient descent on the population loss and not the empirical loss.  It is worth emphasizing this to the reader as one in practice only has access to the empirical loss.

On page 2 you state “Under the assumptions that the weights do not change much during training, one can consider the NTK in the mean-field regime given the underlying time-independent distribution of $W$, i.e., $K_\infty(x, x') = \mathbb{E}_W[K(x, x' ;W)]$ (Du et al., 2018).”
 I think it is more appropriate to call this regime the NTK regime and not the mean-field regime.  The parameterization and dynamics in the NTK setting are distinct from the mean field setting [2].

On page 4 you state “As a consequence of Proposition 1, the matrix $H^\infty P$ has positive eigenvalues, which we denote by $\lambda_{n - 1} \geq \cdots \geq \lambda_0 > 0$”.  This is true but it is not obvious that $H^\infty P$ is diagonalizable.  Worth referencing your discussion in page 1 in the Appendix here.

The formatting of Theorem 1 seems off.  Can you put the theorem statement in a theorem environment which should put the statement in italics.  Or alternatively, could you introduce a different formatting to make the statement stick out from the surrounding text.

On page 5 you state “Therefore, our rate of convergence does not vanish as
the number of training data points $n \rightarrow \infty$, provided that $M_{\textbf{P}} = \mathcal{O}(n^{−1/2})$
. This is the case when $P = n^{-1}I$ which corresponds to eq. (1).  So, we overcome the vanishing convergence rate issue that appears in previous analyses of frequency biasing (Arora et al., 2019; Basri et al., 2019). As $\eta$ decreases, the gradient descent algorithm gets closer to the gradient flow algorithm (Du et al., 2018), which allows us to more accurately quantify the frequency biasing (see section 4).”  This is not a fair comparison because Arora et al. [3]  and Basri et al. [4] work with the unnormalized squared error ($P = I$ without $1/n$ normalization) versus the choice $P = n^{-1} I$ corresponds to the mean squared error.  If you set $P = I$ you still have the same issue with vanishing convergence and learning rates as you send $n \rightarrow \infty$.

On page 6 you state “Under the reasonable assumption that our quadrature rule satisfies eq. (17), we can bound the quadrature errors appearing in Theorem 2.”  Is there any way to verify this assumption, at least in reasonable settings?

On page 6 in Theorem 3, may be worth stating that $\ell$ in the theorem comes from equation (17)

On page 8 you state that you “compute the quadrature weights $(c_i)_{i=1}^n$ for the loss function $\tilde{\Phi}$ in eq. (3).” Perhaps mention how you compute the quadrature weights.

On page 9 in the last paragraph you state “Here, we present the results of the autoencoder for image denoising using the MNIST dataset LeCun et al. (2010)”, however the figure is earlier on the page.  Also there is no conclusion section and the article ends abruptly.  This issue should be fixed before the final version.

On page 1 in the Appendix you state “Since $H^\infty$ are symmetric positive definite matrices (see Proposition 1), $H^\infty P$ has positive real eigenvalues.”  I think you meant to say “Since $H^\infty$, $P$ are symmetric positive definite matrices”

References:

[1] Ge Yang, Anurag Ajay, and Pulkit Agrawal. Overcoming the spectral bias of neural value approximation. In International Conference on Learning Representations, 2022. URL https:
//openreview.net/forum?id=vIC-xLFuM6.

[2] Mei et al. “A mean field view of the landscape of two-layer neural
networks”. In: Proceedings of the National Academy of Sciences (2018).

[3] S. Arora, S. S. Du, W. Hu, Z. Li, and R. Wang. Fine-grained analysis of optimization and generalization for overparameterized two-layer neural networks. In Inter. Conf. Mach. Learn., pp. 322–332. PMLR, 2019.

[4] R. Basri, D. Jacobs, Y. Kasten, and S. Kritchman. The convergence rate of neural networks for learned functions of different frequencies. Adv. Neur. Info. Proc. Syst., 32, 2019.

---

> ### Author Response · Authors · 2022-11-12
> **Response to the Reviewer**
>
> We thank the reviewer for the careful review and comments. Below we address the questions and comments raised in this review.
>
> * "Note on page 2 that…":
>
> Thank you for pointing out this issue. The formula assumes $\Phi$ (i.e. the population loss) is used. It is emphasized in the paper.
>
> * "On page 2 you state…":
>
> We have reworded the sentence and removed the term mean-field regime.
>
>
> * "On page 4 you state…":
>
> We have added equation (10) to show that $\mathbf{H}^\infty \mathbf{P}$ is diagonalizable.
>
> * "The formatting of…":
>
> We have italicized the statements of all our results.
>
> * "On page 5 you state…":
>
> Thank you for bringing up this subtle point. The learning rate stated in both [Arora et al., 2019] and [Basri et al., 2019] is $\mathcal{O}(\lambda_0/n^2)$. In their papers, if instead the mean-squared error is being used, then the $n^{-2}$ indeed disappears. However, the learning rate still depends on $\lambda_0$, which vanishes as the number of training data goes to infinity since the eigenvalues of $K^\infty$ have a limit point at $0$. Instead, our learning rate does not depend on $\lambda_0$. This issue was discussed more rigorously in detail in section 4.1 of [Su and Yang, 2019]. We have rephrased the corresponding sentence in our paper to make it fairer.
>
> * "On page 6 you state…":
>
> We believe that by saying "Is there any way to verify this assumption," the reviewer was asking about criteria for checking that $\gamma\_{\ell,n} \rightarrow 0$ as $n \rightarrow \infty$. It turns out that if we assume an underlying distribution density of the training data, then there exists a quadrature rule that guarantees that $\gamma\_{\ell,n} \rightarrow 0$ as $n \rightarrow \infty$ as long as the density is supported on the entire $\mathbb{S}^{d-1}$. For example, using $c_i = 1/n p(\mathbf{x}\_i)$ gives Monte Carlo convergence $\mathcal{O}(1/\sqrt{n})$ where $p(\mathbf{x})$ is the estimated density (e.g., see the textbook _Density estimation for statistics and data analysis_ by Bernard Silverman) On the other hand, if the density is not supported on an open set of $\mathbb{S}^{d-1}$, then we cannot find such a quadrature rule. However, this also means that we can never recover the entire target function from the training data, regardless of the method we use. Since the frequency we studied is a global notion, the missing support prevents us from making frequency bias a mathematically well-defined term. Hence, we only consider the case where the distribution of the training data is a.e. supported on $\mathbb{S}^{d-1}$.
>
> * "On page 6 in Theorem 3…":
>
> We have added a sentence mentioning the definition of $\gamma\_{n,\ell}$.
>
> * "On page 8…":
>
> We have mentioned that the quadrature weights are computed using a constrained quadratic program and referred interested readers to Appendix F.1.
>
> * "On page 9…":
>
> We have added a short conclusion to the paper. The position of Figure 4 is a bit hard to change because it cannot be moved to the next page (page 10). Instead, we rephrased a couple of sentences in Section 6.3 to remove possible confusion.
>
> * "On page 1…":
>
> Yes, the referee is right as we meant $\mathbf{H}^\infty$ and $\mathbf{P}$ are SPD. We have corrected this in our manuscript. Thank you very much for the careful review.

---

> > ### Comment · Reviewer_XAwZ · 2022-11-13
> > **Response to author's initial response.**
> >
> > Thank you for your response.  I am mostly satisfied with the revisions, however I do have a couple remaining questions.
> >
> > My remaining questions are in response to your statements "Thank you for bringing up this subtle point. The learning rate stated in both [Arora et al., 2019] and [Basri et al., 2019] is $O(\lambda_0 / n^2)$. In their papers, if instead the mean-squared error is being used, then the $1/n^2$  indeed disappears. However, the learning rate still depends on $\lambda_0$, which vanishes as the number of training data goes to infinity since the eigenvalues of $K^\infty$  have a limit point at $0$. Instead, our learning rate does not depend on $\lambda_0$. This issue was discussed more rigorously in detail in section 4.1 of [Su and Yang, 2019]. We have rephrased the corresponding sentence in our paper to make it fairer."
> >
> > * In regard to "The learning rate stated in both [Arora et al., 2019] and [Basri et al., 2019] is $O(\lambda_0 / n^2)$".  Which result in Basri et al 2019 are you referring to?  The reference you link to in the manuscript is [3].  [3] characterize the NTK spectrum, they do not introduce optimization guarantees like [4] (Arora et al.).  Thus I don't see which vanishing convergence rate issue they have in that paper, since they seek to characterize the spectrum and not provide optimization guarantees.  Perhaps you are confusing with their more recent work [1] https://arxiv.org/pdf/2003.04560.pdf which offers an optimization guarantee in Theorem 2?
> >
> > * In regard to "However, the learning rate still depends on $\lambda_0$, which vanishes as the number of training data goes to infinity since the eigenvalues of $K^\infty$  have a limit point at $0$". Again this issues is more subtle due to the normalization.  The eigenvalues of $\frac{1}{n} H^\infty$ converge to $K^\infty$ (whose eigenvalues converge to zero), however in [4] $\lambda_0$ is the smallest eigenvalue of $H^\infty$ _not_ $\frac{1}{n} H^\infty$.  The smallest eigenvalue of $H^\infty$ is $\Omega(1)$ for fairly generic data, see for example [2].
> >
> > * In the manuscript you state "So, we overcome the vanishing convergence rate issue due to the diminishing $\lambda_0$ as $n \rightarrow \infty$ that appears in previous analyses of frequency biasing".  Again in your result Theorem 1 in the worst case when the labels $y$ lie in the bottom eigenspace of $P H^\infty$ the convergence rate is of order $\eta \lambda_{min}(P H^\infty)$.  Thus if you set $P = 1/n$ since $\eta = O(1)$ in the worst case this behaves like $\lambda_{min}(H^\infty) / n$ which goes to zero.  This is intrinsic to the NTK regime and can't be overcome.
> >
> > In sum, I do not think the authors have fully appreciated the subtleties of comparing with results that use a different normalization for the loss function.
> >
> > References:
> >
> > [1] Ronen Basri, Meirav Galun, Amnon Geifman, David Jacobs, Yoni Kasten, and Shira Kritchman.
> > Frequency bias in neural networks for input of non-uniform density. In Proceedings of the 37th
> > International Conference on Machine Learning, volume 119 of Proceedings of Machine Learning
> > Research, pp. 685–694. PMLR, 2020. URL https://proceedings.mlr.press/v119/
> > basri20a.html
> >
> > [2] Quynh Nguyen, Marco Mondelli, and Guido Montufar. Tight bounds on the smallest eigenvalue ´
> > of the neural tangent kernel for deep ReLU networks. In Proceedings of the 38th International Conference on Machine Learning, volume 139 of Proceedings of Machine Learning Research, pp. 8119–8129. PMLR, 2021. URL https://proceedings.mlr.press/v139/
> > nguyen21g.html.
> >
> > [3] R. Basri, D. Jacobs, Y. Kasten, and S. Kritchman. The convergence rate of neural networks for
> > learned functions of different frequencies. Adv. Neur. Info. Proc. Syst., 32, 2019.
> >
> > [4] S. Arora, S. S. Du, W. Hu, Z. Li, and R. Wang. Fine-grained analysis of optimization and generalization for overparameterized two-layer neural networks. In Inter. Conf. Mach. Learn., pp. 322–332.
> > PMLR, 2019.

---

> > > ### Author Response · Authors · 2022-11-13
> > > **Response to the Reviewer's Follow-up Questions**
> > >
> > > Thank you for carefully going through our revisions. We have reviewed the follow-up questions raised by the reviewer and were convinced that the comparison to [Arora et al., 2019] and [Basri et al., 2019] that used to appear after Theorem 1 is problematic. As pointed out by the reviewer, as $n$ goes to infinity, there exists an $\mathbf{y}$ depending on $n$ such that the rate of convergence vanishes. This issue is intrinsic to the NTK analysis and occurs in both [Arora et al., 2019] and our work. Nevertheless, if the target function is bandlimited, then one can show that the convergence rate in early epochs stays constant. We have removed the problematic comparison after Theorem 1 and replaced it with this new message, which can be seen in [Su and Yang, 2019], [Cao et al., 2019], and [Basri et al., 2019] (in the uniform case), and then suggested that the similar observations are made in our paper. Again, we are extremely grateful to the reviewer for preventing us from making an unfair comparison.

---

> > > > ### Comment · Reviewer_XAwZ · 2022-11-13
> > > > **Response to author(s)**
> > > >
> > > > I looked at the new revision and find the new discussion accurate.  Thank you and good work.

---

### Official Review · Reviewer_F3bk · 2022-10-30

**Confidence:** 3
**Correctness:** 3
**Technical Novelty And Significance:** 3
**Empirical Novelty And Significance:** 2
**Recommendation:** 6

**Clarity, Quality, Novelty And Reproducibility:**

Clarity:

The motivation to study the problem is not clear and requires more explanation. Some sentences and definitions seem incompplete/imprecise and needs more elucidation.

e.g. in the para before Section 4 :  "our rate of convergence does not vanish ..." needs more clarification.
e.g. Equation 15: E_c is defined for single functions over the domain. Does E_c[g_j g_l] stand for E_c of the pointwise product of the two functions g_j, g_l? If so, this needs clarification.





**Strength And Weaknesses:**

Strengths:

The paper is well-written and the key ideas in the Theorems are reasonably easy to understand.

Weaknesses:

1. The main issue I have with the paper is the objective of training. The paper assumes one has access to training data from a non-uniform distribution, but the final goal is to minimize error according to the uniform distribution (or L2 loss). This looks like a domain adaptation setting. If this is indeed the goal of the paper this should be made clear. Even if this is so, it seems like a very restricted setting of domain adaptation.

2. For example, in Figure 1, what is the loss (and the final/intermediate trained model output) for training with $\Phi$ and $\tilde\Phi$? Looking at how all the three dashed curves are below the solid curves seems to indicate that the model trained with $\tilde \Phi$ is strictly better than the model trained with $\Phi$ at all points, but that seems at odds with the intuition that the model trained with $\Phi$ would be better in the region of the circle with more data points than average.


3. The Sobolev norm frequency biasing doesn't seem to do any meaningful tradeoffs (in Figure 2). i.e. choosing a larger value of s does seem to make the loss on low-frequency components significantly worse, but the loss on high-frequency doesn't seem to reduce nearly as much.  Is the main claim that choosing negative s and emphasising low-frequency components in the optimisation better than not emphasising any particular component by choosing s=0? If so, couldn't this same effect be achieved by NTK from a different initialisation with smaller variance?

4. Similar comments on Figure 3. Choosing s=-1 makes the red and yellow curve higher than s=0, without meaningfully lowering the blue curve.

5. MNIST experiment: There seems to be some confusion between high-frequency label noise components on the dataset and high-frequency noise components on a single image. The low-frequency bias on NTK says nothing about high or low frequency noise on a single image.


======================

After discussion phase. Having made the domain adaptation setting made clear solves several of the issues I had with the paper. I am raising my score to a marginal accept.


**Summary Of The Paper:**

Summary:

The paper considers the case of non-uniform data in a sphere, and gives a way to demonstrate the frequency bias of neural net learning in the NTK regime.


**Summary Of The Review:**

The paper tackles a novel problem whose significance seems questionable. The technical tools developed seem somewhat useful nevertheless.

---

> ### Author Response · Authors · 2022-11-12
> **Response to the Reviewer (2/2)**
>
> * "The Sobolev norm frequency biasing...":
>
> When we change $s$, we also change the SPD matrix $\mathbf{P}\_s$ that defines the loss and hence the constant $M\_{\mathbf{P}\_s}$. The larger the $s$, the larger the $M\_{\mathbf{P}\_s}$, and, by Theorem 1, our step size $\eta$ needs to be smaller. Intuitively, one could interpret it as the $H^s$ loss' gradient having a Lipschitz constant $L_s$ that increases with $s$. For gradient descent convergence, the step size $\eta \sim 1/L\_s$ has to be smaller for larger $s$. The referee also points out a great aspect of changing $s$: using smaller $s$ could lower the Lipschitz constant and thus increase step size (learning rate), while bigger $s$ could suffer from slow convergence under the gradient descent framework.
>
> Apart from reflecting the impact on step size from different $H^s$ loss,  Figure 2 demonstrates that by changing $s$, we can change the _relative_ convergence speed among different frequencies of the mismatch (residual), i.e., the order of convergence. In short, using different $H^s$ norms intrinsically **changes the convergence order among different frequency components**. For example, consider low-frequency noises in the training data with the $L^2$ loss, this low-frequency noise will always be learned faster than the high-frequency content, resulting in overfitting. Decreasing the variance of the initialization will only make Theorem 2 more "accurate" in the sense that $\varepsilon_3$ is smaller, but it does not change the fact that $L^2$ training loss overfits low-frequency noise. Instead, if we use the $H^s$ loss with $s>0$, we have to use a smaller step size $\eta$, which slows down convergence. However, it changes the order of fitting the data: high-frequency residual will be learned first, and low-frequency noises will be learned much later (possibly not learned at all under early stopping). The change of order makes a qualitative difference in terms of avoiding overfitting.
>
> * "Similar comments on Figure 3...":
>
> This is also addressed by the point above.
>
> * "MNIST experiment: There seems to be...":
>
> Thank you for pointing out this issue. There was indeed a gap between the theoretical part of the paper and the MNIST experiment. We have bridged this gap by adding Appendix G, which formally discusses frequency bias in operator learning. While this experiment is not a direct verification of Theorem 4, it still falls under the NTK regime, which is extensively considered in our paper. In addition, we hope to use this experiment to convey an important message of the paper: the $H^s$ loss is useful in tuning frequency bias.
>
> * "In the para before Section 4...":
>
> We have provided a more formal definition of the convergence rate. We then discussed the issue of the "vanishing convergence rate" and suggested that the issue can be overcome if $g$ is bandlimited.
>
> * "Equation 15: $E\_\boldsymbol{c}$ is defined...":
>
> Yes, the product is entrywise. We have clarified this notation. Thank you for pointing it out.

---

> ### Author Response · Authors · 2022-11-12
> **Response to the Reviewer (1/2)**
>
> We thank the reviewer for the careful review and comments. Below we address the concerns and questions raised in this review.
>
> * "The main issue I have with the paper...":
>
> The main goal of this paper is to analyze and tune the learning rates of different frequencies. While training aims to minimize the $L^2$ loss (population or empirical risk), in practice, people also care about the relative decaying speed for each spectral component (frequency) of the data mismatch. We have clarified this motivation in the introduction. Therefore, the main objective of the paper is to understand the order of $\widehat{\big(\mathcal{N}-g\big)}\_{\ell,p}$ decay instead of $\\|{\mathcal{N}-g}\\|\_{L^2}$. Understanding the spectral property of the training given nonuniform data has always been challenging, even for classical kernel regression [Williams-Rasmussen, 2006]. It is known that the deviation of the "nonuniform distribution" from the "uniform loss" makes frequency bias so intricate [Basri et al., 2020]. One main contribution of this work is to tackle this challenge of nonuniform data training by pointing out that frequency bias cannot be guaranteed under the common loss $\Phi$ and rigorously proving the clear low-frequency bias under the loss $\tilde{\Phi}$. Furthermore, we prove that changing the loss function from $L^2$ to the general $H^s$-based Sobolev norms, in either the domain or co-domain, can further tune the frequency bias in NN training, which provides extra freedom for practitioners.
>
>
> * "For example, in Figure 1, what is the loss...":
>
> We have added figures to show the evolution of the NNs trained with $\Phi$ and $\tilde{\Phi}$ in Figure 5 of Appendix F. It shows that the NN trained with $\Phi$ is more accurate in the region where we have more training data, whereas the NN trained with $\tilde{\Phi}$ has better overall accuracy on the entire domain. This aligns with our intuition and does not contradict Figure 1. In Figure 1, the $y$-axes are the absolute values of the (discrete) Fourier coefficients of $\mathcal{N} - g$, where $g = \sum\_{\ell=1}^9 \sin(\ell\mathbf{x})$ is the target function and $\mathcal{N}$ is the NN. Roughly speaking, if we expand $\mathcal{N} - g$ uniquely as
> \\[
>     \mathcal{N}(\mathbf{x}) - g(\mathbf{x}) = a + \sum\_{\ell=1}^\infty (c\_\ell \sin(\ell\mathbf{x}) + d\_\ell \cos(\ell\mathbf{x})), \qquad a, c\_\ell, d\_\ell \in \mathbb{R},
> \\]
> then what we plotted in Figure 1 are approximately $\sqrt{c_\ell^2 + d_\ell^2}$ for $\ell = 1, 5, 9$. Intuitively, this quantity is "the amplitude difference at frequency-$\ell$ between $\mathcal{N}$ and $g$." The definition and computation of the quantities are also discussed in Appendix F. We also added a reference to it in the main text. Moreover, by Parseval's identity, we have
> \\[
>     \\|{\mathcal{N}-g}\\|\_{L^2}^2 = 2\pi a^2 + C\sum\_{\ell=1}^\infty (c\_\ell^2 + d\_\ell^2),
> \\]
> where the $L^2$-norm is with respect to the Lebesgue measure and $C > 0$ is a normalizing constant. Since Figure 5 shows that the NN trained with $\tilde{\Phi}$ is overall more accurate on the entire domain (i.e., the $L^2$ norm is much smaller), it is not surprising that in Figure 1, the quantities associated with $\tilde{\Phi}$ are smaller than those associated with $\Phi$. Nevertheless, the main message conveyed by Figure 1 is that while using $\Phi$ does not always guarantee the frequency bias proved in theory, using $\tilde{\Phi}$ fixes this issue.

---

> > ### Comment · Reviewer_F3bk · 2022-11-13
> > **Thanks for the reply**
> >
> > Thanks for the reply. Couple more questions
> >
> >
> > 1. The training data is non-uniform on the sphere but you care about the L2 loss (or equivalently expected loss with uniform distribution on the sphere) and its frequency components. Why? Why are train and test data different? This needs to be clarified. (i.e. is the algorithm in the standard ML setting where train and test are from the same distribution, or in the domain adaptation setting where they are different)
> >
> > 2. If we are in the standard ML setting, then error on the part of the sphere close to the train data should matter more than the error on the part of the spehere containing no support from the data distribution. This is not captured by frequency component analysis. (Appendix Figure 5).
> >
> > 3. For the purpose of the message in the last line of the review reply. "... using $\tilde \Phi$ fixes the issue.". I don't think there is an "issue" to be fixed here. If train and test distributions are same, one worries more about total loss than individual components. Frequency bias is interesting because it explains some of the overfitting robustness of a neural net. But if ensuring that property causes the training loss or test loss on the actual data to decrease, it is not a desired trait.

---

> > > ### Author Response · Authors · 2022-11-14
> > > **Response to the Reviewer's Follow-up Questions**
> > >
> > > We thank the reviewer for the thoughtful follow-up questions. This gives us an excellent chance to explain the motivation for our setup. We have revised the second paragraph of the introduction to highlight these points to the reader.
> > >
> > > * "The training data is non-uniform on the sphere but you care about the $L^2$ loss..."
> > >
> > > We regard the ML problem as learning a function $g(\mathbf{x})$ on $\mathbb{S}^{d-1}$. Training data being nonuniform on the sphere is a limitation of the availability of training data and could be due to undersampling. Of course, we would prefer the training data to be uniform, but nonuniform distribution could be what is given. Moreover, besides the generalization error, we are interested in the convergence rate of frequency components, which is relevant when the training data is noisy.
> > >
> > > * "If we are in the standard ML setting, then error..."
> > >
> > > There are several situations where the training and test data have different distributions. An example that is quick to describe and is extremely popular these days is in physics-informed neural networks (PINNs) for PDE learning. Here, one typically places more samples where the PDE solution is believed to have a singularity or is oscillatory [Mao et al., 2020]. Therefore, nonuniform training data is used to capture the non-regular behavior of the solution, but one is still interested in the uniform approximation accuracy across the whole solution domain.
> > >
> > > * "For the purpose of the message in the last line of the review reply..."
> > >
> > > We totally agree with the referee in the sense that if the training data and test data both follow the same distribution, then there is no issue to fix. However, if one cares about the standard $L^2$ generalization error but nonuniform training data are given (i.e., training and test data have different distributions), our framework helps well-approximate the target $g$ in terms of its natural basis in the standard $L^2$ functional domain rather than "over-fits" the nonuniform training data without approximating the low-frequency modes of the function.
> > >
> > > We address that using a new loss function defined by the matrix $\mathbf{P}$ does _not_ prevent the training error from being arbitrarily small (see Theorem 1). In the discrete setting, $\exists C\_1, C\_2 >0$ (that depend on $\mathbf{P}$) such that $C_1 \Phi \leq \tilde{\Phi} \leq C\_2 \Phi$, showing both norms can be made small by training. However, norm equivalence does not imply similar convergence properties in NN training gradient dynamics (e.g., $\ell^1$ and $\ell^2$ are norm-equivalent but have different traits in optimization). One of our main messages is that even when given nonuniform training data, with $\tilde{\Phi}$ and its generalization ($H^s$ loss), one can carefully control the spectral convergence properties of the NN's misfit, which makes it more robust to noise and avoids overfitting.
> > >
> > > Z. Mao, A.D. Jagtap, and G.E. Karniadakis. Physics-informed neural networks for high-speed flows. _Computer Methods in Applied Mechanics and Engineering_, 2020.

---

### Author Response · Authors · 2022-11-12
**A List of Changes Made in Rebuttal Revision**

We are very thankful to the reviewers for their many valuable questions and comments. In addition to our response to each reviewer, we provide a list of nontrivial rebuttal changes to the manuscript. We hope that the reviewers will update their evaluations considering the revisions.

* In order to provide more theoretical background to the experiment on the MNIST dataset, we added Appendix G to discuss frequency bias in operator learning and relate it to the squared-$H^s$ loss.

* Immediately after the proofs of Theorem 1, 2, and 4, we added Theorem 1', 2', and 4', respectively. These results are almost equivalent to the ones presented in the main text. However, instead of posing requirements on the NN parameters and the algorithm parameters (e.g., $m, n, \eta$, etc.) to guarantee an error bound $\epsilon$, we explicitly show how the error bound depends on the parameters. We hope this adds clarity to the role of each parameter in the theorem.

* In Appendix D.1 and E.2, after the proof of Theorem 2 and 4, respectively, we added discussions of how one can obtain upper bounds on the generalization errors from the theorems.

* In Appendix F.1, we refined our discussion of the computation of the quadrature rule in the first experiment by adding more details on the algorithm and motivating the objective using Theorem 3.

* In Appendix F.1, we provided figures that show the evolution of the NNs trained by $\Phi$ and $\tilde{\Phi}$, respectively, which aligns with our intuition.

* We slightly rewrote the introduction to provide more motivation for our work. In particular, we emphasized that, apart from the $L^2$ generalization error, the convergence rate for each spectral component is important because it tells us the robustness of the NN under noises of different frequencies.

* In Appendix A, we remarked that the operator defined by $\mathbf{H}^\infty \mathbf{P}$ is self-adjoint in the inner-product space induced by $\mathbf{P}$.

* In Section 2, we showed that $\mathbf{H}^\infty \mathbf{P}$ is diagonalizable.

* In Section 3, we corrected a mistake we made in discussing the "vanishing convergence rate" issue and suggested that a bandlimited target function does not suffer the "vanishing convergence rate."

* Finally, we have added a short conclusion.

---

### Decision · Program_Chairs · 2023-01-20

**Decision:**

Accept: poster

**Justification For Why Not Higher Score:**

While the extension of special bias to non-uniform distribution is interesting, the result is limited to the neural tangent kernel regime

**Justification For Why Not Lower Score:**

N/A

**Metareview: Summary, Strengths And Weaknesses:**

Based on the observation that most training data sets are not drawn from uniform distribution over a sphere, This paper provides a NTK-based analysis for two-layer neural networks to quantify the frequency biasing of NN training over non-uniform data.

Strengths:

+The paper is well written

+The extension of special bias to non-uniform data is interesting.

Weaknesses:

-The result is limited to the neural tangent kernel regime

The reviewers are generally positive about this paper after the author's response and discussion. Thus, I recommend acceptance.


**Note From Pc:**

if the above contains the word "oral" or "spotlight" please see: "oral" presentation means -> notable-top-5% and "spotlight" means -> notable-top-25%. As stated in our emails, we are disassociating presentation type from AC recommendations

**Summary Of Ac-Reviewer Meeting:**

N/A